# Optimal Learning from Label Proportions with General Loss Functions

**Lorne Applebaum** [1]  **Travis Dick** [1]  **Claudio Gentile** [1]  **Haim Kaplan** [2]  **Tomer Koren** [2]

## Abstract

Motivated by problems in online advertising, we address the task of Learning from Label Proportions (LLP). We introduce a novel and versatile low-variance debiasing methodology to learn from aggregate label information, significantly advancing the state of the art in LLP. Our debiasing approach exhibits remarkable flexibility, seamlessly accommodating a broad spectrum of practically relevant loss functions across both binary and multi-class classification settings. By carefully combining our estimators with standard techniques, we improve sample complexity guarantees for a large class of losses of practical relevance. We also empirically validate the efficacy of our proposed approach across a diverse array of benchmark datasets, demonstrating compelling empirical advantages over standard baselines.

## 1. Introduction

Learning from Label Proportions (LLP) is a type of weakly supervised problem where, instead of individual labels for each training instance, the algorithm has only access to the proportion of different labels within groups of instances, called *bags*. This may occur due to privacy/legal restrictions (Rueping, 2010; Wojtusiak et al., 2011), cost of supervision (Chen et al., 2004), lack of resolution in labeling instruments (Dery et al., 2018), etc.

While initial formulations of LLP date back to at least 15 to 20 years ago (e.g., (Chen et al., 2004; de Freitas & Kuck, 2005; Musicant et al., 2007; Quadrianto et al., 2008; Rueping, 2010)), it is only with the recent changes in the Web advertising industry that we see a substantial resurgence of interest in this problem. Web advertising, arguably one of the largest scale real-world applications of Machine Learning, is undergoing important changes in recent years.

Advertisers (or other entities working on their behalf, often called *AdTechs*) train models for predicting the chance that a user will *convert* (e.g., sign up for an account, buy a product, etc.) after they clicked advertisements posted on a publisher website. The output of such models is typically used as inputs when computing the bid price in online auctions that power automated bidding over the Web. Now, since the ad click and conversion events happen on different websites (the auction and the click event is on a publisher website, the conversion event is on an advertiser website), collecting training labels amounts to linking the two events, thus tracking the user behavior across the two sites.

Historically, third-party cookies or link decoration has made this straightforward. Nonetheless, even acknowledging the critical role that online advertising is playing on the web (and the impossibility of perfect user tracking protection), growing privacy concerns have led major web browsers to introduce ad hoc APIs to measure ad performance while, at the same time, protecting user privacy. These include APIs from Apple's Safari (Wilander, 2019), and Mozilla's Firefox (Crouch & Crawford, 2022) which have both deprecated third-party cookies in past years. These APIs allow AdTechs to collect cross-site data only *in aggregate*. For instance, from the publisher viewpoint, where the auction is run, it is of paramount importance to obtain effective *per-interaction* predictions. Yet, what the publisher observes is only a list of ad interactions (the click events) and an *unordered* list of conversion events (that is, their *number*) that originate from those interactions on the advertiser side. We view LLP as a principled technique for making the best use of these APIs for training conversion models.

Given the above context, in this paper we unveil a versatile and effective debiasing methodology that extracts signals from aggregate label information. This novel low-variance approach represents an important leap forward in LLP, as it allows us to effortlessly accommodate a panoply of practically relevant loss functions, in both binary and multi-class classification settings. Below we discuss our contribution in the context of the relevant LLP literature.

**Related work.**  As elucidated, e.g., by Franco et al. (2023), many variants of LLP exist depending on the postulated relationships among bags, features and labels. We consider here the variant where bags are generated at random, *inde-*

---

[1]Google Research, NY, USA [2]Google Research and Tel Aviv University, Tel Aviv, Israel. Correspondence to: Claudio Gentile <cgentile@google.com>.

*Proceedings of the 43rd International Conference on Machine Learning*, Seoul, South Korea. PMLR 306, 2026. Copyright 2026 by the author(s).

*pendent* of the data content. While relatively simple, we shall see that this setting already poses substantial technical challenges when it comes to studying sample complexity guarantees. The most relevant papers to our work in this setting are (Busa-Fekete et al., 2023; Li et al., 2024; Busa-Fekete et al., 2025). We defer to Appendix A for a more comprehensive survey of the LLP literature.

Like our work, the three aforementioned studies operate within a standard statistical learning framework with random, non-overlapping bags, but unlike ours, they are limited to *binary* classification. Busa-Fekete et al. (2023) propose a general LLP reduction to standard learning, applicable to any bounded loss, which turns aggregate labels into unbiased individual estimates, albeit increasing variance. They show $k/\sqrt{n}$ regret guarantees for methods like ERM and SGD, where $n$ is the sample size and $k$ is the bag size. Li et al. (2024) and Busa-Fekete et al. (2025) concentrate solely on square loss, and improve in various ways over (Busa-Fekete et al., 2023). Specifically, Li et al. (2024) achieve fast rates of the form $k^3/n$ in (restricted) realizable settings, subsequently improved to the optimal rate $k/n$ in (Busa-Fekete et al., 2025). The latter paper also contains non-realizable sample complexity bounds of the form $\sqrt{k/n}$, again improving over (Busa-Fekete et al., 2023) when the loss is the square loss.

**Contributions.** Inspired by Busa-Fekete et al. (2023)'s broad scope, our work refines and significantly extends prior LLP investigations. Notably, our analysis embraces a diverse spectrum of practical, even unbounded, loss functions—unlike the constraints in (Busa-Fekete et al., 2023; Li et al., 2024; Busa-Fekete et al., 2025). We refine and significantly extend these prior investigations in multiple ways. For instance, we show that the regret guarantees of $\sqrt{k/n}$ for the non-realizable setting can also be achieved for arbitrary binary and multi-class classification scenarios with unbounded losses (like the log loss/cross entropy loss) under bounded moment conditions (Theorem 3.2 and Theorem C.2). We also extend Busa-Fekete et al. (2025)'s optimal $k/n$ realizable regret far beyond square loss (Theorem 3.5). More importantly, we provide the first unified treatment of multi-class LLP. Within this framework, we introduce a fundamental distinction between *full histogram* and *total* multi-class scenarios, based on the granularity of class-wise aggregate label information. This practically motivated differentiation directly dictates regret guarantees: full histogram often yields $c$-independent bounds ($c$ being the number of classes) whereas total multi-class regret scales as $c^2$. Further, we report the results of a suite of experiments on a diverse set of real-world benchmarks, where we compare our debiasing technique to representative baselines available in the literature having the same breadth of applicability. We follow, whenever possible, prior experimental setups (like the one in (Busa-Fekete et al., 2023)), but choose

to report log loss test set results, as more in line with the afore-mentioned motivation about conversion prediction. Though somewhat preliminary in nature, these experiments reveal the effectiveness of our method, specifically when dealing with large bag sizes. This is indeed the scenario we are mostly concerned with for Web advertising, where the aggregate signal might even be at the level of ad campaigns (hence, very large bags).

## 2. Preliminaries and Notation

We denote by $\mathcal{X}$ the input space (sometimes called feature space or instance space), and by $\mathcal{Y}$ the label space, which can be either binary, $\mathcal{Y} = \{0, 1\}$, or multi-class, $\mathcal{Y} = \{0, \ldots, c - 1\}$. We denote by $\mathcal{D}$ an unknown distribution over $\mathcal{X} \times \mathcal{Y}$, and denote the corresponding random variables by $x$ and $y$. We also denote by $\mathcal{D}_{\mathcal{Y}|x}$ the conditional distribution of random variable $y$ given $x$. For each class $r = 0, \ldots, c - 1$, we denote by $\eta_r(x)$ the probability $\eta_r(x) = \mathbb{P}_{\mathcal{D}_{\mathcal{Y}|x}}(y = r|x) = \mathbb{E}_{y \sim \mathcal{D}_{\mathcal{Y}|x}}[\{y = r\}|x]$, where $\{\cdot\}$ is the indicator function of the predicate at argument. We shall often gather such probabilities into a $c$-dimensional (column) vector $\eta(x) = [\eta_0(x), \ldots, \eta_{c-1}(x)]^\top$. Note that $\sum_{r=0}^{c-1} \eta_r(x) = 1$ for all $x \in \mathcal{X}$. In the binary case, where $y \in \{0, 1\}$, there is no need for a vector notation, and we simply define $\eta(x) = \mathbb{P}_{\mathcal{D}_{\mathcal{Y}|x}}(y = 1|x) = \mathbb{E}_{y \sim \mathcal{D}_{\mathcal{Y}|x}}[y|x]$ to be the probability of drawing label 1 conditioned on feature vector $x$. When clear from the surrounding context, we shall often shorten or remove the subscripts from probabilities and expectations and write, e.g., $\mathbb{E}_{(x,y)}[f(x, y)]$, instead of $\mathbb{E}_{(x,y) \sim \mathcal{D}}[f(x, y)]$, and $\mathbb{E}[y|x]$ instead of $\mathbb{E}_{y \sim \mathcal{D}_{\mathcal{Y}|x}}[y|x]$. No ambiguity will arise.

We take the standard viewpoint of statistical learning, and operate within a given function class $\mathcal{H}$ (sometimes called hypothesis space or hypothesis class), where each $h \in \mathcal{H}$ is a deterministic mapping $h : \mathcal{X} \rightarrow \mathcal{Z}$, where $\mathcal{Z}$ is some *output* space. In the binary label case, we shall work with $\mathcal{Z} = [0, 1]$, so that $h(x)$ can be interpreted as the probability that $y = 1$ given $x$ according to model $h$. In the multi-class case, the output space $\mathcal{Z}$ will depend on the loss function (see below). For instance, we may have $\mathcal{Z} = \Delta_c$, the $c$-dimensional probability simplex $\Delta_c = \{(h_0, \ldots, h_{c-1}) : h_i \geq 0, \sum_{r=0}^{c-1} h_r = 1\}$, thus $h_r(x)$ can be interpreted as the probability that $y = r$ given $x$ according to model $h$.

A loss function $\ell : \mathcal{Z} \times \mathcal{Y} \rightarrow \mathbb{R}^+$ measures the discrepancy between $h(x)$ and $y$. Notable examples of loss functions where $\mathcal{Z} = \Delta_c$ include the square loss $\ell(h(x), y) = \frac{1}{c} \sum_{r=0}^{c-1} (\{y = r\} - h_r(x))^2$ (sometimes called *Brier* score), which reduces in the binary case to the more familiar expression $\ell(h(x), y) = (y - h(x))^2$, and the log loss $\ell(h(x), y) = -\log h_y(x)$, which in the binary case can be rewritten as $\ell(h(x), y) = -y \log h(x) - (1 - y) \log(1 - h(x))$. Another loss function, often used to train

and evaluate conversion prediction models in ad domains, is the (capped) Poisson Log Loss, where $\mathcal{Z} = [0, c - 1]$, and $\ell(h(x), y) = h(x) - y \log h(x)$, see, e.g., (Li et al., 2010).

Disregarding the capping, this can be seen as the negative log likelihood of $y$ generated by Poisson distribution with parameter $h(x)$. Hence minimizing this loss is equivalent to maximizing the log likelihood of the parameter $h(x)$ given the observation $y$.

Following very recent works on the subject (Busa-Fekete et al., 2025), we decided to skip the heavy machinery of empirical process theory (e.g., (Bartlett et al., 2005; Boucheron et al., 2012; Vershynin, 2018; Lugosi & Mendelson, 2016; Høgsgaard & Paudice, 2025)), and restrict to the case where the hypothesis space $\mathcal{H}$ is finite. The main ideas contained in this paper can be lifted to infinite hypothesis cases, at the cost of mathematical details which are, however, not specific to the LLP framework.

Given $\mathcal{D}$, $\mathcal{H}$, and $\ell$, the *population loss* (or statistical *risk*) of $h \in \mathcal{H}$ is $\mathcal{L}(h) = \mathbb{E}_{(x,y)\sim\mathcal{D}}[\ell(h(x), y)]$. The *excess risk*, or (simple) *regret* $Reg(h)$ of $h$ is defined as $Reg(h) = \mathcal{L}(h) - \mathcal{L}(\widehat{h}^\star)$, where $\widehat{h}^\star = \arg\min_{h \in \mathcal{H}} \mathcal{L}(h)$ is the *best-in-class* hypothesis, that is, the hypothesis in $\mathcal{H}$ having the smallest population loss. The learning setting is *realizable* when the Bayes-optimal predictor $h^\star = \arg\min_{h : \mathcal{X} \to \mathcal{Z}} \mathcal{L}(h)$, belongs to $\mathcal{H}$ (here the minimum is taken over all possible (measurable) functions), and is otherwise *non-realizable*. Thus, in the realizable setting, $\widehat{h}^\star = h^\star$.

### 2.1. Our Learning with Label Proportions problem

An *example* is a pair $(x, y)$ drawn according to $\mathcal{D}$. A dataset $S$ of size $n$ is an i.i.d. sequence of examples $S = (x_1, y_1), \ldots, (x_n, y_n)$. In Learning with Label Proportions (LLP), the feature vectors $x_i$ in $S$ are available in the clear, while the labels are obfuscated by randomly grouping them into $m$ *bags* of a given size $k$, with $n = m \times k$. That is, the dataset $S$ is only available in the form

$$S = \underbrace{((x_{1,1}, \ldots, x_{1,k}), \alpha_1)}_{(\mathcal{B}_1, \alpha_1)}, \ldots, \underbrace{((x_{m,1}, \ldots, x_{m,k}), \alpha_m)}_{(\mathcal{B}_m, \alpha_m)},$$
(1)

where $\alpha_j$ is the aggregate label value associated with the $j$-th bag $\mathcal{B}_j = \{x_{j,i} : i \in [k]\}$. In the binary label case, $\alpha_j = \frac{1}{k} \sum_{i=1}^{k} y_{j,i}$ is simply the label proportion (fraction of positive labels) in the $j$-th bag. In the interpretation of click events and conversion events we alluded to in the introduction, $x_{j,1}, \ldots, x_{j,k}$ encode a sequence of (publisher side) click events and associated features occurring either within a given time frame or within a campaign (or aggregated in some other way), and $k\alpha_j$ is the *number* of conversions associated with those clicks on the advertiser side. The multi-class case is a scenario where multiple conversions may be associated with a given click. In this case, we need

to separate two kinds of aggregate labels, corresponding to two different practical settings. We are in the *full histogram* multi-class case when the label proportion for the $j$-th bag is a $c$-dimensional vector $\boldsymbol{\alpha}_j = [\alpha_{j,0}, \ldots, \alpha_{j,c-1}]^\top$ where, for each class $r \in \{0, \ldots, c-1\}$, $\alpha_{j,r} = \frac{1}{k} \sum_{i=1}^{k} \{y_{j,i} = r\}$ is the fraction of times label $r$ occurs in the $j$-th bag (note that $\boldsymbol{\alpha}_j$ is a probability vector, in that $\sum_{r=0}^{c-1} \alpha_{j,r} = 1$). We are in the *total* multi-class case when the classes $0, 1, \ldots, c-1$ can themselves be interpreted as *counts* (e.g., $y$ is the number of purchases associated with a given user $x$ within a given time frame, capped at $c - 1$) and, for each bag $j$ only the overall fractional count $\alpha_j = \frac{1}{k} \sum_{i=1}^{k} y_{j,i}$ within the bag is available. Note that in the binary case, the two notions collapse, and are in turn equivalent to the above-mentioned fraction of positive labels in a bag. The pair $(\mathcal{B}_j, \alpha_j)$ will often be denoted by $z_j$.

Therefore, in LLP the learning algorithm receives information about the $n$ labels $y_{j,i}$ in $S$ only in the aggregate form determined by the $m$ signals $\alpha_j$. This signal is a (probability) vector $\boldsymbol{\alpha}_j$ in the full histogram multi-class case, and a single number in the total multi-class case (as well as in the binary case). Note that for a fixed dataset size $n = mk$, increasing $k$ reduces available label information. Additionally, the multi-class task is naturally more difficult than the binary task, with the total scenario being harder than the full histogram scenario.

Given a distribution $\mathcal{D}$ over $\mathcal{X} \times \mathcal{Y}$, a hypothesis space $\mathcal{H}$ and a loss function $\ell$, our goal is to find $\widehat{h} \in \mathcal{H}$ such that the population loss $\mathcal{L}(\widehat{h})$ is as small as possible with high probability over the random draw of $S$. It is important to stress that in LLP the goal is that of minimizing the population loss at the *individual* label level $y$, still receiving information in the form of *aggregate* labels (1). We aim to design LLP algorithms for $\widehat{h}$, and quantify $Reg(\widehat{h})$ in both realizable and non-realizable settings, analyzing its dependence on bag size $k$ (label obfuscation), total examples $n$, properties of the loss function $\ell$, and class count $c$. Extending beyond recent square-loss investigations ((Li et al., 2024; Busa-Fekete et al., 2025)), we cover a wide range of practical, potentially unbounded, loss functions. Utilizing a Median-of-Means Tournament (Lugosi & Mendelson, 2016) (see Section 3.1 for details), we derive sample complexity guarantees that significantly advance prior art in two key directions: general loss functions for binary classification, and multi-class classification with general losses (with either full histogram or total label information).

## 3. Bag-level Estimators for Arbitrary Losses

This section contains our general treatment for the binary classification case. Due to space limitations, the multi-class setting is mainly contained in Appendix C.

Given a model $h : \mathcal{X} \to [0, 1]$ for binary labels, consider a generic loss function $\ell(h(x), y)$. Note that, because labels $y$ are binary in $\{0, 1\}$, we can always rewrite $\ell(h(x), y)$ as

$$\ell(h(x), y) = \overbrace{\ell(h(x), 0)}^{f_1(h(x))} + y \overbrace{\left(\ell(h(x), 1) - \ell(h(x), 0)\right)}^{f_2(h(x))} .$$

Hence, for suitable functions $f_1, f_2 : [0, 1] \to \mathbb{R}$, any loss depending on binary labels is *linear* in the label $y$, and can be written as $\ell(h(x), y) = f_1(h(x)) + y f_2(h(x))$. For instance, the log loss is obtained by setting $f_1(h(x)) = \log \frac{1}{1-h(x)}$, $f_2(h(x)) = \log \frac{1-h(x)}{h(x)}$, and the square loss is obtained by $f_1(h(x)) = (h(x))^2$, and $f_2(h(x)) = 1 - 2h(x)$, where in both cases $h(x) \in [0, 1]$ for all $x$.

For bag $z = ((x_1, \ldots, x_k), \alpha)$, consider now the *bag-z* loss

$$\ell_b(h, z) = \mathbb{E}_x[f_1(h(x)) + p\, f_2(h(x))] \tag{2}$$

$$+ (\alpha - p) \left( \sum_{i=1}^{k} f_2(h(x_i)) - k\mathbb{E}_x[f_2(h(x))] \right) ,$$

where $p = \mathbb{E}[y]$. We use here and throughout the notation $\ell_b$ to emphasize the role of (2) as a bag-level loss, and to clearly differentiate it from the original loss $\ell$. Loss $\ell_b(h, z)$ is a bag-level counterpart to $\ell(h(x), y)$, based on the centered variables $\alpha - p$ and $\sum_{i=1}^{k} f_2(h(x_i)) - k\mathbb{E}_x[f_2(h(x))]$, where exact knowledge of $p$, $\mathbb{E}_x[f_1(h(x))]$, and $\mathbb{E}_x[f_2(h(x))]$ is assumed. This assumption will be lifted later. The following lemma shows that $\ell_b(h, z)$ is an unbiased estimator of $\mathbb{E}_{(x,y)}[\ell(h(x), y)]$. Moreover, the centering helps reduce the variance to a constant which depends on the loss $\ell$ and the model $h$ at hand but, crucially, is *independent* of the bag size $k$. All proofs are given in Appendix B.1.

**Lemma 3.1.** *Let* $(x_1, y_1), \ldots, (x_k, y_k) \in \mathcal{X} \times \{0, 1\}$ *be drawn i.i.d. from* $\mathcal{D}$. *Let* $z = ((x_1, \ldots, x_k), \alpha)$ *be the corresponding bag. For any function* $h \in \mathcal{H}$, *and any binary loss* $\ell(h(x), y) = f_1(h(x)) + y f_2(h(x))$, *we have*

$$\mathbb{E}_z[\ell_b(h, z)] = \mathbb{E}_{(x,y)}[\ell(h(x), y)] ,$$

*and* $\mathrm{Var}_z(\ell_b(h, z)) \leq \frac{5}{2} \mathbb{E}_x\left[\left(f_2(h(x))\right)^2\right] .$

A few comments are in order at this point. First, we shall massage Lemma 3.1 to make it useful for our analysis (see Section 3.1 below). Yet, what this lemma is intuitively saying should be clear: any loss function for which the $n$ examples $(x_i, y_i)$ are available in the clear can be replaced in expectation by a loss function where only the $m$ bags $z_j = ((x_{j,1}, \ldots, x_{j,k}), \alpha_j)$ are observable instead. Moreover, the fact that the variance of this bag-level loss is independent of $k$ makes the price for this replacement quite crisp: Instead of dealing with $n$-many supervised signals $y_i$, we make do with only $m$-many aggregate signals $\alpha_j$. This is

similar in spirit to the result contained in (Busa-Fekete et al., 2023) (Proposition 4.2 therein), where an unbiased estimator for general losses is also provided. The important difference here is that whereas the variance of the estimator in (Busa-Fekete et al., 2023) grows *linearly* with $k$ (as shown in their Theorem 4.3), the variance of our estimator is *independent* of $k$. In terms of regret, this directly translates into a bound of the form $1/\sqrt{m}$ (or $1/m$ in realizable scenarios) as opposed to $\sqrt{k/m}$, as given in (Busa-Fekete et al., 2023). Note that when all $n$ labels are available, the best possible bounds read $1/\sqrt{n} = 1/\sqrt{mk}$ or $1/n = 1/(mk)$. Second, the very same arguments above can be adapted to the case where dataset $S$ has bags of different sizes $k_1, k_2, \ldots, k_m$, with $\sum_{j=1}^{m} k_j = n$, as we do still have $m$-many supervised signals, $k$ being replaced now by $\widehat{k}$, the *average* size of the bags in $S$. Third, this is essentially the best one can hope for in terms of how regret depends on $n$, when the dataset is arranged into $m = n/k$ non-overlapping bags of size $k$. As shown in Theorem 3.1 of (Busa-Fekete et al., 2025) (see also Theorem 8 in (Li et al., 2024)), there are simple realizable problems where the regret of any algorithm cannot be better than $O(k/n) = O(1/m)$. Fourth, the centering of the variables in the estimator (2) is reminiscent of what the authors of (Busa-Fekete et al., 2025) have very recently proposed for square loss, in order to reduce variance. Yet, our estimator (2) virtually applies to any loss, not just to square loss and, even for square loss, it does *not* reduce to the estimator in (Busa-Fekete et al., 2025) (the right comparison is to what those authors call the non-clipped version of the loss at the bag level in Eq. (2) therein).

### 3.1. The Median of Means Tournament Algorithm

In order to achieve regret rates which are fast when realizable conditions allow them, it is well-known (e.g., (Massart, 2000; Mendelson, 2002; Bartlett et al., 2005)) that instead of estimating (population) losses we need to estimate loss *differences*, that is, for each $h_1, h_2 \in \mathcal{H}$, we would like to actually use the estimator

$$\Delta\ell_b(h_1, h_2; z)$$
$$= \ell_b(h_1, z) - \ell_b(h_2, z)$$
$$= \mathbb{E}_x[\Delta f_1(h_1, h_2; x)] + p\, \mathbb{E}_x[\Delta f_2(h_1, h_2; x)] \tag{3}$$
$$+ (\alpha - p) \left( \sum_{i=1}^{k} \Delta f_2(h_1, h_2; x_i) - k\mathbb{E}_x[\Delta f_2(h_1, h_2; x)] \right) ,$$

where: $\quad \Delta f_1(h_1, h_2; x) = f_1(h_1(x)) - f_1(h_2(x))$
$\quad\quad\quad\quad \Delta f_2(h_1, h_2; x) = f_2(h_1(x)) - f_2(h_2(x)) .$

In addition, since we want to cover losses of practical relevance, like the log loss, which are *unbounded*, we resort to

a Median-of-Means (MoM) estimator.[1] This is in contrast to empirical mean estimators, which are the basis of the Empirical Risk Minimization (ERM) methods in (Busa-Fekete et al., 2023; Li et al., 2024; Busa-Fekete et al., 2025). (ERM is sensitive to the range of the loss values, even if mean and variance of the loss are bounded.)

Our algorithm is thus shaped as a MoM *tournament*. We first need to recall what a MoM estimator is. Given a set of i.i.d. random variables $S = \{v_1, \ldots, v_n\}$ with finite mean $\mu$ and variance, consider the index set $[n] = \{1, \ldots, n\}$, and randomly partition it into $r$ groups $\mathcal{I}_1, \ldots, \mathcal{I}_r$, each of size $|\mathcal{I}_i| \geq \lfloor \frac{n}{r} \rfloor$. Then let $\widehat{\mathbb{E}}(\mathcal{I}_i) = \frac{1}{|\mathcal{I}_i|} \sum_{j \in \mathcal{I}_i} v_j$ be the empirical mean for the $i$-th group. Then define the MoM estimator $\widehat{\mu}_{\text{MoM}}(S)$ of $\mu$ based on $S$ as[2]

$$\widehat{\mu}_{\text{MoM}}(S) = \text{median}\left\{ \widehat{\mathbb{E}}(\mathcal{I}_1), \ldots, \widehat{\mathbb{E}}(\mathcal{I}_r) \right\}.$$

As is standard for MoM estimators (e.g., (Lugosi & Mendelson, 2016)), we shall set throughout $r = \lceil 8 \log(1/\delta) \rceil$ (and assume in all our applications of MoM that $n \geq r$), where $\delta \in (0, 1)$ is the desired confidence level.

Now, in order to remove the knowledge of $p$, $\mathbb{E}_x[\Delta f_1(h_1, h_2, x)]$, and $\mathbb{E}_x[\Delta f_2(h_1, h_2; x)]$ in (3), we replace the three expectations by empirical estimates on separate datasets. Specifically, we partition the set of bags $S$ into three disjoint subsets, $S_1, S_2,$ and $S_3$ of equal size:

$$S = \{\underbrace{z_1, \ldots, z_m}_{S_1}, \underbrace{z_{m+1}, \ldots, z_{2m}}_{S_2}, \underbrace{z_{2m+1}, \ldots, z_{3m}}_{S_3}\}. \quad (4)$$

Subset $S_1$ will be used to estimate $\mathbb{E}_x[\Delta f_1(h_1, h_2, x)]$ and $\mathbb{E}_x[\Delta f_2(h_1, h_2; x)]$, subset $S_2$ will be used to estimate $p$, and subset $S_3$ will be the one out of which the LLP estimator (3) will be constructed. More specifically, we replace in (3) the average $p$ by $\widehat{p}_{S_2} = \frac{1}{m} \sum_{j=m+1}^{2m} \alpha_j$, that is, a simple empirical average of the label proportions observed on $S_2$. The average $\mathbb{E}_x[\Delta f_1(h_1, h_2, x)]$ will be estimated as

$$\widehat{\mathbb{E}}_{S_1}[\Delta f_1(h_1, h_2)]$$
$$= \widehat{\mu}_{\text{MoM}}\left( \{v_{1,1}, \ldots, v_{1,k}, \ldots, v_{m,1}, \ldots, v_{m,k}\} \right),$$

where, for each $j \in [m]$ and $i \in [k]$, $v_{j,i} = \Delta f_1(h_1, h_2, x_{j,i})$, with $x_{j,i}$ being the $i$-th feature vector in the $j$-th bag in $S_1$ (recall that all $x_{j,i}$ are available in the clear). Likewise, $\mathbb{E}_x[\Delta f_2(h_1, h_2; x)]$ is estimated through $\widehat{\mathbb{E}}_{S_1}[\Delta f_2(h_1, h_2)]$ where, this time, $v_{j,i} =$

---

[1]Other estimators working with unbounded random variables could be used instead, see, e.g., the survey (Lugosi & Mendelson, 2019). Moreover, if we restricted to bounded losses, then a standard empirical average would suffice here.

[2]If the median is not uniquely defined (which may happen when $r$ is an even number) we take the average of the two middle points. So, for instance median$\{1, 2, 3\} = 2$, but median$\{1, 2, 3, 4\} = 2.5$. Note that this always implies $\widehat{\mu}_{\text{MoM}}(-S) = -\widehat{\mu}_{\text{MoM}}(S)$. E.g., median$\{-1, -2, -3\} = -2$, and median$\{-1, -2, -3, -4\} = -2.5$.

$\Delta f_2(h_1, h_2; x_{j,i})$. For each bag $z_j \in S_3$ and pair $h_1, h_2 \in \mathcal{H}$ then define

$$\widetilde{\Delta}\ell_b(h_1, h_2; z_j) \quad (5)$$
$$= \widehat{\mathbb{E}}_{S_1}[\Delta f_1(h_1, h_2)] + \widehat{p}_{S_2} \, \widehat{\mathbb{E}}_{S_1}[\Delta f_2(h_1, h_2)]$$
$$+ \left( \alpha_j - \widehat{p}_{S_2} \right) \left( \sum_{i=1}^{k} \Delta f_2(h_1, h_2; x_{j,i}) \right.$$
$$\left. - k\,\widehat{\mathbb{E}}_{S_1}[\Delta f_2(h_1, h_2)] \right).$$

Finally, we define a MoM estimator based on the bags of $S_3$. Specifically, consider the $m$ bags $z_j$ of $S_3 = \{z_{2m+1}, \ldots, z_{3m}\}$, and set

$$Q(h_1, h_2; S)$$
$$= \text{MoM}\left( \widetilde{\Delta}\ell_b(h_1, h_2; z_{2m+1}), \ldots, \widetilde{\Delta}\ell_b(h_1, h_2; z_{3m}) \right).$$

Note that, by the way we defined the median, $Q(h_1, h_2; S) = -Q(h_2, h_1; S)$.

Algorithm 1, based on the Median-of-Means tournament approach (Lugosi & Mendelson, 2016) (see also Chapter 6 in (Devroye & Lugosi, 2001), and (Daskalakis et al., 2012)), iteratively eliminates losing functions from pairs in $\mathcal{H}$ when confidence is sufficient ($|Q(h_1, h_2; S)| > \beta/2$), outputting any remaining function in the pool. Theorem 3.2 (proof in Appendix B.2) provides its regret guarantee under a bounded second moment assumption.

---

**Algorithm 1** MoM tournament algorithm.

**input** : Dataset $S$ made up of $3m$ bags, each of size $k \geq 1$, desired regret bound $\beta > 0$

- Set $\mathcal{P} = \mathcal{H}$
- **for all** $h_1, h_2 \in \mathcal{H}, h_1 \neq h_2$ **do**
  - **If** $Q(h_1, h_2; S) > \beta/2$ **then** $\mathcal{P} \to \mathcal{P} \setminus \{h_1\}$
  - **If** $Q(h_1, h_2; S) < -\beta/2$ **then** $\mathcal{P} \to \mathcal{P} \setminus \{h_2\}$

**output** : $\widehat{h}$ is any $h \in \mathcal{P}$

---

**Theorem 3.2.** *Let $S = (x_1, y_1), \ldots, (x_n, y_n)$ be drawn i.i.d. from a distribution $\mathcal{D}$ over $\mathcal{X} \times \{0, 1\}$. Let the loss $\ell(\cdot, y) = f_1(\cdot) + y f_2(\cdot)$ be such that the maximal second moments $\max_{h \in \mathcal{H}} \mathbb{E}_x\left[ \left( f_1(h(x)) \right)^2 \right]$ and $\max_{h \in \mathcal{H}} \mathbb{E}_x\left[ \left( f_2(h(x)) \right)^2 \right]$ are finite. Let $\widehat{h}^\star = \min_{h \in \mathcal{H}} \mathcal{L}(h)$ be the best-in-class hypothesis. If $S$ is split into $3m$ random bags of size $k$, with $n = 3m \times k$ as in (4), then for all $\beta > 0$, the hypothesis $\widehat{h}$ output by Algorithm 1 satisfies $\text{Reg}(\widehat{h}) \leq \beta$ with probability*

*at least $1 - \delta$, provided*

$$n = O\left( \frac{k \, \sigma_{f_2}^2(\mathcal{H}) \, \log(|\mathcal{H}|/\delta)}{\beta^2} \right.$$

$$+ \frac{\sigma_{f_1}^2(\mathcal{H}) \, \log(|\mathcal{H}|/\delta) + \mathbb{E}^2[\mathcal{H}] \, \log(1/\delta)}{\beta^2}$$

$$\left. + k \, \log(|\mathcal{H}|/\delta) \right),$$

*where the big-oh only hides absolute constants, and*

$$\sigma_{f_i}^2(\mathcal{H}) = \max_{h \in \mathcal{H}} \mathrm{Var}_x \Big( f_i(h(x)) - f_i(\widehat{h}^\star(x)) \Big), \ i = 1, 2$$

$$\mathbb{E}^2[\mathcal{H}] = \max_{h \in \mathcal{H}} \mathbb{E}_x^2 \Big[ f_2(h(x)) - f_2(\widehat{h}^\star(x)) \Big] .$$

For completeness, we provide in Appendix B.3 (Theorem B.2) a more general statement that applies to bags of different sizes $k_1, k_2, \ldots$, where $k$ in Theorem 3.2 is replaced by the average bag size $\widehat{k}$. Similar generalization can be proven for the subsequent Theorem 3.5, as well as for the algorithms operating in the multiclass settings.

Theorem 3.2 can be extended to infinite $\mathcal{H}$ using Høgsgaard & Paudice (2025)'s machinery. The tournament in Algorithm 1 is replaced by a simpler Empirical Risk Minimization, using MoM estimators instead of standard averages.

*Example* 3.3. We now apply Theorem 3.2 to standard losses, like square loss and log loss, viewed here as canonical representatives of bounded (square) and unbounded (log) loss functions. The square loss $\ell(h(x), y) = (h(x) - y)^2$ is obtained by $f_1(h) = h^2$ and $f_2(h) = 1 - 2h$, with $h \in [0, 1]$. Both $f_1(\cdot)$ and $f_2(\cdot)$ are bounded; it is immediate to see that $\sigma_{f_1}^2(\mathcal{H}) \leq 1$, $\sigma_{f_2}^2(\mathcal{H}) \leq 4$, and $\mathbb{E}^2[\mathcal{H}] \leq 4$.

The log loss $\ell(h(x), y) = -y \log h(x) - (1 - y) \log (1 - h(x))$ is obtained by $f_1(h) = \log \frac{1}{1-h}$, and $f_2(h) = \log \frac{1-h}{h}$, for $h \in [0, 1]$. Neither $f_1(\cdot)$ nor $f_2(\cdot)$ is bounded. Yet, in order to guarantee the finiteness of the variance parameters in Theorem 3.2, it suffices to assume the finiteness for all $h \in \mathcal{H}$ of the second moments $\mathbb{E}_x \left[ \log^2 \frac{1}{h(x)} \right]$ and $\mathbb{E}_x \left[ \log^2 \frac{1}{1-h(x)} \right]$. A paradigmatic example where this is the case is when the models $h(x)$ are represented as sigmoidal functions $h(x) = \sigma(w_h(x))$, where $\sigma(a) = \frac{e^a}{1+e^a}$, $a \in \mathbb{R}$, and $w_h : \mathcal{X} \to \mathbb{R}$ is the so-called *logit* of model $h$. With this representation, one can see that the above finiteness conditions is implied by the finiteness of $\mathbb{E}_x[(w_h(x))^2]$. These are rather folklore facts. For completeness we give the details of these calculations in Appendix B.4.

The sample complexity from Theorem 3.2 is of the form $n = \widetilde{O}(k/\beta^2)$. The bound applies to the general non-realizable setting, and constitutes a so-called *slow* rate, as

the dependence on the regret $\beta$ is inverse *quadratic*. A sample complexity guarantee with a similar scope as ours is the one from (Busa-Fekete et al., 2023), whose bound is of the form $n = \widetilde{O}(k^2/\beta^2)$, that is, a slow rate with, in addition, a quadratic dependence on $k$, instead of linear (the authors work under general, but bounded, loss functions).

The more recent papers (Li et al., 2024; Busa-Fekete et al., 2025) do contain fast rates for realizable settings, but they only apply to the square loss case. In particular, the guarantee in (Li et al., 2024) is of the form $n = \widetilde{O}(k^3/\beta)$, but it holds only in the more restricted realizability condition where $\mathcal{L}(h^\star) = 0$, and the models in $\mathcal{H}$ have binary output, hence disallowing noise in the labels. The realizable guarantee in (Busa-Fekete et al., 2025), on the other hand, is a fast rate of the form $n = \widetilde{O}(k/\beta)$ which, as we already recalled, is best possible (up to log factors).

We next show that a similar fast rate can be achieved by Algorithm 1 on a wider family of loss functions that are more relevant in practice than square loss.

Given loss function $\ell(a, y) = f_1(a) + y f_2(a)$, with $a \in [0, 1]$ and $y \in \{0, 1\}$, consider its extension $\bar{\ell} : [0, 1] \times [0, 1] \to \mathbb{R}$, defined as $\bar{\ell}(a, b) = f_1(a) + b f_2(a)$, where we have simply turned the range of the second argument from $\{0, 1\}$ to $[0, 1]$. We say that $\ell$ is quadratically sandwiched w.r.t. $f_1$ and $f_2$ if there exist constants $C_1 > c_1 > 0$, and $C_2 > c_2 > 0$ such that, for all $a, b \in [0, 1]$, and $i = 1, 2$,

$$\frac{C_i}{2} \Big( f_i(a) - f_i(b) \Big)^2 \geq \bar{\ell}(a, b) - \bar{\ell}(b, b) \geq \frac{c_i}{2} \Big( f_i(a) - f_i(b) \Big)^2 . \tag{6}$$

As explained below, in order for fast rates in realizable settings to hold, only the constants $c_1$ and $c_2$ that appear in the lower bound will matter.

*Example* 3.4. We illustrate (6) through some examples. For square loss, recalling Example 3.3, condition (6) is clearly verified for $f_2$, since $(f_2(a) - f_2(b))^2 = 4(a - b)^2 = 4(\bar{\ell}(a, b) - \bar{\ell}(b, b))$, resulting in $C_2 = c_2 = 2$. As for $f_1$, we have $c_1 = 1/2$, but in order to get a finite $C_1$, we have to require that either $a$ or $b$ is bounded away from 0. E.g., if $a + b \geq \epsilon > 0$ then $C_1 = \frac{2}{\epsilon^2}$. Yet, the constant $C_1$ will be immaterial in realizable settings – see below.

For log loss, one can easily see that $\bar{\ell}(a, b) - \bar{\ell}(b, b) = b \log \frac{b}{a} + (1 - b) \log \frac{1-b}{1-a} = \mathrm{KL}(b, a)$, the KL-divergence between two Bernoulli random variables with biases $b$ and $a$. As for condition (6), consider again the sigmoidal representation in Example 3.3. This condition boils down to the strong convexity ($c_2$ side) and strong smoothness ($C_2$ side) of the function $w_h \to \log(1 + e^{w_h})$. This function is known to be strongly smooth, but the strong convexity requires the logits $w_h$ to be bounded. Boundedness is also needed in order to enforce (6) on $C_1$ and $c_1$. Appendix B.5 contains detailed calculations for these conditions.

The following theorem specifies the properties of Algorithm 1 for quadratically sandwiched losses. The proof is in Appendix B.6.

**Theorem 3.5.** *Under the same assumptions and notation as in Theorem 3.2, let the loss $\ell$ satisfy the sandwich condition (6). Then for all $\beta > 0$, the hypothesis $\widehat{h}$ output by Algorithm 1 satisfies $Reg(\widehat{h}) \leq \beta$ with probability at least $1 - \delta$, provided*

$$
n = O\Bigg(\Bigg(\frac{\frac{k}{c_2}\beta + k\left(\frac{C_2}{c_2}+1\right)\gamma_{f_2}(\widehat{h}^\star, h^\star)}{\beta^2}
$$
$$
+ \frac{\frac{1}{c_1}\beta + \left(\frac{C_1}{c_1}+1\right)\gamma_{f_1}(\widehat{h}^\star, h^\star)}{\beta^2} + k\Bigg)\log\frac{|\mathcal{H}|}{\delta}\Bigg) ,
$$

*where $\gamma_{f_i}(h, \widehat{h}^\star) = \mathbb{E}_x\left[(f_i(h(x)) - f_i(\widehat{h}^\star(x)))^2\right]$, $i = 1, 2$, and the big-oh notation only hides absolute constants. In the realizable case $(\widehat{h}^\star = h^\star)$ the bound reduces to*

$$
n = O\left(\left(\left(\frac{k}{c_2} + \frac{1}{c_1}\right)\frac{1}{\beta} + k\right)\log\frac{|\mathcal{H}|}{\delta}\right) .
$$

Thus in the realizable case the bound has the form $n = \widetilde{O}(k/\beta)$ we expect for fast rates of convergence. We have a specific dependence on the curvature properties of the loss function, as quantified by (6) via the lower bound constants $c_1$ and $c_2$. Note that the upper constants $C_1$ and $C_2$ do not play a role in the realizable bound. More lax conditions can be formulated in order to achieve fast rates, as available in the standard statistical learning literature. We could adapt such conditions to LLP as well. The reader is referred, e.g., to (Grunwald & Mehta, 2020), and references therein.

### 3.2. The Multi-Class Case (Sketch)

Any loss function $\ell(h(x), y)$ for multi-class labels $y \in \{0, \ldots, c-1\}$ can always be written as $\ell(h(x), y) = \sum_{r=0}^{c-1}\{y = r\}\,\ell(h(x), r)$. The bag-level estimator for the *full histogram* multi-class scenario is a generalization of the one for binary labels, where the role of the two functions $f_1(h(x))$ and $f_2(h(x))$ is now played by the $c$ loss components $\ell(h(x), r)$, $r = 0, \ldots, c-1$. The LLP algorithm is the very same as Algorithm 1, but applied to the alluded bag-level estimator for full histogram multi-class—see Appendix C for details.

On the other hand, we are able to handle the *total* multi-class case only when $\ell$ has the affine form $\ell(h(x), y) = f_1(h(x)) + y f_2(h(x))$, where now, $y \in \{0, \ldots, c-1\}$. The loss admits this representation if it can be interpreted as the negative log-likelihood of a distribution in the (single-parameter canonical form) exponential family for which $y$ is a sufficient statistic. Notable examples are the Poisson Log Loss $\ell(h(x), y) = h(x) - y \log h(x)$ (negative

log-likelihood of Poisson distribution) and the square loss $\ell(h(x), y) = (y - h(x))^2$ (negative log-likelihood of a Gaussian distribution with unit variance; note that the $y^2$ term can be disregarded here, as it is independent of $h(x)$). The multi-class log loss/cross-entropy does not admit this representation.

From Appendix C, one can see that the regret guarantee we obtain for our algorithm in the full histogram case (Theorem C.2) is of the form

$$
n = O\Big(\frac{k\,\Delta\ell^2(\mathcal{H})\,\log(|\mathcal{H}|/\delta)}{\beta^2}\Big) ,
$$

where $\Delta\ell^2(\mathcal{H}) = \max_{h \in \mathcal{H}}\mathbb{E}_x\big[\max_r\big(\ell(h(x), r)\big)^2\big]$ is the factor hiding the dependence on the number of classes $c$. This dependence can be removed completely if the loss is bounded, and turned into a mild $\log c$ when the loss is unbounded but has light tails. On the contrary, the bounds for the total multi-class case mimic rather closely those for binary labels (Theorem 3.2 and Theorem 3.5) but with an extra $c^2$ factor, due to the inflated variance terms. Details are again contained in Appendix C.

For practical reasons, in the experiments that follow we depart from the MoM tournament scheme, and rather test our bag-level estimators combined with SGD-like algorithms operating with log loss.

## 4. Experiments

We compare our general unbiased proportion matching loss (2) against standard baselines, where the goal is to minimize the log loss, which is more aligned with our goal of coping with conversion prediction problems in ad domains. We evaluate against EasyLLP (Busa-Fekete et al., 2023) and a standard proportion matching method (e.g., (Yu et al., 2013)), which is shown to be a strong baseline in (Busa-Fekete et al., 2023). Dataset preparation and experimental setup is heavily inspired by (Busa-Fekete et al., 2023). We conduct two kinds of experiments: (i) batch experiments, making many training passes over a dataset aimed at achieving low instance-level log loss on held-out test data, and (ii) an online experiment on the Criteo Display Advertising Challenge dataset (Tien et al., 2014), where the goal is to make accurate predictions on a stream of data as it arrives.

**Datasets and Models.** We conduct batch learning experiments on the MNIST (LeCun et al., 2010), CIFAR-10 (Krizhevsky, 2009), Higgs (Baldi et al., 2014), and Adult (Kohavi & Becker, 1996) datasets. For MNIST the training labels are binarized based on whether each digit is even or odd, while for CIFAR-10 we consider two binarization strategies: Animal-vs-Machine and Cat-vs-Rest. We train convolutional neural networks for MNIST and CIFAR-10 and fully-connected networks for Higgs and Adult. Our

online experiment is conducted on the Criteo Display Advertising Challenge (Tien et al., 2014), which is a click prediction task where each training example contains information about an ad impression and the label corresponds to whether the ad was clicked or not. For this dataset, we use a deep embedding network, where each feature value is associated with a learned embedding vector and the model's prediction is the output of several fully-connected layers applied to the concatenated feature embeddings. Further details are given in Appendix D.

**LLP Losses.** GENERALUPM is our unbiased proportion matching loss (2), EASYLLP is the LLP loss of Busa-Fekete et al. (2023), and PM is the standard proportion matching method. In the batch setup, we instantiate each LLP loss with binary cross-entropy with smoothed labels; in the online setup no label smoothing is applied. Note that label smoothing only changes the loss definition, the event-level labels are not modified. Details of the losses and implementation of GENERALUPM and EASYLLP are in Appendix D.

**Batch Experimental Setup.** LLP training data is constructed by shuffling and grouping consecutive examples into non-overlapping bags of the desired size. If the number of training examples $n$ is not divisible by $k$, the leftover examples are discarded. We train models on each dataset using the Adam optimizer (Kingma & Ba, 2015) to minimize each LLP loss. We train for $E$ epochs (passes through the training data) where on each epoch the order of the LLP bags is shuffled. Note that we do *not* change the examples present in each bag per epoch, since that would gradually leak additional label information. We use bag sizes $k = 2^i$ for $i = 0, \ldots, 11$ and batches that contain $2^{12} = 4096$ training examples regardless of $k$. Learning rates are tuned from $10^{-7}$ to $10^{-1}$ (16 log-spaced values); the label marginal $p$ is estimated as the average label proportion over the entire training dataset. For each dataset, LLP loss, bag size, and learning rate, experiments are repeated 10 times with randomized data shuffling and model initialization. After each epoch, the average instance-level log loss is measured on test data. The lowest average test log loss achieved is then reported for each dataset, LLP loss, and bag size.

**Online Experimental Setup.** This is similar to the batch setup with the following modifications. We read the data in consecutive chunks of $2^{16}$ examples. On each chunk, the model is evaluated before the parameters are updated. The model updates are performed similarly to the batch setting: the chunk examples are shuffled and partitioned into consecutive bags. This simulates ad serving where predictions precede aggregate label updates of the model weights. The online experiment is repeated 5 times with different model initialization and data shuffling seeds. For each bag size, we tune the learning rate over the same grid to optimize the average log loss over all chunks. Finally, the

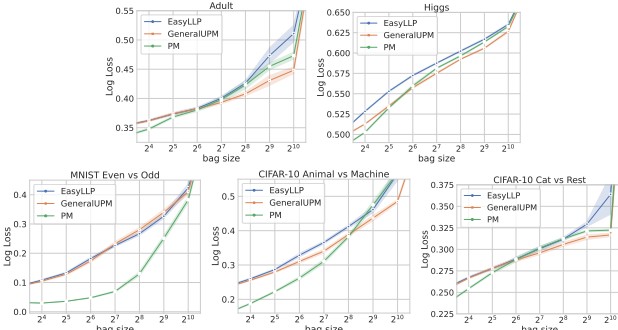

*Figure 1.* Minimum average test log loss achieved for each $k$, optimized across various learning rates and stopping epochs. Results are averaged over 10 repetitions, with error bars representing one standard error.

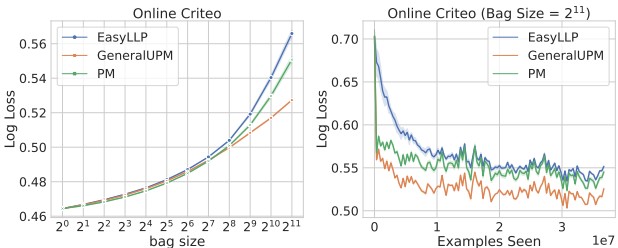

*Figure 2.* **Left:** Average chunk log loss for each LLP loss and bag size. **Right:** the per-chunk log losses during online training for bag size $2^{11}$. Error bars indicate one standard error in the mean across 5 repetitions.

label marginal $p$ is estimated for each chunk as the average label proportion for bags in that chunk. This is required to track the time-varying click rate in the Criteo dataset.

**Results.** Fig. 1 shows the test log loss achieved by each LLP loss in the batch experimental setup. On all batch datasets and LLP losses, the performance with bag size $2^{11}$ is trivial, so we omit this bag size. We also omit bag sizes smaller than $2^4$ in order to focus on the more challenging settings.

We note that for smaller $k$, PM consistently achieves the lowest loss. We conjecture that this is due to the fact that in realizable scenarios PM operating with a proper loss function (the log loss is indeed proper) can be shown to approximate the Bayes optimal odds $\mathbb{P}(y = 1|x)$ (see, e.g., Theorem 3.2 and Corollary 3.3 in (Busa-Fekete et al., 2023)). This may explain, in particular, the very good performance of PM on MNIST Even-vs-Odd, given how easy this dataset is. As we depart from realizable scenarios, the situation becomes a bit murky. As $k$ increases, the bias of PM increases as well, causing a degradation of its relative performance against GENERALUPM (which remains unbiased in all cases). Except on MNIST Even-vs-Odd, we consistently see that for larger $k$ GENERALUPM performs better than the baselines. Figure 2 shows the average log loss on the Criteo dataset where we see the same trend: PM performs

well at small bag sizes $k$, but GENERALUPM is much better at larger $k$. We also include a plot of the per-chunk log loss for bag size $2^{11}$, where we see that GENERALUPM has better performance over the entire duration. See also Appendix D for corresponding results based on AUC and plots including all bag sizes.

## 5. Conclusions and Limitations

Our LLP method applies to virtually any loss function. As in past investigations, we are centered around designing (almost) unbiased estimates of the instance level population loss through bag-level information. However, unlike past investigations, we have been able to reduce the variance of our estimators to a quantity that is *independent* of $k$. On one hand, this directly translates to improved/optimal sample size results in both realizable and non-realizable settings, extending to a broader array of losses than recent square-loss studies. On the other hand, this gives our method a desired robustness to varying bag sizes, as the regret guarantees only depend on the number $m$ of available bags, irrespective of their sizes. Experiments on binary datasets demonstrate improved performance over baselines in large $k$ regimes, a crucial aspect for modern applications of LLP to online advertising.

From a theoretical standpoint, the tightness of our multi-class bounds regarding $c$ remains unknown. Additionally, we currently do not have a fast rate analysis for the full histogram multi-class case. Moreover, the necessity of the affine loss form we assumed for the total multi-class setting, which currently limits our technique's applicability, is unclear. Finally, the paper is limited in its experimental investigation, as it currently restricts to binary classification.

## Acknowledgements

We thank the anonymous ICML reviewers for their constructive comments and the Area Chair for their wise handling of this paper.

## Impact Statement

"This paper presents work whose goal is to advance the field of Machine Learning. There are many potential societal consequences of our work, none which we feel must be specifically highlighted here."

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

# A. Related Work

The problem of Learning with Label Proportions (LLP) has historically been motivated by real-world applications where labels are available not for each data point, but only in the form of aggregate labels for bags of data. This may occur due to privacy and legal restrictions (e.g., (Rueping, 2010; Wojtusiak et al., 2011)), cost of supervision (e.g., (Chen et al., 2004)), lack of resolution in labeling instruments (e.g., (Dery et al., 2018)), etc.

LLP garnered initial attention in the research community with early contributions from (de Freitas & Kuck, 2005; Musicant et al., 2007; Quadrianto et al., 2008; Rueping, 2010; Stolpe & Morik, 2011; Yu et al., 2013; Patrini et al., 2014). In their work, de Freitas & Kuck (2005) introduced a hierarchical model that generates labels according to the provided label proportions, and proposed an inference method based on Markov Chain Monte Carlo which, however, did not scale effectively to larger datasets. Musicant et al. (2007) emphasized the possibility of adapting conventional supervised learning techniques, such as Support Vector Machines (SVM) and $k$-Nearest Neighbors, to the LLP setting by reformulating their objective functions. However, their paper did not include any experimental results on classification problems. Quadrianto et al. (2008); Patrini et al. (2014) developed algorithms for learning an exponential generative model. Nevertheless, their approaches relied on significant modeling assumptions, notably conditional exponential models, which are often inadequate for contemporary machine learning architectures, particularly Deep Neural Networks (DNNs). Rueping (2010) and Yu et al. (2013) both explored the adaptation of SVMs for LLP. Notably, Yu et al. (2013) presented an SVM regression algorithm that optimizes the SVM loss by aligning with the given label proportions. This general strategy is referred to as "proportion matching" in (Busa-Fekete et al., 2023; 2025). However, their method was constrained to linear models within a specific feature space. Similar limitations were present in other SVM-based studies on LLP, such as (Qi et al., 2017; Shi et al., 2019).

The natural appeal of the proportion matching concept led to its extension to other types of classifiers. For instance, Li & Taylor (2015) applied this idea to Convolutional Neural Networks using a generative model where the maximum likelihood estimator was computed via the Expectation Maximization algorithm. Regrettably, this approach proved to be computationally prohibitive even for moderately sized DNN architectures. Empirical approaches based on DNNs for large-scale and multi-class data (Kotzias et al., 2015; Liu et al., 2019; Dulac-Arnold et al., 2019; Nandy et al., 2022) have also been developed. Other approaches similar in spirit to proportion matching are (Ardehaly and Culotta, 2017) and (Havaldar et al., 2024). The paper by Ardehaly and Culotta (2017) is closer in spirit to our Proportion Matching (PM) in using KL-vicinity between average model prediction and average label within bags. (Havaldar et al., 2024) is another experimental paper that uses a Gibbs distribution to softly enforce a similar (but square loss-based) vicinity constraint. We believe these approaches are orthogonal to ours, in that we think our loss design can be combined with these approaches.

Delving into the more theoretically-grounded explorations of LLP, a notable body of work includes (Saket, 2021; 2022; Brahmbhatt et al., 2023; Lu et al., 2019; 2021; Scott & Zhang, 2020; Zhang et al., 2022), and the already cited (Busa-Fekete et al., 2023; Li et al., 2024; Busa-Fekete et al., 2025).

The theoretical framework presented by Saket (2021; 2022); Brahmbhatt et al. (2023) is predominantly confined to linear-threshold functions, with some investigations further taking advantage of overlapping bags. In stark contrast, our approach aligns with (Busa-Fekete et al., 2023; Li et al., 2024; Busa-Fekete et al., 2025), where we operate within the more conventional setting of non-overlapping and i.i.d. bags, but cope with a broader and more versatile spectrum of model classes.

Adjacent to LLP is the problem of *learning from multiple unlabeled datasets* (Lu et al., 2019; 2021). While the authors put forth a debiasing method for the underlying loss function and demonstrate consistency (akin to our results and those in (Busa-Fekete et al., 2023; Li et al., 2024; Busa-Fekete et al., 2025)), their method relies on restrictive assumptions, such as the separation of class priors over bags. This necessitates access to class-conditional distributions. In contrast, our setup involves bags drawn i.i.d. from the same prior, a scenario where the algorithms of Lu et al. (2019; 2021) would be ineffective.

Scott & Zhang (2020); Zhang et al. (2022) offer an alternative perspective on LLP, involving the reduction of the LLP problem to the problem of learning with label noise. In this framework, bags are paired, and each pair is analyzed as a noisy label scenario, with label proportions interpreted as label flipping probabilities. Nevertheless, this bag pairing, much like in (Lu et al., 2019), fundamentally depends on the statistical variations between bags, a property that we deliberately exclude in our work. Additionally, the balanced risk they focus on as their population loss differs from the population losses we are concerned with in this study.

Franco et al. (2023) point out that many nuances exist in LLP formulations that require careful consideration of the

dependence structure involving bags, feature items, and labels. Among other things, the paper proposes a taxonomy of LLP tasks based on such dependencies.

## B. Technical Appendices and Supplementary Material for the Binary Case

This appendix contains the proofs omitted from the main text for the binary label case.

### B.1. Proof of Lemma 3.1

*Proof.* Recall that $\eta(x) = \mathbb{P}_{\mathcal{D}_{\mathcal{Y}|x}}(y = 1|x) = \mathbb{E}_{y \sim \mathcal{D}_{\mathcal{Y}|x}}[y|x]$.

We have

$$
\begin{aligned}
\mathbb{E}_{(x,y)}[\ell(h(x), y)] &= \mathbb{E}_x\left[\mathbb{E}_{y|x}[\ell(h(x), y) \mid x]\right] \\
&= \mathbb{E}_x\left[\mathbb{E}_{y|x}[f_1(h(x)) + yf_2(h(x)) \mid x]\right] \\
&= \mathbb{E}_x\left[f_1(h(x)) + \eta(x)f_2(h(x))\right] \\
&= \mathbb{E}[f_1(h(x))] + \mathbb{E}[\eta(x)f_2(h(x))] .
\end{aligned}
\tag{7}
$$

Define the intermediate bag-level quantity

$$
g(h, z) = \mathbb{E}\left[f_1(h(x))\right] + \left(\frac{1}{k}\sum_{i=1}^{k}\widetilde{y}_i\right)\left(\frac{1}{k}\sum_{i=1}^{k}\widetilde{f}_{2,i}\right),
$$

with $\widetilde{y}_i = y_i - p$, and $\widetilde{f}_{2,i} = f_2(h(x_i)) - \mathbb{E}[f_2(h(x))]$. The function $g(h, z)$ is shaped so as to match (7) as much as possible.

Then

$$
\begin{aligned}
\mathbb{E}[g(h, z)] &= \mathbb{E}[f_1(h(x))] + \frac{1}{k^2}\mathbb{E}_z\left[\sum_{i=1}^{k}\widetilde{y}_i\widetilde{f}_{2,i} + \sum_{i\neq j}\widetilde{y}_i\widetilde{f}_{2,j}\right] \\
&= \mathbb{E}[f_1(h(x))] + \frac{1}{k^2}\mathbb{E}_z\left[\sum_{i=1}^{k}\widetilde{y}_i\widetilde{f}_{2,i}\right] + \frac{1}{k^2}\sum_{i\neq j}\mathbb{E}_z\left[\widetilde{y}_i\right]\mathbb{E}_z\left[\widetilde{f}_{2,j}\right] \\
&= \mathbb{E}[f_1(h(x))] + \frac{1}{k^2}\mathbb{E}_z\left[\sum_{i=1}^{k}\widetilde{y}_i\widetilde{f}_{2,i}\right] \\
&\quad (\text{as } \mathbb{E}_z\left[\widetilde{y}_i\right] = \mathbb{E}_z\left[\widetilde{f}_{2,j}\right] = 0) \\
&= \mathbb{E}[f_1(h(x))] + \frac{1}{k}\mathbb{E}_z\left[\widetilde{y}_1\widetilde{f}_{2,1}\right] \\
&= \mathbb{E}[f_1(h(x))] + \frac{1}{k}\mathbb{E}_{(x,y)}\left[(y - p)(f_2(h(x)) - \mathbb{E}[f_2(h(x))])\right] .
\end{aligned}
$$

But

$$
\begin{aligned}
\mathbb{E}_{(x,y)}\left[(y - p)(f_2(h(x)) - \mathbb{E}[f_2(h(x))])\right] &= \mathbb{E}_{(x,y)}\left[(y - p)f_2(h(x))\right] - \mathbb{E}_{(x,y)}\left[(y - p)\right]\mathbb{E}[f_2(h(x))] \\
&= \mathbb{E}_{(x,y)}\left[(y - p)f_2(h(x))\right] \\
&= \mathbb{E}_{(x,y)}\left[yf_2(h(x))\right] - p\mathbb{E}_{(x,y)}\left[f_2(h(x))\right] \\
&= \mathbb{E}_x\left[\eta(x)f_2(h(x))\right] - p\mathbb{E}_x\left[f_2(h(x))\right]
\end{aligned}
$$

where the last equality follows from an argument similar to the derivation in Equation (7). Combining the last two derivations we get that

$$
\mathbb{E}[g(h, z)] = \mathbb{E}[f_1(h(x))] + \frac{1}{k}\mathbb{E}_x\left[\eta(x)f_2(h(x))\right] - \frac{p}{k}\mathbb{E}_x\left[f_2(h(x))\right] .
$$

Thus, comparing (7) to the above expectation, we see that

$$\mathbb{E}_{(x,y)}[\ell(h(x), y)] = k \, \mathbb{E}_z[g(h, z)] + p \, \mathbb{E}_x[f_2(h(x))] - (k-1)\mathbb{E}_x[f_1(h(x))] \,,$$

making

$$\ell_b(h, z) = k \, g(h, z) + p \, \mathbb{E}_x[f_2(h(x))] - (k-1)\mathbb{E}_x[f_1(h(x))]$$

an unbiased estimator of the sample-level loss $\mathbb{E}_{(x,y)}[\ell(h(x), y)]$.

Substituting back the definition of the bag-level quantity $g(h, z)$ we get that the estimator is

$$\ell_b(h, z) = \mathbb{E}_x[f_1(h(x)) + p \, f_2(h(x))] + (\alpha - p) \left( \sum_{i=1}^{k} f_2(h(x_i)) - k \, \mathbb{E}[f_2(h(x))] \right). \tag{8}$$

This proves the first part of the lemma.

As for the variance, we introduce the short-hand notation

$$\widetilde{\alpha} = \alpha - p \,, \qquad \widetilde{f_2} = \frac{1}{k} \sum_{i=1}^{k} f_2(h(x_i)) - \mathbb{E}[f_2(h(x))] \,,$$

and let $z = (\mathcal{B}, \alpha)$, with $\mathcal{B} = (x_1, \ldots, x_k)$. We can write using the total variance formula

$$\begin{aligned}
\frac{1}{k^2} \, \mathrm{Var}_z(\ell_b(h, z)) &= \mathrm{Var}_z\left(\widetilde{\alpha} \, \widetilde{f_2}\right) \\
&= \mathbb{E}_{\mathcal{B}}\left[ \mathrm{Var}_\alpha(\widetilde{\alpha}_j \, \widetilde{f_2} \mid \mathcal{B}) \right] + \mathrm{Var}_{\mathcal{B}}\left( \mathbb{E}_\alpha\left[ \widetilde{\alpha} \, \widetilde{f_2} \mid \mathcal{B} \right] \right) \\
&= \mathbb{E}_{\mathcal{B}}\left[ (\widetilde{f_2})^2 \, \mathrm{Var}_\alpha(\widetilde{\alpha} \mid \mathcal{B}) \right] + \mathrm{Var}_{\mathcal{B}}\left( \widetilde{f_2} \, \mathbb{E}_\alpha\left[ \widetilde{\alpha} \mid \mathcal{B} \right] \right) \\
&= \mathbb{E}_{\mathcal{B}}\left[ (\widetilde{f_2})^2 \, \frac{\sum_{i=1}^{k} \eta(x_i)(1 - \eta(x_i))}{k^2} \right] + \mathrm{Var}_{\mathcal{B}}\left( \widetilde{f_2} \left( \frac{1}{k} \sum_{i=1}^{k} \eta(x_i) - p \right) \right) \\
&\leq \frac{1}{4k} \mathbb{E}_{\mathcal{B}}\left[ (\widetilde{f_2})^2 \right] + \mathrm{Var}_{\mathcal{B}}\left( \widetilde{f_2} \left( \frac{1}{k} \sum_{i=1}^{k} \eta(x_i) - p \right) \right) \\
&= \frac{1}{4k} \, \mathrm{Var}_{\mathcal{B}}\left[ \frac{1}{k} \sum_{i=1}^{k} f_2(h(x_i)) \right] + \mathrm{Var}_{\mathcal{B}}\left( \widetilde{f_2} \left( \frac{1}{k} \sum_{i=1}^{k} \eta(x_i) - p \right) \right) \\
&= \frac{1}{4k^2} \, \mathrm{Var}_x(f_2(h(x))) + \mathrm{Var}_{\mathcal{B}}\left( \widetilde{f_2} \left( \frac{1}{k} \sum_{i=1}^{k} \eta(x_i) - p \right) \right). \tag{9}
\end{aligned}$$

The term

$$\mathrm{Var}_{\mathcal{B}}\left( \widetilde{f_2} \left( \frac{1}{k} \sum_{i=1}^{k} \eta(x_i) - p \right) \right)$$

can be treated as follows. Define

$$\widetilde{\eta}_i = \eta(x_i) - p$$

and, as before,

$$\widetilde{f}_{2,i} = f_2(h(x_i)) - \mathbb{E}[f_2(h(x))] \,.$$

We have

$$\begin{aligned}
\mathrm{Var}_{\mathcal{B}}\left( \widetilde{f_2} \left( \frac{1}{k} \sum_{i=1}^{k} \eta(x_i) - p \right) \right) &= \mathrm{Var}_{\mathcal{B}}\left( \left( \frac{1}{k} \sum_{i=1}^{k} \widetilde{f}_{2,i} \right) \left( \frac{1}{k} \sum_{i=1}^{k} \widetilde{\eta}_i \right) \right) \\
&= \frac{1}{k^4} \, \mathrm{Var}_{\mathcal{B}}\left( \left( \sum_{i=1}^{k} \widetilde{f}_{2,i} \right) \left( \sum_{i=1}^{k} \widetilde{\eta}_i \right) \right) \\
&\leq \frac{1}{k^4} \mathbb{E}_{\mathcal{B}}\left[ \left( \sum_{i=1}^{k} \widetilde{f}_{2,i} \right)^2 \left( \sum_{i=1}^{k} \widetilde{\eta}_i \right)^2 \right].
\end{aligned}$$

Now, the square within the above expectation can also be rewritten as

$$\left(\sum_{i=1}^{k} \widetilde{f}_{2,i}\right)^2 \left(\sum_{i=1}^{k} \widetilde{\eta}_i\right)^2 = \left(\sum_{a=1}^{k}(\widetilde{f}_{2,a})^2 + \sum_{a\neq b}\widetilde{f}_{2,a}\widetilde{f}_{2,b}\right)\left(\sum_{c=1}^{k}(\widetilde{\eta}_c)^2 + \sum_{c\neq d}\widetilde{\eta}_c\widetilde{\eta}_d\right).$$

Since the variables $\widetilde{f}_{2,i}$ and $\widetilde{\eta}_j$ are zero mean, and independent of one another when $i \neq j$, the only terms that survive once we take the expectations are those of the form

$$(\widetilde{f}_{2,a})^2(\widetilde{\eta}_c)^2 \qquad \forall a, c \in [k]$$

which are exactly $k^2$, and those of the forms

$$\widetilde{f}_{2,a}\,\widetilde{f}_{2,b}\,\widetilde{\eta}_a\,\widetilde{\eta}_b\,, \qquad \widetilde{f}_{2,a}\,\widetilde{f}_{2,b}\,\widetilde{\eta}_b\,\widetilde{\eta}_a \qquad \forall a, b \in [k]\,, a \neq b$$

which are exactly $2k(k-1)$. Therefore

$$\mathbb{E}_{\mathcal{B}}\left[\left(\sum_{i=1}^{k}\widetilde{f}_{2,i}\right)^2\left(\sum_{i=1}^{k}\widetilde{\eta}_i\right)^2\right] = \mathbb{E}\left[\sum_{a,c}(\widetilde{f}_{2,a})^2(\widetilde{\eta}_c)^2 + 2\sum_{a\neq b}\widetilde{f}_{2,a}\,\widetilde{f}_{2,b}\,\widetilde{\eta}_a\,\widetilde{\eta}_b\right]$$

$$= \mathbb{E}\left[\sum_{a}(\widetilde{f}_{2,a})^2(\widetilde{\eta}_a)^2 + \sum_{a\neq c}(\widetilde{f}_{2,a})^2(\widetilde{\eta}_c)^2 + 2\sum_{a\neq b}\left(\widetilde{f}_{2,a}\,\widetilde{\eta}_a\right)\left(\widetilde{f}_{2,b}\,\widetilde{\eta}_b\right)\right]$$

$$= k\,\mathbb{E}[(\widetilde{f}_{2,1})^2(\widetilde{\eta}_1)^2] + k(k-1)\,\mathbb{E}[(\widetilde{f}_{2,1})^2]\,\mathbb{E}[(\widetilde{\eta}_1)^2] + 2k(k-1)\mathbb{E}^2[\widetilde{f}_{2,1}\,\widetilde{\eta}_1]$$

$$\leq k\,\mathbb{E}[(\widetilde{f}_{2,1})^2(\widetilde{\eta}_1)^2] + k(k-1)\,\mathbb{E}[(\widetilde{f}_{2,1})^2]\,\mathbb{E}[(\widetilde{\eta}_1)^2] + 2k(k-1)\mathbb{E}[(\widetilde{f}_{2,1})^2\,(\widetilde{\eta}_1)^2]$$

$$= (2k^2 - k)\,\mathbb{E}[(\widetilde{f}_{2,1})^2(\widetilde{\eta}_1)^2] + k(k-1)\,\mathbb{E}[(\widetilde{f}_{2,1})^2]\,\mathbb{E}[(\widetilde{\eta}_1)^2]\,.$$

But

$$\mathbb{E}[(\widetilde{f}_{2,1})^2(\widetilde{\eta}_1)^2] = \mathbb{E}_x\left[\left(f_2(h(x)) - \mathbb{E}[f_2(h(x))]\right)^2\left(\eta(x) - p\right)^2\right]$$

$$\leq \mathbb{E}_x\left[\left(f_2(h(x)) - \mathbb{E}[f_2(h(x))]\right)^2\right] = \mathrm{Var}_x(f_2(h(x)))\,,$$

$$\mathbb{E}[(\widetilde{f}_{2,1})^2] = \mathrm{Var}(\widetilde{f}_{2,1}) = \mathrm{Var}_x(f_2(h(x)))\,,$$

$$\mathbb{E}[(\widetilde{\eta}_1)^2] = \mathrm{Var}_x(\eta(x)) = \mathbb{E}_x[\eta^2(x)] - \mathbb{E}_x^2[\eta(x)] \leq p(1-p) \leq \frac{1}{4}\,.$$

Thus

$$\mathbb{E}_{\mathcal{B}}\left[\left(\sum_{i=1}^{k}\widetilde{f}_{2,i}\right)^2\left(\sum_{i=1}^{k}\widetilde{\eta}_i\right)^2\right] \leq (2k^2 - k)\,\mathrm{Var}_x(f_2(h(x))) + \frac{k(k-1)}{4}\,\mathrm{Var}_x(f_2(h(x)))$$

$$\leq \frac{9}{4}k^2\,\mathrm{Var}_x(f_2(h(x)))\,, \tag{10}$$

and

$$\mathrm{Var}_{\mathcal{B}}\left(\widetilde{f}_2\left(\frac{1}{k}\sum_{i=1}^{k}\eta(x_i) - p\right)\right) \leq \frac{9}{4k^2}\,\mathrm{Var}_x(f_2(h(x)))\,.$$

Substituting back into Equation (9) we get

$$\mathrm{Var}_z(\ell_b(h, z)) \leq \frac{1}{4}\,\mathrm{Var}_x(f_2(h(x))) + \frac{9}{4}\,\mathrm{Var}_x(f_2(h(x))) = \frac{5}{2}\,\mathrm{Var}_x(f_2(h(x)))\,,$$

as claimed. $\qquad\square$

## B.2. Proof of Theorem 3.2

Denote by $\mathcal{L}(h)$ the population loss of model $h$ when the underlying loss is $\ell$, that is

$$\mathcal{L}(h) = \mathbb{E}_{(x,y)}[\ell(h(x), y)] \,.$$

Also set, for each pair $h_1, h_2 \in \mathcal{H}$,

$$\Delta\mathcal{L}(h_1, h_2) = \mathcal{L}(h_1) - \mathcal{L}(h_2) \,.$$

Recall that $Q(h_1, h_2; S)$ is the MoM of the estimators $\widetilde{\Delta}\ell_b(h_1, h_2; z_j)$ per bag $z_j \in S_3$ (which use the estimates obtained using $S_1$ and $S_2$).

Then, e.g., (Lugosi & Mendelson, 2019) (Theorem 2 therein[3] ) shows that if $m \geq r = \lceil 8 \log(1/\delta) \rceil$, we have

$$\mathbb{P}\left( \forall h_1, h_2 \in \mathcal{H} \times \mathcal{H} \; \left| Q(h_1, h_2; S) - \mathbb{E}_z[\widetilde{\Delta}\ell_b(h_1, h_2; z) \,|\, S_1, S_2] \right| \right.$$
$$\left. \leq \sigma(h_1, h_2)\sqrt{\frac{32 \log(|\mathcal{H}|^2/\delta)}{m}} \;\middle|\; S_1, S_2 \right) \geq 1 - \delta \,, \tag{11}$$

where $\sigma^2(h_1, h_2) = \mathrm{Var}_z(\widetilde{\Delta}\ell_b(h_1, h_2; z) \,|\, S_1, S_2)$, and the probability above is taken over $S_3$, conditioned on $S_1$ and $S_2$.

We shall henceforth assume the stronger condition $m = \Omega(\log(|\mathcal{H}|/\delta))$.

We first need to relate $\mathbb{E}_z[\widetilde{\Delta}\ell_b(h_1, h_2; z)]$ to the desired $\Delta\mathcal{L}(h_1, h_2)$, and also bound $\mathrm{Var}_z(\widetilde{\Delta}\ell_b(h_1, h_2; z) \,|\, S_1, S_2)$.

From Theorem 2 in (Lugosi & Mendelson, 2019) one can also see that

$$\mathbb{P}\left( \forall h_1, h_2 \in \mathcal{H} \times \mathcal{H} \; \left| \widehat{\mathbb{E}}_{S_1}[\Delta f_1(h_1, h_2)] - \mathbb{E}_x[\Delta f_1(h_1, h_2, x)] \right| \right.$$
$$\left. \leq \sigma_{f_1}(h_1, h_2)\sqrt{\frac{32 \log(|\mathcal{H}|^2/\delta)}{mk}} \right) \geq 1 - \delta \tag{12}$$

with $\sigma_{f_1}^2(h_1, h_2) = \mathrm{Var}_x(\Delta f_1(h_1, h_2; x))$, and

$$\mathbb{P}\left( \forall h_1, h_2 \in \mathcal{H} \times \mathcal{H} \; \left| \widehat{\mathbb{E}}_{S_1}[\Delta f_2(h_1, h_2)] - \mathbb{E}_x[\Delta f_2(h_1, h_2; x)] \right| \right.$$
$$\left. \leq \sigma_{f_2}(h_1, h_2)\sqrt{\frac{32 \log(|\mathcal{H}|^2/\delta)}{mk}} \right) \geq 1 - \delta \tag{13}$$

with $\sigma_{f_2}^2(h_1, h_2) = \mathrm{Var}_x(\Delta f_2(h_1, h_2; x))$. Both of the above probabilities have been taken w.r.t. $S_1$

Moreover, a standard Hoeffding bound reveals that

$$\mathbb{P}\left( \left| \widehat{p}_{S_2} - p \right| \leq \sqrt{\frac{\log(2/\delta)}{2mk}} \right) \geq 1 - \delta \,, \tag{14}$$

the probability being over $S_2$.

From Lemma 3.1 we have

$$\left| \mathbb{E}_z[\widetilde{\Delta}\ell_b(h_1, h_2; z) \,|\, S_1, S_2] - \Delta\mathcal{L}(h_1, h_2) \right| = \left| \mathbb{E}_z[\widetilde{\Delta}\ell_b(h_1, h_2; z) \,|\, S_1, S_2] - \mathbb{E}_z[\Delta\ell_b(h_1, h_2; z)] \right| \,.$$

We now compute separately the two expectations $\mathbb{E}_z[\widetilde{\Delta}\ell_b(h_1, h_2; z) \,|\, S_1, S_2]$ and $\mathbb{E}_z[\Delta\ell_b(h_1, h_2; z)]$.

---

[3]This theorem holds for any median involved in the MoM definition, hence also for our "anti-symmetric" median defined in Section 3.1.

Let us introduce some shorthand notation first. Set, for fixed $h_1, h_2$,

$$\mathbb{E}_1 = \mathbb{E}_x[\Delta f_1(h_1, h_2; x)]\,, \quad \mathbb{E}_2 = \mathbb{E}_x[\Delta f_2(h_1, h_2; x)]\,, \quad \Sigma = \sum_{i=1}^{k} \Delta f_2(h_1, h_2; x_i)$$

$$\widehat{\mathbb{E}}_1 = \widehat{\mathbb{E}}_{S_1}[\Delta f_1(h_1, h_2)]\,, \quad \widehat{\mathbb{E}}_2 = \widehat{\mathbb{E}}_{S_1}[\Delta f_2(h_1, h_2)]\,, \quad \widehat{p} = \widehat{p}_{S_2}\,.$$

Then

$$\mathbb{E}_z[\widetilde{\Delta}\ell_b(h_1, h_2; z) \mid S_1, S_2] = \widehat{\mathbb{E}}_1 + \widehat{p}\,\widehat{\mathbb{E}}_2 + \mathbb{E}_z\Big[(\alpha - \widehat{p})\Big(\Sigma - k\widehat{\mathbb{E}}_2\Big) \mid S_1, S_2\Big]$$

$$= \widehat{\mathbb{E}}_1 + \widehat{p}\,\widehat{\mathbb{E}}_2 + \mathbb{E}_z[\alpha\Sigma] - kp\widehat{\mathbb{E}}_2 - \widehat{p}\,k\mathbb{E}_2 + k\widehat{p}\,\widehat{\mathbb{E}}_2\,.$$

Since for $i \neq j$, the variables $y_i$ and $\Delta f_2(h_1, h_2; x_j)$ are independent, we have

$$\mathbb{E}_z[\alpha\Sigma] = \frac{1}{k}\,\mathbb{E}\Big[\Big(\sum_{i=1}^{k} y_i\Big)\Big(\sum_{i=1}^{k} \Delta f_2(h_1, h_2; x_i)\Big)\Big] = \mathbb{E}_{(x,y)}[y\Delta f_2(h_1, h_2; x)] + (k-1)p\mathbb{E}_2\,,$$

so that

$$\mathbb{E}_z[\widetilde{\Delta}\ell_b(h_1, h_2; z) \mid S_1, S_2]$$
$$= \widehat{\mathbb{E}}_1 + (k+1)\widehat{p}\,\widehat{\mathbb{E}}_2 + \mathbb{E}_{(x,y)}[y\Delta f_2(h_1, h_2; x)] + (k-1)p\mathbb{E}_2 - kp\widehat{\mathbb{E}}_2 - \widehat{p}\,k\mathbb{E}_2\,. \tag{15}$$

Moreover, from Lemma 3.1,

$$\mathbb{E}_z[\Delta\ell_b(h_1, h_2; z)] = \mathbb{E}_{(x,y)}[\ell(h_1(x), y)] - \mathbb{E}_{(x,y)}[\ell(h_2(x), y)] = \mathbb{E}_1 + \mathbb{E}_{(x,y)}[y\Delta f_2(h_1, h_2; x)]\,.$$

Thus we can write

$$\Big|\mathbb{E}_z[\widetilde{\Delta}\ell_b(h_1, h_2; z) \mid S_1, S_2] - \mathbb{E}_z[\Delta\ell_b(h_1, h_2; z)]\Big|$$
$$= \Big|\widehat{\mathbb{E}}_1 - \mathbb{E}_1 + \widehat{p}\,\widehat{\mathbb{E}}_2 + (k-1)p\mathbb{E}_2 - kp\widehat{\mathbb{E}}_2 - \widehat{p}\,k\mathbb{E}_2 + k\widehat{p}\,\widehat{\mathbb{E}}_2\Big|$$
$$= \Big|\widehat{\mathbb{E}}_1 - \mathbb{E}_1 + \widehat{p}\,\widehat{\mathbb{E}}_2 - p\mathbb{E}_2 + k(\mathbb{E}_2 - \widehat{\mathbb{E}}_2)(p - \widehat{p})\Big|$$
$$\leq \Big|\widehat{\mathbb{E}}_1 - \mathbb{E}_1\Big| + \Big|\widehat{p}\,\widehat{\mathbb{E}}_2 - p\mathbb{E}_2\Big| + k\Big|\mathbb{E}_2 - \widehat{\mathbb{E}}_2\Big|\,\Big|p - \widehat{p}\Big|$$
$$= \Big|\widehat{\mathbb{E}}_1 - \mathbb{E}_1\Big| + \Big|\widehat{p}\,\widehat{\mathbb{E}}_2 - \widehat{p}\,\mathbb{E}_2 + \widehat{p}\,\mathbb{E}_2 - p\mathbb{E}_2\Big| + k\Big|\mathbb{E}_2 - \widehat{\mathbb{E}}_2\Big|\,\Big|p - \widehat{p}\Big|$$
$$\leq \Big|\widehat{\mathbb{E}}_1 - \mathbb{E}_1\Big| + \widehat{p}\Big|\widehat{\mathbb{E}}_2 - \mathbb{E}_2\Big| + |\mathbb{E}_2|\,\Big|p - \widehat{p}\Big| + k\Big|\mathbb{E}_2 - \widehat{\mathbb{E}}_2\Big|\,\Big|p - \widehat{p}\Big|$$
$$\leq \Big|\widehat{\mathbb{E}}_1 - \mathbb{E}_1\Big| + \Big|\widehat{\mathbb{E}}_2 - \mathbb{E}_2\Big| + |\mathbb{E}_2|\,\Big|p - \widehat{p}\Big| + k\Big|\mathbb{E}_2 - \widehat{\mathbb{E}}_2\Big|\,\Big|p - \widehat{p}\Big|\,.$$

At this point, we use (12) to bound $\Big|\widehat{\mathbb{E}}_1 - \mathbb{E}_1\Big|$, (13) to bound $\Big|\widehat{\mathbb{E}}_2 - \mathbb{E}_2\Big|$, and (14) to bound $|p - \widehat{p}|$. This yields, with probability at least $1 - 3\delta$ over the generation of $S_1$ and $S_2$, uniformly over $h_1, h_2 \in \mathcal{H}$,

$$\left| \mathbb{E}_z[\widetilde{\Delta}\ell_b(h_1, h_2; z) \,|\, S_1, S_2] - \mathbb{E}_z[\Delta\ell_b(h_1, h_2; z)] \right|$$

$$\leq \sigma_{f_1}(h_1, h_2)\sqrt{\frac{32\log(|\mathcal{H}|^2/\delta)}{mk}} + \sigma_{f_2}(h_1, h_2)\sqrt{\frac{32\log(|\mathcal{H}|^2/\delta)}{mk}} + |\mathbb{E}_2|\sqrt{\frac{\log(2/\delta)}{2mk}}$$

$$+ k\,\sigma_{f_2}(h_1, h_2)\sqrt{\frac{32\log(|\mathcal{H}|^2/\delta)}{mk}}\sqrt{\frac{\log(2/\delta)}{2mk}}$$

$$= O\left( \left(\sigma_{f_1}(h_1, h_2) + \sigma_{f_2}(h_1, h_2)\right)\sqrt{\frac{\log(|\mathcal{H}|/\delta)}{mk}} \right) + O\left( |\mathbb{E}_2|\sqrt{\frac{\log(1/\delta)}{mk}} \right)$$

$$+ O\left( \sigma_{f_2}(h_1, h_2)\sqrt{\frac{\log(|\mathcal{H}|/\delta)}{m}}\sqrt{\frac{\log(1/\delta)}{m}} \right)$$

$$= O\left( \sigma_{f_1}(h_1, h_2)\sqrt{\frac{\log(|\mathcal{H}|/\delta)}{mk}} \right) + O\left( |\mathbb{E}_2|\sqrt{\frac{\log(1/\delta)}{mk}} \right) + O\left( \sigma_{f_2}(h_1, h_2)\sqrt{\frac{\log(1/\delta)}{m}} \right) , \qquad (16)$$

where the last equality uses the assumption $m = \Omega(\log(|\mathcal{H}|/\delta))$.

As for the variance $\mathrm{Var}_z(\widetilde{\Delta}\ell_b(h_1, h_2; z) \,|\, S_1, S_2)$, notice that in the conditional space where $S_1$ and $S_2$ are given, the bound on this variance can be obtained by adapting the argument contained in Lemma 3.1.

Let $z = (\mathcal{B}, \alpha)$, with $\mathcal{B} = (x_1, \ldots, x_k)$, and introduce the short-hand notation

$$\widetilde{\alpha} = \alpha - \widehat{p}, \qquad \widetilde{\Delta}f_2 = \frac{1}{k}\sum_{i=1}^{k}\Delta f_2(h_1, h_2; x_i)) - \widehat{\mathbb{E}}_2 = \frac{\Sigma}{k} - \widehat{\mathbb{E}}_2 .$$

With this notation

$$\widetilde{\Delta}\ell_b(h_1, h_2; z) = \widehat{\mathbb{E}}_1 + \widehat{p}\widehat{\mathbb{E}}_2 + k\,\widetilde{\alpha}\,\widetilde{\Delta}f_2 .$$

Thus, in the conditional space where $S_1$ and $S_2$ are given, we follow the argument in Lemma 3.1 (and omit the conditioning on $S_1$ and $S_2$ for simplicity of notation). We have

$$\frac{1}{k^2}\,\mathrm{Var}_z(\widetilde{\Delta}\ell_b(h_1, h_2; z)) = \mathrm{Var}_z\left( \widetilde{\alpha}\,\widetilde{\Delta}f_2 \right)$$

$$= \mathbb{E}_{\mathcal{B}}\left[ \mathrm{Var}_\alpha(\widetilde{\alpha}_j\,\widetilde{\Delta}f_2 \,|\, \mathcal{B}) \right] + \mathrm{Var}_{\mathcal{B}}\left( \mathbb{E}_\alpha\left[ \widetilde{\alpha}\,\widetilde{\Delta}f_2 \,|\, \mathcal{B} \right] \right)$$

$$= \mathbb{E}_{\mathcal{B}}\left[ (\widetilde{\Delta}f_2)^2\,\mathrm{Var}_\alpha(\widetilde{\alpha} \,|\, \mathcal{B}) \right] + \mathrm{Var}_{\mathcal{B}}\left( \widetilde{\Delta}f_2\,\mathbb{E}_\alpha\left[ \widetilde{\alpha} \,|\, \mathcal{B} \right] \right)$$

$$= \mathbb{E}_{\mathcal{B}}\left[ (\widetilde{\Delta}f_2)^2\,\frac{\sum_{i=1}^{k}\eta(x_i)(1 - \eta(x_i))}{k^2} \right] + \mathrm{Var}_{\mathcal{B}}\left( \widetilde{\Delta}f_2\left( \frac{1}{k}\sum_{i=1}^{k}\eta(x_i) - \widehat{p} \right) \right)$$

$$\leq \frac{1}{4k}\mathbb{E}_{\mathcal{B}}\left[ (\widetilde{\Delta}f_2)^2 \right] + \mathrm{Var}_{\mathcal{B}}\left( \widetilde{\Delta}f_2\left( \frac{1}{k}\sum_{i=1}^{k}\eta(x_i) - \widehat{p} \right) \right) .$$

Now,

$$\mathbb{E}_{\mathcal{B}}\left[ (\widetilde{\Delta}f_2)^2 \right] = \mathbb{E}_{\mathcal{B}}\left[ \left( \frac{\Sigma}{k} - \widehat{\mathbb{E}}_2 \right)^2 \right]$$

$$= \mathbb{E}_{\mathcal{B}}\left[ \left( \frac{\Sigma}{k} - \mathbb{E}_2 + \mathbb{E}_2 - \widehat{\mathbb{E}}_2 \right)^2 \right]$$

$$\leq 2\mathbb{E}_{\mathcal{B}}\left[ \left( \frac{\Sigma}{k} - \mathbb{E}_2 \right)^2 \right] + 2\left( \mathbb{E}_2 - \widehat{\mathbb{E}}_2 \right)^2$$

$$= 2\,\mathrm{Var}_{\mathcal{B}}\left( \frac{\Sigma}{k} \right) + 2\left( \mathbb{E}_2 - \widehat{\mathbb{E}}_2 \right)^2$$

$$= \frac{2}{k}\sigma_{f_2}^2(h_1, h_2) + 2\left( \mathbb{E}_2 - \widehat{\mathbb{E}}_2 \right)^2 .$$

Plugging back gives

$$
\begin{aligned}
\frac{1}{k^2}\,\mathrm{Var}_z(\widetilde{\Delta}\ell_b(h_1,h_2;z)) &\leq \frac{1}{2k^2}\sigma_{f_2}^2(h_1,h_2) + \frac{1}{2k}\left(\mathbb{E}_2 - \widehat{\mathbb{E}}_2\right)^2 + \mathrm{Var}_{\mathcal{B}}\left(\widetilde{\Delta}f_2\left(\frac{1}{k}\sum_{i=1}^{k}\eta(x_i) - \widehat{p}\right)\right) \\
&\leq \frac{1}{2k^2}\sigma_{f_2}^2(h_1,h_2) + \sigma_{f_2}^2(h_1,h_2)\frac{32\log(|\mathcal{H}|^2/\delta)}{2mk^2} \\
&\quad + \mathrm{Var}_{\mathcal{B}}\left(\widetilde{\Delta}f_2\left(\frac{1}{k}\sum_{i=1}^{k}\eta(x_i) - \widehat{p}\right)\right),
\end{aligned}
$$

the last inequality holding with probability at least $1 - \delta$ over the generation of $S_1$ by virtue of (13).

The term $\mathrm{Var}_{\mathcal{B}}\left(\widetilde{\Delta}f_2\left(\frac{1}{k}\sum_{i=1}^{k}\eta(x_i) - \widehat{p}\right)\right)$ can be treated similar to Lemma 3.1, with the caveat that now the variables involved are no longer zero mean.

Similar to Lemma 3.1, define

$$
\widetilde{\eta}_i' = \eta(x_i) - \widehat{p} = \widetilde{\eta}_i + p - \widetilde{p}\,.
$$

and

$$
\widetilde{\Delta}f_{2,i}' = \Delta f_2(h_1,h_2;x_i) - \widehat{\mathbb{E}}_2 = \widetilde{\Delta}f_{2,i} + \mathbb{E}_2 - \widehat{\mathbb{E}}_2\,.
$$

Hence $\widetilde{\eta}_i$ and $\widetilde{\Delta}f_{2,i}$ are zero mean random variables.

We have

$$
\begin{aligned}
\mathrm{Var}_{\mathcal{B}}\left(\widetilde{\Delta}f_2\left(\frac{1}{k}\sum_{i=1}^{k}\eta(x_i) - \widehat{p}\right)\right) &= \mathrm{Var}_{\mathcal{B}}\left(\left(\frac{1}{k}\sum_{i=1}^{k}\widetilde{\Delta}f_{2,i}'\right)\left(\frac{1}{k}\sum_{i=1}^{k}\widetilde{\eta}_i'\right)\right) \\
&= \frac{1}{k^4}\,\mathrm{Var}_{\mathcal{B}}\left(\left(\sum_{i=1}^{k}\widetilde{\Delta}f_{2,i}'\right)\left(\sum_{i=1}^{k}\widetilde{\eta}_i'\right)\right) \\
&\leq \frac{1}{k^4}\,\mathbb{E}_{\mathcal{B}}\left[\left(\sum_{i=1}^{k}\widetilde{\Delta}f_{2,i}'\right)^2\left(\sum_{i=1}^{k}\widetilde{\eta}_i'\right)^2\right] \\
&= \frac{1}{k^4}\,\mathbb{E}_{\mathcal{B}}\left[\left(\sum_{i=1}^{k}\widetilde{\Delta}f_{2,i} + \mathbb{E}_2 - \widehat{\mathbb{E}}_2\right)^2\left(\sum_{i=1}^{k}\widetilde{\eta}_i + p - \widetilde{p}\right)^2\right] \\
&\leq \frac{4}{k^4}\,\mathbb{E}_{\mathcal{B}}\left[\left(\left(\sum_{i=1}^{k}\widetilde{\Delta}f_{2,i}\right)^2 + \left(\mathbb{E}_2 - \widehat{\mathbb{E}}_2\right)^2\right)\left(\left(\sum_{i=1}^{k}\widetilde{\eta}_i\right)^2 + \left(p - \widetilde{p}\right)^2\right)\right].
\end{aligned}
$$

Now, from (13) and (14) we have, with probability at least $1 - 2\delta$ over the generation of $S_1$ and $S_2$, and uniformly over $h_1, h_2$,

$$
\left(\mathbb{E}_2 - \widehat{\mathbb{E}}_2\right)^2 \leq \sigma_{f_2}^2(h_1,h_2)\frac{32\log(|\mathcal{H}|^2/\delta)}{mk}\,, \qquad \left(p - \widetilde{p}\right)^2 \leq \frac{\log(2/\delta)}{2mk}\,,
$$

Thus

$$\mathrm{Var}_{\mathcal{B}}\left(\widetilde{\Delta}f_2\left(\frac{1}{k}\sum_{i=1}^{k}\eta(x_i)-\widehat{p}\right)\right)$$

$$\leq \frac{4}{k^4}\,\mathbb{E}_{\mathcal{B}}\left[\left(\sum_{i=1}^{k}\widetilde{\Delta}f_{2,i}\right)^2\left(\sum_{i=1}^{k}\widetilde{\eta}_i\right)^2\right]$$

$$+\frac{4}{k^4}\,\frac{\log(2/\delta)}{2mk}\,\mathbb{E}_{\mathcal{B}}\left[\left(\sum_{i=1}^{k}\widetilde{\Delta}f_{2,i}\right)^2\right]$$

$$+\frac{4}{k^4}\,\sigma_{f_2}^2(h_1,h_2)\frac{32\log(|\mathcal{H}|^2/\delta)}{mk}\,\mathbb{E}_{\mathcal{B}}\left[\left(\sum_{i=1}^{k}\widetilde{\eta}_i\right)^2\right]$$

$$+\frac{4}{k^4}\,\sigma_{f_2}^2(h_1,h_2)\frac{32\log(|\mathcal{H}|^2/\delta)}{mk}\,\frac{\log(2/\delta)}{2mk}\;.$$

Now, as in the proof of Lemma 3.1 (see Eq. (10) there),

$$\frac{4}{k^4}\,\mathbb{E}_{\mathcal{B}}\left[\left(\sum_{i=1}^{k}\widetilde{\Delta}f_{2,i}\right)^2\left(\sum_{i=1}^{k}\widetilde{\eta}_i\right)^2\right]\leq \frac{4}{k^4}\,\frac{9}{4}\,k^2\,\sigma_{f_2}^2(h_1,h_2)=\frac{9}{k^2}\,\sigma_{f_2}^2(h_1,h_2)\;.$$

Moreover, since the variables $\widetilde{\Delta}f_{2,i}$ and $\widetilde{\eta}_i$ are zero mean, we also have (see again the proof of Lemma 3.1, just above Eq. (10))

$$\mathbb{E}_{\mathcal{B}}\left[\left(\sum_{i=1}^{k}\widetilde{\Delta}f_{2,i}\right)^2\right]=k\,\sigma_{f_2}^2(h_1,h_2)$$

and

$$\mathbb{E}_{\mathcal{B}}\left[\left(\sum_{i=1}^{k}\widetilde{\eta}_i\right)^2\right]=k\,\mathrm{Var}_x(\eta(x))\leq \frac{k}{4}\;.$$

Plugging back we conclude that

$$\mathrm{Var}_{\mathcal{B}}\left(\widetilde{\Delta}f_2\left(\frac{1}{k}\sum_{i=1}^{k}\eta(x_i)-\widehat{p}\right)\right)=O\left(\sigma_{f_2}^2(h_1,h_2)\left(\frac{1}{k^2}+\frac{\log(|\mathcal{H}|/\delta)}{m\,k^4}+\frac{\log(|\mathcal{H}|/\delta)\,\log(1/\delta)}{m^2\,k^6}\right)\right)\;.$$

Therefore,

$$\frac{1}{k^2}\,\mathrm{Var}_z(\widetilde{\Delta}\ell_b(h_1,h_2;z))=O\left(\sigma_{f_2}^2(h_1,h_2)\left(\frac{1}{k^2}+\frac{\log(|\mathcal{H}|/\delta)}{m\,k^2}+\frac{\log(|\mathcal{H}|/\delta)\,\log(1/\delta)}{m^2\,k^6}\right)\right)\;,$$

and

$$\sigma^2(h_1,h_2)=\mathrm{Var}_z(\widetilde{\Delta}\ell_b(h_1,h_2;z)\,|\,S_1,S_2)=O\left(\sigma_{f_2}^2(h_1,h_2)\left(1+\frac{\log(|\mathcal{H}|/\delta)}{m}+\frac{\log(|\mathcal{H}|/\delta)\,\log(1/\delta)}{m^2\,k^4}\right)\right)$$

holds with probability at least $1-3\delta$ over the random generation of $S_1$ and $S_2$, uniformly over $h_1,h_2\in\mathcal{H}$. Since we assumed

$$m=\Omega\left(\log(|\mathcal{H}|/\delta)\right)$$

this finally implies

$$\sigma^2(h_1,h_2)=\mathrm{Var}_z(\widetilde{\Delta}\ell_b(h_1,h_2;z)\,|\,S_1,S_2)=O\left(\sigma_{f_2}^2(h_1,h_2)\right) \tag{17}$$

with probability at least $1-\delta$ over the random generation of $S_1$ and $S_2$, uniformly over $h_1,h_2\in\mathcal{H}$.

Let us henceforth assume that $S_1$ and $S_2$ have been drawn in such a way that both (16) and (17) simultaneously hold for all $h_1,h_2$.

Then, combining (16), (17), and (11) one can easily see that

$$
\mathbb{P}\left( \forall h_1, h_2 \in \mathcal{H} \times \mathcal{H} \ \left| Q(h_1, h_2; S_3) - \Delta\mathcal{L}(h_1, h_2) \right| \leq O\left( \sigma_{f_2}(h_1, h_2) \sqrt{\frac{\log(|\mathcal{H}|/\delta)}{m}} \right) \right.
$$

$$
\left. + O\left( \sigma_{f_1}(h_1, h_2) \sqrt{\frac{\log(|\mathcal{H}|/\delta)}{mk}} \right) + O\left( |\mathbb{E}_2| \sqrt{\frac{\log(1/\delta)}{mk}} \right) + O\left( \sigma_{f_2}(h_1, h_2) \sqrt{\frac{\log(1/\delta)}{m}} \right) \right)
$$

$$
= \mathbb{P}\left( \forall h_1, h_2 \in \mathcal{H} \times \mathcal{H} \ \left| Q(h_1, h_2; S_3) - \Delta\mathcal{L}(h_1, h_2) \right| \leq C_\delta(h_1, h_2, |\mathcal{H}|, m, k) \right) \geq 1 - \delta , \tag{18}
$$

holds under the condition $m = \Omega(\log(|\mathcal{H}|/\delta))$, where we set for brevity

$$
C_\delta(h_1, h_2, |\mathcal{H}|, m, k) = O\left( \sigma_{f_2}(h_1, h_2) \sqrt{\frac{\log(|\mathcal{H}|/\delta)}{m}} + \sigma_{f_1}(h_1, h_2) \sqrt{\frac{\log(|\mathcal{H}|/\delta)}{mk}} + |\mathbb{E}_2| \sqrt{\frac{\log(1/\delta)}{mk}} \right) ,
$$

and where now the probability is w.r.t. the generation of $S_1, S_2$, and $S_3$.

We are now ready to prove the correctness and the sample complexity of Algorithm 1 in Section 3.

**Proposition B.1.** *Let $\mathcal{H}_\beta = \{ h \in \mathcal{H} \ : \ Reg(h) \geq \beta \}$, and assume $m$ is such that, for $h \in \mathcal{H}_{\beta/4}$, we have*

$$
C_\delta(h, \widehat{h}^\star, |\mathcal{H}|, m, k) \leq \Delta\mathcal{L}(h, \widehat{h}^\star)/4 , \tag{19}
$$

*for all $h \notin \mathcal{H}_{\beta/4}$ we have*

$$
C_\delta(h, \widehat{h}^\star, |\mathcal{H}|, m, k) \leq \frac{\beta}{4} , \tag{20}
$$

*along with $m = \Omega(\log(|\mathcal{H}|/\delta))$. Then with probability $\geq 1 - \delta$ over the generation of $S_1, S_2$, and $S_3$ the following holds:*

1. *If $Reg(h) = \Delta\mathcal{L}(h, \widehat{h}^\star) \geq \beta$ then $h$ is eliminated by $\widehat{h}^\star$ ;*

2. *$\widehat{h}^\star$ is not eliminated. In particular, at the end of the tournament $\mathcal{P}$ is not empty (as it contains at least $\widehat{h}^\star$).*

*Proof.* Let $h \in \mathcal{H}_{\beta/4}$. We have, with probability $\geq 1 - \delta$

$$
Q(h, \widehat{h}^\star; S) \geq \Delta\mathcal{L}(h, \widehat{h}^\star) - C_\delta(h, \widehat{h}^\star, |\mathcal{H}|, m, k)
$$
$$
\text{(from (18))}
$$
$$
\geq \frac{3}{4}\Delta\mathcal{L}(h, \widehat{h}^\star)
$$
$$
\text{(from (19))} .
$$

Now, if $\Delta\mathcal{L}(h, \widehat{h}^\star) \geq \beta$ (clearly this implies $h \in \mathcal{H}_{\beta/4}$) then $Q(h, \widehat{h}^\star; S) \geq \frac{3}{4}\beta > \beta/2$, thus $h$ is eliminated by $\widehat{h}^\star$. This proves the first item.

As for the second item, consider any $h \in \mathcal{H}$, and note that $\Delta\mathcal{L}(h, \widehat{h}^\star) \geq 0$ (since $\widehat{h}^\star$ is the best in class) and that, by definition, $Q(\widehat{h}^\star, h; S) = -Q(h, \widehat{h}^\star; S)$.

We separate two cases:

- $\Delta\mathcal{L}(h, \widehat{h}^\star) \geq \beta/4$ (which is equivalent to $h \in \mathcal{H}_{\beta/4}$). In this case, from the previous chain of inequalities, $Q(h, \widehat{h}^\star; S) \geq \frac{3}{16}\beta > 0$, and $\widehat{h}^\star$ is not eliminated as a second argument of $Q$. On the other hand,

$$
Q(\widehat{h}^\star, h; S) = -Q(h, \widehat{h}^\star; S) \leq -\frac{3}{16}\beta < \beta/2 .
$$

  Hence $\widehat{h}^\star$ cannot be eliminated as first argument of $Q$ either.

- $\Delta\mathcal{L}(h, \widehat{h}^\star) < \beta/4$ (which is equivalent to $h \notin \mathcal{H}_{\beta/4}$). In this case, we have, with probability $\geq 1 - \delta$,

$$Q(h, \widehat{h}^\star; S) \geq \Delta\mathcal{L}(h, \widehat{h}^\star) - C_\delta(h, \widehat{h}^\star, |\mathcal{H}|, m, k)$$
$$\text{(from (18))}$$
$$\geq \Delta\mathcal{L}(h, \widehat{h}^\star) - \beta/4$$
$$\text{(from (20))}.$$

The above expression larger than $-\beta/2$ since $\Delta\mathcal{L}(h, \widehat{h}^\star) \geq 0$. Hence, $\widehat{h}^\star$ will not be eliminated as second argument of $Q$. On the other hand,

$$Q(\widehat{h}^\star, h; S) = -Q(h, \widehat{h}^\star; S) \leq -\Delta\mathcal{L}(h, \widehat{h}^\star) + \beta/4 \leq \beta/4 < \beta/2 .$$

Hence $\widehat{h}^\star$ will not be eliminated as first argument of $Q$ either.

This concludes the proof. $\qquad\qquad\qquad\qquad\qquad\qquad\qquad\qquad\qquad\qquad\qquad\qquad\qquad\qquad\qquad\square$

*Proof.* [of Theorem 3.2] From Proposition B.1 we see that with probability $\geq 1 - \delta$ the pool $\mathcal{P}$ at the end of the tournament contains $\widehat{h}^\star$, plus possibly other $h \in \mathcal{H}$ such that $Reg(h) \leq \beta$, so any remaining $h \in \mathcal{P}$ will have $Reg(h) \leq \beta$.

We are left with the task of computing the sample complexity of Algorithm 1. From (19) we know that we must guarantee that

$$O\left(\sigma_{f_2}(h, \widehat{h}^\star)\sqrt{\frac{\log(|\mathcal{H}|/\delta)}{m}} + \sigma_{f_1}(h, \widehat{h}^\star)\sqrt{\frac{\log(|\mathcal{H}|/\delta)}{mk}} + |\mathbb{E}_2|\sqrt{\frac{\log(1/\delta)}{mk}}\right) \leq \frac{\Delta\mathcal{L}(h, \widehat{h}^\star)}{4} \tag{21}$$

for all $h \in \mathcal{H}_{\beta/4}$ and, from (20),

$$O\left(\sigma_{f_2}(h, \widehat{h}^\star)\sqrt{\frac{\log(|\mathcal{H}|/\delta)}{m}} + \sigma_{f_1}(h, \widehat{h}^\star)\sqrt{\frac{\log(|\mathcal{H}|/\delta)}{mk}} + |\mathbb{E}_2|\sqrt{\frac{\log(1/\delta)}{mk}}\right) \leq \frac{\beta}{4} , \tag{22}$$

for all $h \notin \mathcal{H}_{\beta/4}$, where $\mathbb{E}_2$ above has to be interpreted as

$$\mathbb{E}_2 = \mathbb{E}_x[\Delta f_2(h, \widehat{h}^\star; x)] .$$

Moreover, the condition $m = \Omega(\log(|\mathcal{H}|/\delta))$ should also be satisfied.

As for (21), since $h \in \mathcal{H}_{\beta/4}$, that is, $\Delta\mathcal{L}(h, \widehat{h}^\star) \geq \beta/4$, for this inequality to hold, it suffices to have

$$m = \Omega\left(\frac{\sigma_{f_2}^2(h, \widehat{h}^\star)\log(|\mathcal{H}|/\delta)}{\beta^2} + \frac{\sigma_{f_1}^2(h, \widehat{h}^\star)\log(|\mathcal{H}|/\delta) + \mathbb{E}_x^2[\Delta f_2(h, \widehat{h}^\star; x)]\log(1/\delta)}{k\beta^2}\right) .$$

for all $h \in \mathcal{H}_{\beta/4}$.

On the other hand, the condition in (22) is satisfied when

$$m = \Omega\left(\frac{\sigma_{f_2}^2(h, \widehat{h}^\star)\log(|\mathcal{H}|/\delta)}{\beta^2} + \frac{\sigma_{f_1}^2(h, \widehat{h}^\star)\log(|\mathcal{H}|/\delta) + \mathbb{E}_x^2[\Delta f_2(h, \widehat{h}^\star; x)]\log(1/\delta)}{k\beta^2}\right) .$$

for all $h \notin \mathcal{H}_{\beta/4}$. Furthermore we need to add $m = \Omega(\log(|\mathcal{H}|/\delta))$.

Overall,

$$m = \Omega\left(\frac{\sigma_{f_2}^2(h, \widehat{h}^\star)\log(|\mathcal{H}|/\delta)}{\beta^2} + \frac{\sigma_{f_1}^2(h, \widehat{h}^\star)\log(|\mathcal{H}|/\delta) + \mathbb{E}_x^2[\Delta f_2(h, \widehat{h}^\star; x)]\log(1/\delta)}{k\beta^2}\right) .$$

for all $h \in \mathcal{H}$, along with $m = \Omega(\log(|\mathcal{H}|/\delta))$.

In the above, we further upper bound the relevant factors as

$$\sigma^2_{f_i}(h, \widehat{h}^\star) = \mathrm{Var}_x\Big( f_i(h(x)) - f_i(\widehat{h}^\star(x)) \Big)$$
$$\leq \max_{h \in \mathcal{H}} \mathrm{Var}_x\Big( f_i(h(x)) - f_i(\widehat{h}^\star(x)) \Big)$$
$$= \sigma^2_{f_i}(\mathcal{H}) ,$$

along with

$$\mathbb{E}^2_x[\Delta f_2(h, \widehat{h}^\star; x)] \leq \max_{h \in \mathcal{H}} \mathbb{E}^2_x[\Delta f_2(h, \widehat{h}^\star; x)] = \mathbb{E}^2[\mathcal{H}]$$

to conclude the proof. $\square$

### B.3. Version of Theorem 3.2 with Variable Bag Sizes

Denote by $k_j$ the size of the $j$-th bag, and consider the split

$$S = \{\underbrace{z_1, \ldots, z_m}_{S_1}, \underbrace{z_{m+1}, \ldots z_{2m}}_{S_2}\} . \tag{23}$$

Now subset $S_1$ will be used to estimate $\mathbb{E}_x[\Delta f_1(h_1, h_2; x)]$, $\mathbb{E}_x[\Delta f_2(h_1, h_2; x)]$, and $p$, and subset $S_2$ will be the one out of which the LLP estimator (3) will be constructed. Before applying the split into $S_1$ and $S_2$, though, we rearrange the $2m$ bags so that the last $m$ bags (those that will end up being in $S_2$) now correspond to the $m$ smallest bags (ties broken arbitrarily). For $i = 1, 2$, denote by $\widehat{k}_i$, the average bag size in $S_i$, and by $\widehat{k}$ the average bag size over the entire dataset $S$. Then, by construction, $\widehat{k}_2 \leq \widehat{k} = (\widehat{k}_1 + \widehat{k}_2)/2$.

Apart from this simple difference in the way the split is constructed, the algorithm for the variable bag size is the very same as the one with constant bag sizes. The following generalization of Theorem 3.2 holds.

**Theorem B.2.** *Let $S = (x_1, y_1), \ldots, (x_n, y_n)$ be drawn i.i.d. from a distribution $\mathcal{D}$ over $\mathcal{X} \times \{0, 1\}$. Let the loss $\ell(\cdot, y) = f_1(\cdot) + y f_2(\cdot)$ be such that the maximal second moments $\max_{h \in \mathcal{H}} \mathbb{E}_x\big[(f_1(h(x)))^2\big]$ and $\max_{h \in \mathcal{H}} \mathbb{E}_x\big[(f_2(h(x)))^2\big]$ are finite. Let $\widehat{h}^\star = \min_{h \in \mathcal{H}} \mathcal{L}(h)$ be the best-in-class hypothesis. If $S$ is split into $2m$ random bags as described above with $n = m \times (\widehat{k}_1 + \widehat{k}_2)$, then for all $\beta > 0$, the hypothesis $\widehat{h}$ output by Algorithm 1 satisfies $Reg(\widehat{h}) \leq \beta$ with probability at least $1 - \delta$, provided*

$$n = O\left( \frac{\widehat{k}\, \sigma^2_{f_2}(\mathcal{H})\, \log(|\mathcal{H}|/\delta)}{\beta^2} + \frac{\sigma^2_{f_1}(\mathcal{H})\, \log(|\mathcal{H}|/\delta) + \mathbb{E}^2[\mathcal{H}]\, \log(1/\delta)}{\beta^2} + \widehat{k}\, \log(|\mathcal{H}|/\delta) \right) ,$$

*where the big-oh notation only hides absolute constants, and*

$$\sigma^2_{f_i}(\mathcal{H}) = \max_{h \in \mathcal{H}} \mathrm{Var}_x\Big( f_i(h(x)) - f_i(\widehat{h}^\star(x)) \Big), \ i = 1, 2$$
$$\mathbb{E}^2[\mathcal{H}] = \max_{h \in \mathcal{H}} \mathbb{E}^2_x\Big[ f_2(h(x)) - f_2(\widehat{h}^\star(x)) \Big] .$$

*Proof.* The proof is very similar to the proof of Theorem 3.2, so in what follows we only sketch the main differences.

We shall apply Lemma 3.1 to $S_2$ with average bag size $\widehat{k}_2 \leq \widehat{k}$. Equation (11) holds independent of bag sizes. Equations (12) and (13) apply with $k$ therein replaced by $\widehat{k}_1$, but also Equation (14) applies with $k$ replaced by $\widehat{k}_1$. Compared to the proof of Theorem 3.2, since we are using the same $S_1$ for all these estimators, we only have to replace $\delta$ by $\delta/2$ throughout.

Following the proof of Theorem 3.2, we conclude that

$$\Big| \mathbb{E}_z[\widetilde{\Delta}\ell_b(h_1, h_2; z) \mid S_1, S_2] - \mathbb{E}_z[\Delta\ell_b(h_1, h_2; z)] \Big| \leq \Big| \widehat{\mathbb{E}}_1 - \mathbb{E}_1 \Big| + \Big| \widehat{\mathbb{E}}_2 - \mathbb{E}_2 \Big| + |\mathbb{E}_2|\, \Big| p - \widehat{p} \Big| + \widehat{k}_2 \Big| \mathbb{E}_2 - \widehat{\mathbb{E}}_2 \Big|\, \Big| p - \widehat{p} \Big| .$$

As in the proof of Theorem 3.2, we use (12) to bound $\left|\widehat{\mathbb{E}}_1 - \mathbb{E}_1\right|$, (13) to bound $\left|\widehat{\mathbb{E}}_2 - \mathbb{E}_2\right|$, and (14) to bound $|p - \widehat{p}|$. We conclude that with probability at least $1 - 2\delta$ over the generation of $S_1$, uniformly over $h_1, h_2 \in \mathcal{H}$,

$$
\left|\mathbb{E}_z[\widetilde{\Delta}\ell_b(h_1, h_2; z) \mid S_1, S_2] - \mathbb{E}_z[\Delta\ell_b(h_1, h_2; z)]\right|
$$

$$
\leq \sigma_{f_1}(h_1, h_2)\sqrt{\frac{32\log(|\mathcal{H}|^2/\delta)}{m\widehat{k}_1}} + \sigma_{f_2}(h_1, h_2)\sqrt{\frac{32\log(|\mathcal{H}|^2/\delta)}{m\widehat{k}_1}} + |\mathbb{E}_2|\sqrt{\frac{\log(2/\delta)}{2m\widehat{k}_1}}
$$

$$
+ \widehat{k}_2\,\sigma_{f_2}(h_1, h_2)\sqrt{\frac{32\log(|\mathcal{H}|^2/\delta)}{m\widehat{k}_1}}\sqrt{\frac{\log(2/\delta)}{2m\widehat{k}_1}}
$$

$$
= O\left(\left(\sigma_{f_1}(h_1, h_2) + \sigma_{f_2}(h_1, h_2)\right)\sqrt{\frac{\log(|\mathcal{H}|/\delta)}{m\widehat{k}_1}}\right) + O\left(|\mathbb{E}_2|\sqrt{\frac{\log(1/\delta)}{m\widehat{k}_1}}\right)
$$

$$
+ O\left(\sigma_{f_2}(h_1, h_2)\sqrt{\frac{\log(|\mathcal{H}|/\delta)}{m}}\sqrt{\frac{\log(1/\delta)}{m}}\right)
$$

$$
= O\left(\sigma_{f_1}(h_1, h_2)\sqrt{\frac{\log(|\mathcal{H}|/\delta)}{m\widehat{k}_1}}\right) + O\left(|\mathbb{E}_2|\sqrt{\frac{\log(1/\delta)}{m\widehat{k}_1}}\right) + O\left(\sigma_{f_2}(h_1, h_2)\sqrt{\frac{\log(1/\delta)}{m}}\right) .
$$

Moreover, as for the constant bag size case,

$$
\sigma^2(h_1, h_2) = \mathrm{Var}_z(\widetilde{\Delta}\ell_b(h_1, h_2; z) \mid S_1, S_2) = O\left(\sigma_{f_2}^2(h_1, h_2)\right)
$$

with probability at least $1 - \delta$ over the random generation of $S_1$, uniformly over $h_1, h_2 \in \mathcal{H}$.

Combining bias and variance above gives

$$
\mathbb{P}\left(\forall h_1, h_2 \in \mathcal{H} \times \mathcal{H}\ \left|Q(h_1, h_2; S_3) - \Delta\mathcal{L}(h_1, h_2)\right| \leq O\left(\sigma_{f_2}(h_1, h_2)\sqrt{\frac{\log(|\mathcal{H}|/\delta)}{m}}\right)\right.
$$

$$
\left. + O\left(\sigma_{f_1}(h_1, h_2)\sqrt{\frac{\log(|\mathcal{H}|/\delta)}{m\widehat{k}_1}}\right) + O\left(|\mathbb{E}_2|\sqrt{\frac{\log(1/\delta)}{m\widehat{k}_1}}\right) + O\left(\sigma_{f_2}(h_1, h_2)\sqrt{\frac{\log(1/\delta)}{m}}\right)\right)
$$

$$
= \mathbb{P}\left(\forall h_1, h_2 \in \mathcal{H} \times \mathcal{H}\ \left|Q(h_1, h_2; S_3) - \Delta\mathcal{L}(h_1, h_2)\right| \leq C_\delta(h_1, h_2, |\mathcal{H}|, m, \widehat{k}_1)\right) \geq 1 - \delta ,
$$

holds under the condition $m = \Omega(\log(|\mathcal{H}|/\delta))$, where we set for brevity

$$
C_\delta(h_1, h_2, |\mathcal{H}|, m, \widehat{k}_1) = O\left(\sigma_{f_2}(h_1, h_2)\sqrt{\frac{\log(|\mathcal{H}|/\delta)}{m}} + \sigma_{f_1}(h_1, h_2)\sqrt{\frac{\log(|\mathcal{H}|/\delta)}{m\widehat{k}_1}} + |\mathbb{E}_2|\sqrt{\frac{\log(1/\delta)}{m\widehat{k}_1}}\right) ,
$$

and where the probability is w.r.t. the generation of $S_1$ and $S_2$.

Next, Proposition B.1 holds unchanged when applied to $C_\delta(h_1, h_2, |\mathcal{H}|, m, \widehat{k}_1)$ defined above.

We finally have to compute the algorithm's sample complexity. We must guarantee that

$$
O\left(\sigma_{f_2}(h, \widehat{h}^\star)\sqrt{\frac{\log(|\mathcal{H}|/\delta)}{m}} + \sigma_{f_1}(h, \widehat{h}^\star)\sqrt{\frac{\log(|\mathcal{H}|/\delta)}{m\widehat{k}_1}} + |\mathbb{E}_2|\sqrt{\frac{\log(1/\delta)}{m\widehat{k}_1}}\right) \leq \frac{\Delta\mathcal{L}(h, \widehat{h}^\star)}{4} \tag{24}
$$

for all $h \in \mathcal{H}_{\beta/4}$ and

$$
O\left(\sigma_{f_2}(h, \widehat{h}^\star)\sqrt{\frac{\log(|\mathcal{H}|/\delta)}{m}} + \sigma_{f_1}(h, \widehat{h}^\star)\sqrt{\frac{\log(|\mathcal{H}|/\delta)}{m\widehat{k}_1}} + |\mathbb{E}_2|\sqrt{\frac{\log(1/\delta)}{m\widehat{k}_1}}\right) \leq \frac{\beta}{4} , \tag{25}
$$

for all $h \notin \mathcal{H}_{\beta/4}$, Moreover, we need to enforce the condition $m = \Omega(\log(|\mathcal{H}|/\delta))$.

As for (24), it suffices to have

$$m = \Omega \left( \frac{\sigma_{f_2}^2(h, \widehat{h}^\star) \log(|\mathcal{H}|/\delta)}{\beta^2} + \frac{\sigma_{f_1}^2(h, \widehat{h}^\star) \log(|\mathcal{H}|/\delta)}{\widehat{k}_1 \beta^2} + \frac{\mathbb{E}_x^2[\Delta f_2(h, \widehat{h}^\star; x)] \log(1/\delta)}{\widehat{k}_1 \beta^2} \right) .$$

for all $h \in \mathcal{H}_{\beta/4}$. As for (25), it suffices to have

$$m = \Omega \left( \frac{\sigma_{f_2}^2(h, \widehat{h}^\star) \log(|\mathcal{H}|/\delta)}{\beta^2} + \frac{\sigma_{f_1}^2(h, \widehat{h}^\star) \log(|\mathcal{H}|/\delta)}{\widehat{k}_1 \beta^2} + \frac{\mathbb{E}_x^2[\Delta f_2(h, \widehat{h}^\star; x)] \log(1/\delta)}{\widehat{k}_1 \beta^2} \right) .$$

for all $h \notin \mathcal{H}_{\beta/4}$. Overall,

$$m = \Omega \left( \frac{\sigma_{f_2}^2(h, \widehat{h}^\star) \log(|\mathcal{H}|/\delta)}{\beta^2} + \frac{\sigma_{f_1}^2(h, \widehat{h}^\star) \log(|\mathcal{H}|/\delta)}{\widehat{k}_1 \beta^2} + \frac{\mathbb{E}_x^2[\Delta f_2(h, \widehat{h}^\star; x)] \log(1/\delta)}{\widehat{k}_1 \beta^2} + \log(|\mathcal{H}|/\delta) \right) .$$

Multiplying by $\widehat{k}$ and recalling that $\widehat{k} \leq \widehat{k}_1$ gives the claimed bound on the sample size $n$. $\qquad \square$

## B.4. Finiteness of the Second-Moment Parameters for Log Loss in Theorem 3.2

First, observe that

$$\begin{aligned}
\sigma_{f_1}^2(\mathcal{H}) &\leq \max_{h \in \mathcal{H}} \mathbb{E}_x[(f_1(h(x)) - f_1(\widehat{h}^\star(x)))^2] \\
&\leq \max_{h \in \mathcal{H}} 2\mathbb{E}_x[(f_1(h(x)))^2] + 2\mathbb{E}_x[(f_1(\widehat{h}^\star(x)))^2] \\
&\leq 4 \max_{h \in \mathcal{H}} \mathbb{E}_x[(f_1(h(x)))^2] ,
\end{aligned}$$

and similarly

$$\sigma_{f_2}^2(\mathcal{H}) \leq 4 \max_{h \in \mathcal{H}} \mathbb{E}_x[(f_2(h(x)))^2] .$$

Moreover,

$$\begin{aligned}
\mathbb{E}^2[\mathcal{H}] &= \max_{h \in \mathcal{H}} \mathbb{E}_x^2[f_2(h(x)) - f_2(\widehat{h}^\star(x))] \\
&\leq \max_{h \in \mathcal{H}} \mathbb{E}_x[(f_2(h(x)) - f_2(\widehat{h}^\star(x)))^2] \\
&\leq 4 \max_{h \in \mathcal{H}} \mathbb{E}_x[(f_2(h(x)))^2] .
\end{aligned}$$

Hence, we are reduced to bounding the two second moments $\mathbb{E}_x[(f_1(h(x)))^2]$ and $\mathbb{E}_x[(f_2(h(x)))^2]$ for all $h \in \mathcal{H}$. Now, for log loss,

$$\mathbb{E}_x[(f_1(h(x)))^2] = \mathbb{E}_x \left[ \log^2 \frac{1}{1 - h(x)} \right] , \qquad \mathbb{E}_x[(f_2(h(x)))^2] = \mathbb{E}_x \left[ \log^2 \frac{1 - h(x)}{h(x)} \right] .$$

When $h(x) = \sigma(w_h(x))$, with $\sigma(a) = \frac{e^a}{1+e^a}$, we can also write

$$\log^2 \frac{1}{1 - h(x)} = \log^2(1 + e^{w_h(x)}) \leq 1 + (w_h(x))^2$$

using the fact that, for all $w \in \mathbb{R}$,

$$\log^2(1 + e^w) \leq 1 + w^2$$

On the other hand,

$$\log^2 \frac{1 - h(x)}{h(x)} = (w_h(x))^2 .$$

Hence

$$\mathbb{E}_x[(f_1(h(x)))^2] \leq 1 + \mathbb{E}_x[(w_h(x))^2] , \qquad \mathbb{E}_x[(f_2(h(x)))^2] \leq \mathbb{E}_x[(w_h(x))^2] ,$$

and the finitess of $\mathbb{E}_x[(w_h(x))^2]$ for all $h \in \mathcal{H}$ implies the finiteness of $\sigma_{f_1}^2(\mathcal{H})$, $\sigma_{f_2}^2(\mathcal{H})$, and $\mathbb{E}^2[\mathcal{H}]$.

## B.5. On Condition (6) for Log Loss

Recall that for log loss $f_1(a) = \log\frac{1}{1-a}$, and $f_2(a) = \log\frac{1-a}{a}$. With the representation $a = \sigma(w_a) = \frac{e^{w_a}}{1+e^{w_a}}$, and $b = \sigma(w_b) = \frac{e^{w_b}}{1+e^{w_b}}$, We have

$$(f_1(a) - f_1(b))^2 = \left(\log\frac{1+e^{w_a}}{1+e^{w_b}}\right)^2$$

and

$$(f_2(a) - f_2(b))^2 = (w_a - w_b)^2 .$$

We recall that

$$\bar{\ell}(a,b) - \bar{\ell}(b,b) = b\log\frac{b}{a} + (1-b)\log\frac{1-b}{1-a} = \mathrm{KL}(b,a)$$

Moreover, substituting the sigmoids for $a$ and $b$ we get

$$\mathrm{KL}(b,a) = \log\frac{1+e^{w_a}}{1+e^{w_b}} - \sigma(w_b)(w_a - w_b) ,$$

which is the Bregman divergence

$$d_\psi(w_a, w_b) = \psi(w_a) - \psi(w_b) - \psi'(w_b)(w_a - w_b) ,$$

with

$$\psi(w) = \log(1 + e^w) .$$

Now, it is well known that $\psi$ is strongly smooth, in that

$$d_\psi(w_a, w_b) \le (w_a - w_b)^2 = (f_2(a) - f_2(b))^2 ,$$

implying condition (6) with $C_2 = 2$. On the other hand, by Taylor theorem

$$\psi(w_a) = \psi(w_b) + \psi'(w_b)(w_a - w_b) + \frac{\psi''(\xi)}{2}(w_a - w_b)^2 ,$$

for some $\xi$ in the line connecting $w_a$ to $w_b$. Since $\psi''(\cdot) = \sigma'(\cdot)$, it follows that

$$d_\psi(w_a, w_b) = \frac{\sigma'(\xi)}{2}(w_a - w_b)^2 .$$

If the logits $w_a$ and $w_b$ have bounded range, then so is $\xi$, $\sigma'(\xi)$ will be lower bounded by a positive constant, and $c_2$ in (6) can be set to that constant.

As for $c_1$ and $C_1$, it is known that

$$(f_1(a) - f_1(b))^2 = \left(\log\frac{1+e^{w_a}}{1+e^{w_b}}\right)^2 \le (w_a - w_b)^2$$

for all $w_a, w_b \in \mathbb{R}$. So in order to prove that

$$d_\psi(w_a, w_b) \ge \frac{c_1}{2}(f_1(a) - f_1(b))^2$$

we need again to enforce a bound on the range of $w_a$ and $w_b$.

Finally, in order to prove

$$d_\psi(w_a, w_b) \le \frac{C_1}{2}(f_1(a) - f_1(b))^2 , \tag{26}$$

we observe that

$$f_1(a) - f_1(b) = \psi(w_a) - \psi(w_b) = \psi'(\xi')(w_a - w_b) ,$$

for some $\xi'$ (not necessarily equal to $\xi$) in the line connecting $w_a$ to $w_b$. Thus (26) becomes equivalent to

$$\frac{\sigma'(\xi)}{2}(w_a - w_b)^2 \le \frac{C_1}{2}(\psi'(\xi'))^2(w_a - w_b)^2 \,,$$

that is,

$$\frac{e^\xi}{(1 + e^\xi)^2} \le \frac{C_1}{2}\frac{e^{2\xi'}}{(1 + e^{\xi'})^2} \,.$$

Since $\frac{e^\xi}{(1+e^\xi)^2} \le 1/4$ for all $\xi \in \mathbb{R}$, a sufficient condition turns out to be

$$C_1 = \max_{\xi' \in \left(\min\{w_a, w_b\}, \max\{w_a, w_b\}\right)} \left(\frac{1 + e^{\xi'}}{e^{\xi'}}\right)^2 \,,$$

requiring again for $w_a$ and $w_b$ to have bounded range.

### B.6. Proof of Theorem 3.5

The next lemma is of preliminary importance, as it connects the variance of estimator $\widetilde{\Delta}\ell_b(h, \widehat{h}^\star; z)$ used in Algorithm 1 to the regret $Reg(h)$ of $h$.

**Lemma B.3.** *Let the loss $\ell(h(x), y) = f_1(h(x)) + y f_2(h(x))$ satisfy condition (6) in the main body of the paper. Let $\mathcal{L}(h) = \mathbb{E}_{x,y}[\ell(h(x), y)]$, and define, for any $h_1, h_2 \in \mathcal{H}$,*

$$\gamma_{f_i}(h_1, h_2) = \mathbb{E}_x[(f_i(h_1(x)) - f_i(h_2(x)))^2] \,,$$

*where $i = 1, 2$. Then*

$$\gamma_{f_i}(h, \widehat{h}^\star) \le \frac{4}{c_i}\left(\mathcal{L}(h) - \mathcal{L}(\widehat{h}^\star)\right) + 2\left(\frac{C_i}{c_i} + 1\right)\gamma_{f_i}(\widehat{h}^\star, h^\star) \,, \qquad i = 1, 2 \,,$$

*where $h^\star$ is the Bayes optimal predictor for $\ell$, that is $h^\star = \arg\min_{h : \mathcal{X} \to [0,1]} \mathcal{L}(h)$.*

*Proof.* For either $i = 1$ or $i = 2$, set for brevity $\rho_i = C_i/c_i \ge 1$. Then

$$\gamma_{f_i}(h, \widehat{h}^\star)$$
$$= \mathbb{E}_x\left[\left(f_i(h(x)) - f_i(h^\star(x)) + f_i(h^\star(x)) - f_i(\widehat{h}^\star(x))\right)^2\right]$$
$$= \mathbb{E}_x\left[\left(f_i(h(x)) - f_i(h^\star(x))\right)^2\right] - \rho_i \mathbb{E}_x\left[\left(f_i(h^\star(x)) - f_i(\widehat{h}^\star(x))\right)^2\right]$$
$$\quad + 2\mathbb{E}_x\left[\left(f_i(h^\star(x)) - f_i(\widehat{h}^\star(x))\right)\left(f_i(h(x)) - f_i(\widehat{h}^\star(x))\right)\right] + (\rho_i - 1)\mathbb{E}_x\left[(f_i(h^\star(x)) - f_i(\widehat{h}^\star(x)))^2\right]$$
$$(\text{ using } (a - b + b - c)^2 = (a - b)^2 - \rho(b - c)^2 + 2(b - c)(a - c) + (\rho - 1)(b - c)^2\,)$$
$$= \gamma_{f_i}(h, h^\star) - \rho_i\gamma_{f_i}(\widehat{h}^\star, h^\star) + 2\mathbb{E}_x\left[\left(f_i(h^\star(x)) - f_i(\widehat{h}^\star(x))\right)\left(f_i(h(x)) - f_i(\widehat{h}^\star(x))\right)\right]$$
$$\quad + (\rho_i - 1)\gamma_{f_i}(\widehat{h}^\star, h^\star)$$
$$\le \gamma_{f_i}(h, h^\star) - \rho_i\gamma_{f_i}(\widehat{h}^\star, h^\star) + 2\sqrt{\mathbb{E}_x\left[\left(f_i(h^\star(x)) - f_i(\widehat{h}^\star(x))\right)^2\right]}\sqrt{\mathbb{E}_x\left[\left(f_i(h(x)) - f_i(\widehat{h}^\star(x))\right)^2\right]}$$
$$\quad + (\rho_i - 1)\gamma_{f_i}(\widehat{h}^\star, h^\star)$$
$$(\text{from the Cauchy-Schwarz inequality})$$
$$= \gamma_{f_i}(h, h^\star) - \rho_i\gamma_{f_i}(\widehat{h}^\star, h^\star) + 2\sqrt{\gamma_{f_i}(h^\star, \widehat{h}^\star)}\sqrt{\gamma_{f_i}(h, \widehat{h}^\star)} + (\rho_i - 1)\gamma_{f_i}(\widehat{h}^\star, h^\star) \,. \tag{27}$$

From the linearity of $\ell$ w.r.t. $y$ we have, for all $h$,

$$\mathbb{E}_{y|x}[\ell(h(x), y) \mid x] = \bar{\ell}(h(x), h^\star(x)) \,.$$

Hence
$$\mathcal{L}(h) = \mathbb{E}_x[\bar{\ell}(h(x), h^\star(x))] .$$

Moreover, from Condition (6) we can write, for all $x$,

$$\bar{\ell}(h(x), h^\star(x)) - \bar{\ell}(\widehat{h}^\star(x), h^\star(x))$$
$$= \bar{\ell}(h(x), h^\star(x)) - \bar{\ell}(h^\star(x), h^\star(x)) - \left(\bar{\ell}(\widehat{h}^\star(x), h^\star(x)) - \bar{\ell}(h^\star(x), h^\star(x))\right)$$
$$\geq \frac{c_i}{2}\left(f_i(h(x)) - f_i(h^\star(x))\right)^2 - \frac{C_i}{2}\left(f_i(\widehat{h}^\star(x)) - f_i(h^\star(x))\right)^2 ,$$

so that, taking expectation w.r.t. $x$, and multiplying by $2/c_i$, we obtain

$$\frac{2}{c_i}\left(\mathcal{L}(h) - \mathcal{L}(\widehat{h}^\star)\right) \geq \gamma_{f_i}(h, h^\star) - \rho_i \gamma_{f_i}(\widehat{h}^\star, h^\star) .$$

We plug this back into (27), yielding

$$\gamma_{f_i}(h, \widehat{h}^\star) \leq \frac{2}{c}\left(\mathcal{L}(h) - \mathcal{L}(\widehat{h}^\star)\right) + 2\sqrt{\gamma_{f_i}(h^\star, \widehat{h}^\star)}\sqrt{\gamma_{f_i}(h, \widehat{h}^\star)} + (\rho_i - 1)\gamma_{f_i}(h^\star, \widehat{h}^\star) .$$

Solving for $\gamma_{f_i}(h, \widehat{h}^\star)$ (the above is a quadratic inequality in $\sqrt{\gamma_{f_i}(h, \widehat{h}^\star)}$), and setting for brevity $\widetilde{\Delta\mathcal{L}} = \frac{2}{c_i}\left(\mathcal{L}(h) - \mathcal{L}(\widehat{h}^\star)\right) + (\rho_i - 1)\gamma_{f_i}(h^\star, \widehat{h}^\star)$, we get

$$\gamma_{f_i}(h, \widehat{h}^\star) \leq \widetilde{\Delta\mathcal{L}} + 2\gamma_{f_i}(\widehat{h}^\star, h^\star) + 2\sqrt{\gamma^2_{f_i}(\widehat{h}^\star, h^\star) + \gamma_{f_i}(\widehat{h}^\star, h^\star)\widetilde{\Delta\mathcal{L}}} .$$

Using the inequality $\sqrt{a+b} \leq \sqrt{a} + b/(2\sqrt{a})$ (by Taylor expanding $\sqrt{x}$ around $a$), with $a = \gamma^2_{f_i}(\widehat{h}^\star, h^\star)$ we then have

$$\gamma_{f_i}(h, \widehat{h}^\star) \leq 4\gamma_{f_i}(\widehat{h}^\star, h^\star) + 2\widetilde{\Delta\mathcal{L}} = 2(\rho_i + 1)\gamma_{f_i}(\widehat{h}^\star, h^\star) + \frac{4}{c_i}\left(\mathcal{L}(h) - \mathcal{L}(\widehat{h}^\star)\right) ,$$

as claimed. $\qquad\square$

*Proof.* [of Theorem 3.5] We take a closer look at the proof of Theorem 3.2, in particular at the sample complexity guarantees in (21) and (22).

We first observe that
$$\sigma^2_{f_1}(h, \widehat{h}^\star) \leq \mathbb{E}_x[(f_1(h(x)) - f_1(\widehat{h}^\star(x)))^2] = \gamma_{f_1}(h, \widehat{h}^\star) ,$$

$$\sigma^2_{f_2}(h, \widehat{h}^\star) \leq \mathbb{E}_x[(f_2(h(x)) - f_2(\widehat{h}^\star(x)))^2] = \gamma_{f_2}(h, \widehat{h}^\star) ,$$

and

$$\mathbb{E}^2_x[\Delta f_2(h, \widehat{h}^\star; x)] \leq \mathbb{E}_x[(f_2(h(x)) - f_2(\widehat{h}^\star(x)))^2] = \gamma_{f_2}(h, \widehat{h}^\star) ,$$

where the $\gamma_{f_i}(h, \widehat{h}^\star)$ are the ones defined in Lemma B.3. We first plug these upper bounds in both (21) and (22), and then consider the resulting (21) and (22) separately.

The sample complexity condition (21) in the proof of Theorem 3.2 now reads

$$O\left(\sqrt{\gamma_{f_2}(h, \widehat{h}^\star)}\sqrt{\frac{\log(|\mathcal{H}|/\delta)}{m}} + \sqrt{\gamma_{f_1}(h, \widehat{h}^\star)}\sqrt{\frac{\log(|\mathcal{H}|/\delta)}{mk}} + \sqrt{\gamma_{f_2}(h, \widehat{h}^\star)}\sqrt{\frac{\log(1/\delta)}{mk}}\right)$$
$$\leq \frac{\Delta\mathcal{L}(h, \widehat{h}^\star)}{4} ,$$

which can be simplified to

$$
O\left(\sqrt{\gamma_{f_2}(h,\widehat{h}^\star)}\sqrt{\frac{\log(|\mathcal{H}|/\delta)}{m}} + \sqrt{\gamma_{f_1}(h,\widehat{h}^\star)}\sqrt{\frac{\log(|\mathcal{H}|/\delta)}{mk}}\right) \le \frac{\Delta\mathcal{L}(h,\widehat{h}^\star)}{4}.
$$

For this to hold it suffices to have

$$
m = \Omega\left(\left(\frac{\gamma_{f_2}(h,\widehat{h}^\star)}{(\Delta\mathcal{L}(h,\widehat{h}^\star))^2} + \frac{\gamma_{f_1}(h,\widehat{h}^\star)}{k(\Delta\mathcal{L}(h,\widehat{h}^\star))^2} + 1\right)\log(|\mathcal{H}|/\delta)\right),
$$

for all $h \in \mathcal{H}_{\beta/4}$, where the extra "+1" is meant to incorporate the extra condition $m = \Omega(\log(|\mathcal{H}|/\delta))$. Recall that $Reg(h) = \Delta\mathcal{L}(h,\widehat{h}^\star)$, so that $\mathcal{H}_{\beta/4} = \{h \in \mathcal{H} : Reg(h) \ge \beta/4\}$.

We now leverage the bound on $\gamma_{f_i}(h,\widehat{h}^\star)$ from Lemma B.3:

$$
\gamma_{f_i}(h,\widehat{h}^\star) = O\left(\frac{1}{c_i}Reg(h) + \left(\frac{C_i}{c_i} + 1\right)\gamma_{f_i}(\widehat{h}^\star,h^\star)\right), \qquad i = 1,2.
$$

This gives

$$
m = \Omega\left(\left(\frac{\frac{1}{c_2}Reg(h) + \left(\frac{C_2}{c_2} + 1\right)\gamma_{f_2}(\widehat{h}^\star,h^\star)}{(Reg(h))^2} + \frac{\frac{1}{c_1}Reg(h) + \left(\frac{C_1}{c_1} + 1\right)\gamma_{f_1}(\widehat{h}^\star,h^\star)}{k(Reg(h))^2} + 1\right)\log(|\mathcal{H}|/\delta)\right).
$$

Observe that the function $r \to \frac{r+A}{r^2}$ is decreasing in $r > 0$ for any $A \ge 0$. We use this observation with $r = Reg(h) \ge \beta/4$ to conclude that the resulting sample complexity guarantee delivered by (21) can be obtained by replacing in the above display $Reg(h)$ by $\beta/4$.

As for (22), we follow exactly the same route and obtain, for all $h \notin \mathcal{H}_{\beta/4}$,

$$
m = \Omega\left(\left(\frac{\frac{1}{c_2}Reg(h) + \left(\frac{C_2}{c_2} + 1\right)\gamma_{f_2}(\widehat{h}^\star,h^\star)}{\beta^2} + \frac{\frac{1}{c_1}Reg(h) + \left(\frac{C_1}{c_1} + 1\right)\gamma_{f_1}(\widehat{h}^\star,h^\star)}{k\beta^2} + 1\right)\log(|\mathcal{H}|/\delta)\right).
$$

However, since in this case $Reg(h) < \beta/4$, the same result as for (21) with $Reg(h)$ replaced by $\beta/4$ follows. This concludes the proof. $\qquad\square$

## C. The Multi-class Case

Suppose now we are in the multi-class case, so that $y \in \{0,\ldots c-1\}$, for some $c \ge 2$. We first deal with the full histogram multi-class setting, and then briefly cover the total multi-class setting.

### C.1. The Full Histogram Multi-Class Case

First, note that any loss function $\ell(h(x),y)$ for multi-class labels $y$ can always be written as

$$
\ell(h(x),y) = \sum_{r=0}^{c-1}\{y = r\}\,\ell(h(x),r).
$$

As in the binary case, we start off by introducing an ideal estimator for $\mathbb{E}_{(x,y)}[\ell(h(x),y)]$. The next is the *full histogram multi-class* counterpart to (2). For bag $z = ((x_1,\ldots,x_k),\boldsymbol{\alpha})$, with $\boldsymbol{\alpha} = (\alpha_0,\ldots,\alpha_{c-1})$, introduce the short-hands

$$
\mathbb{E}_r = \mathbb{E}_x[\ell(h(x),r)], \qquad p_r = \mathbb{P}(y = r),
$$

and consider the bag-level estimator

$$
\ell_b(h,z) = \sum_{r=0}^{c-1}(\alpha_r - p_r)\sum_{i=1}^{k}\Big(\ell(h(x_i),r) - \mathbb{E}_r\Big) + \sum_{r=0}^{c-1}p_r\mathbb{E}_r. \tag{28}
$$

The role of $f_1(h(x))$ and $f_2(h(x))$ in the binary case is now played by the $c$ functions $\{\ell(h(x),r)\}_{r=0}^{c-1}$. The following is the full histogram multi-class counterpart to Lemma 3.1. Its proof is given below in Section C.3.

**Lemma C.1.** *Let $(x_1, y_1), \ldots, (x_k, y_k) \in \mathcal{X} \times \{0, \ldots, c-1\}$ be drawn i.i.d. according to $\mathcal{D}$. Let $z = ((x_1, \ldots, x_k), \boldsymbol{\alpha})$, with $\boldsymbol{\alpha} = (\alpha_0, \ldots, \alpha_{c-1})$ be the corresponding bag. For any function $h \in \mathcal{H}$, and any loss $\ell(h(x), y) = \sum_{r=0}^{c-1} \{y = r\} \ell(h(x), r)$ we have*

$$\mathbb{E}_z[\ell_b(h, z)] = \mathbb{E}_{(x,y)}[\ell(h(x), y)], \qquad \mathrm{Var}_z(\ell_b(h, z)) \le 64 \, \mathbb{E}\left[\left(\max_r \ell(h(x), r)\right)^2\right].$$

Our LLP algorithm for the full histogram multi-class setting is the very same Algorithm 1 for binary labels, but applied to a proxy to estimator (28), where knowledge of $p_r$ and $\mathbb{E}_r$ are replaced by empirical averages. Moreover, as in the binary label case, we would like to estimate loss differences.

We thus define, for each $h_1, h_2 \in \mathcal{H}$,

$$
\begin{aligned}
&\Delta\ell_b(h_1, h_2; z) \\
&= \ell_b(h_1, z) - \ell_b(h_2, z) \\
&= \sum_{r=0}^{c-1} (\alpha_r - p_r) \sum_{i=1}^{k} \left(\Delta\ell_r(h_1, h_2; x_i) - \mathbb{E}_x[\Delta\ell_r(h_1, h_2; x)]\right) + \sum_{r=0}^{c-1} p_r \, \mathbb{E}_x[\Delta\ell_r(h_1, h_2; x)],
\end{aligned}
\tag{29}
$$

where

$$\Delta\ell_r(h_1, h_2; x) = \ell(h_1(x), r) - \ell(h_2(x), r).$$

Then we split the data as in (4), and use $S_1$ to estimate $\mathbb{E}_r$, $S_2$ to estimate $p_r$, for $r \in \{0, \ldots, c-1\}$, and $S_3$ to provide bags $z$ on which the LLP estimator (29) is constructed. We thus end up defining a proxy estimator $\widetilde{\ell}_b(h, z)$ and a MoM estimator $Q(h_1, h_2, ; S)$ similar to the one we constructed in Section 3.

In particular, in (29), we replace each $p_r$ by its empirical MoM counterpart on $S_2$, that is,

$$\widehat{p}_{S_2, r} = \widehat{\mu}_{\mathrm{MoM}}\left(\{\alpha_{m+1, r}, \ldots, \alpha_{2m, r}\}\right),$$

and each $\mathbb{E}_x[\Delta\ell_r(h_1, h_2; x)]$ by the estimator

$$\widehat{\mathbb{E}}_{S_1}[\Delta\ell_r(h_1, h_2)] = \widehat{\mu}_{\mathrm{MoM}}\left(\{v_{1,1,r}, \ldots, v_{1,k,r}, \ldots, v_{m,1,r}, \ldots, v_{m,k,r}\}\right),$$

where, for each $j \in [m]$ and $i \in [k]$, $v_{j,i,r} = \Delta\ell_r(h_1, h_2; x_{j,i})$, being $x_{j,i}$ the $i$-th feature vector in the $j$-th bag of $S_1$.

Then, for each bag $z_j \in S_3$ and $h \in \mathcal{H}$ define

$$
\begin{aligned}
&\widetilde{\Delta\ell}_b(h_1, h_2; z_j) \\
&= \sum_{r=0}^{c-1} (\alpha_r - \widehat{p}_{S_2, r}) \sum_{i=1}^{k} \left(\Delta\ell_r(h_1, h_2; x_{j,i}) - \widehat{\mathbb{E}}_{S_1}[\Delta\ell_r(h_1, h_2)]\right) + \sum_{r=0}^{c-1} \widehat{p}_{S_2, r} \, \widehat{\mathbb{E}}_{S_1}[\Delta\ell_r(h_1, h_2)].
\end{aligned}
\tag{30}
$$

Finally, we define a MoM estimator based on the bags of $S_3$ for the full histogram multi-class setting as follows. Consider the $m$ bags $z_j$ of $S_3 = \{z_{2m+1}, \ldots, z_{3m}\}$, and set

$$Q(h_1, h_2; S) = \mathrm{MoM}\left(\widetilde{\Delta\ell}_b(h_1, h_2; z_{2m+1}), \ldots, \widetilde{\Delta\ell}_b(h_1, h_2; z_{3m})\right).$$

The algorithm for the full histogram multi-class setting is Algorithm 1 with $Q(\cdot, \cdot; \cdot)$ defined above.

The following is the main theoretical guarantee for this algorithm. The proof is given in Section C.5.

**Theorem C.2.** *Let $S = (x_1, y_1), \ldots, (x_n, y_n)$ be drawn i.i.d. from a distribution $\mathcal{D}$ over $\mathcal{X} \times \{0, \ldots, c-1\}$. Let the loss $\ell(h(x), y) = \sum_{r=0}^{c-1} \{y = r\} \ell(h(x), r)$ be such that the maximal second moments $\max_{h \in \mathcal{H}} \mathbb{E}_x[\max_r(\ell(h(x), r))^2]$ are bounded. Let $\widehat{h}^\star = \min_{h \in \mathcal{H}} \mathcal{L}(h)$ be the best-in-class hypothesis. If $S$ is split into $3m$ random bags of size $k$, with $n = 3m \times k$ as in (1), then for all $\beta > 0$, the hypothesis $\widehat{h}$ output by Algorithm 1 with $Q(\cdot, \cdot; S)$ defined above satisfies $Reg(\widehat{h}) \le \beta$ with probability at least $1 - \delta$, provided*

$$n = O\left(\frac{(k \log(|\mathcal{H}|/\delta) + c \log(c/\delta)) \, \Delta\ell^2(\mathcal{H})}{\beta^2} + \frac{k \, \Sigma(\mathcal{H}) \log(c|\mathcal{H}|/\delta)}{\beta}\right),$$

*where*

$$\Delta\ell^2(\mathcal{H}) = \max_{h\in\mathcal{H}} \mathbb{E}_x\left[\max_r\left(\ell(h(x),r) - \ell(\widehat{h}^\star(x),r)\right)^2\right], \qquad \Sigma(\mathcal{H}) = \sqrt{\max_{h\in\mathcal{H}}\sum_{r=0}^{c-1}\sigma_r^2(h,\widehat{h}^\star)}.$$

When dealing with multi-class classification in LLP, there are four main quantities/dependencies that are worth considering: i. the regret bound $\beta$, ii. the bag size $k$, iii. the number of classes $c$, and iv. the complexity of the function space $\mathcal{H}$. In terms of $\beta$ and $k$, the bound in Theorem C.2 above is a slow rate of the form $k/\beta^2$ (we have been unable to prove for the full histogram case fast rates akin to Theorem 3.5). As for the dependence on $c$, we highlight an explicit linear dependence in the second term of the bound, and a $\sqrt{c}$ dependence on the third term via $\Sigma(\mathcal{H})$. Notice that both terms are lower order, compared to the first one, since the second term does not get multiplied by $k$ or by $\log|\mathcal{H}|$, and the third term has $\beta$ instead of $\beta^2$ in the denominator. A more implicit dependence on $c$ occurs in the first and second terms term via $\Delta\ell^2(\mathcal{H})$ (due to the presence of "$\max_r$"). This can be clearly removed altogether if the loss is bounded (like square loss). For the main term (that is, the first term), $\max_r$ within $\Delta\ell^2(\mathcal{H})$ can be turned into a mild (logarithmic) dependence on $c$ even when the loss is unbounded, provided we make light tail (e.g., sub-Gaussian) assumptions on the distribution of $\ell(h(x),r) - \ell(\widehat{h}^\star(x),r)$. The reader is referred to Appendix C.4 for details.

## C.2. The Total Multi-Class Case

We now move on to sketch the *total* multi-class case. In this case the estimator (28) is not legit, as it assumes direct access to all multi-class aggregate value $\alpha_0, \ldots, \alpha_{c-1}$. It turns out the total multi-class case we can easily tackle only when the loss function has the affine form $\ell(h(x),y) = f_1(h(x)) + yf_2(h(x))$, where now, $y \in \{0, \ldots, c-1\}$. Notable examples are the Poisson Log Loss $\ell(h(x),y) = h(x) - y\log h(x)$, as well as the square loss for regression $\ell(h(x),y) = (y-h(x))^2$, where the resulting $y^2$ term can be disregarded, since it is independent of the model's prediction $h(x)$. Note, however, that neither the multi-class log loss nor the multi-class square loss (aka Brier score) is an affine function of $y$. The key observation here is that the above affine form of the loss *coincides* with the general form of losses for binary labels we analyzed in Section 3, the only difference being that now the label $y$ has a wider range than binary. In this sense, the binary label case presented in Section 3 is simply a special case of the total multi-class case, the bag-level estimator (2) immediately extending to this multi-class scenario.

From a theoretical standpoint, the corresponding version of Lemma 3.1 will feature an extra $c^2$ factor in the variance bound, and so do as a consequence the variance/second moment terms occurring in Theorem 3.2, and Theorem 3.5. This is due to the need to replace $\eta(\cdot)$ in the proof of Lemma 3.1 with $\mathbb{E}[y|\cdot]$, whose range of values is as wide as $c$ instead of 1. The easy details of this adaptation are omitted.

## C.3. Proof of Lemma C.1

Introduce the short-hand

$$\ell_r(x) = \ell(h(x),r),$$

and consider the intermediate bag-level quantity

$$g(h,z) = \sum_{r=0}^{c-1}(\alpha_r - p_r)\frac{1}{k}\sum_{i=1}^{k}\left(\ell_r(x_i) - \mathbb{E}_r\right).$$

Recall that $\eta_r(x) = \mathbb{P}_{\mathcal{D}_{y|x}}(y = r|x)$ We want to relate the bag-level expectation of $g(h,z)$ to the population loss at the individual label level. We have

$$\mathbb{E}_{(x,y)}[\ell(h(x),y)] = \sum_{r=0}^{c-1}\mathbb{E}_x\mathbb{E}_{y|x}[\{y = r\}\ell_r(x)\,|\,x] = \sum_{r=0}^{c-1}\mathbb{E}_x[\eta_r(x)\ell_r(x)],$$

and

$$\mathbb{E}_z[g(h,z)] = \sum_{r=0}^{c-1} \mathbb{E}_z\Big[(\alpha_r - p_r)\frac{1}{k}\sum_{i=1}^{k}\Big(\ell_r(x_i) - \mathbb{E}_r\Big)\Big]$$

$$= \frac{1}{k^2}\sum_{r=0}^{c-1}\mathbb{E}_z\left[\sum_{j=1}^{k}\underbrace{\Big(\{y_j = r\} - p_r\Big)}_{\widetilde{y}_{j,r}}\sum_{i=1}^{k}\underbrace{\Big(\ell_r(x_i) - \mathbb{E}_r\Big)}_{\widetilde{\ell}_{i,r}}\right]$$

$$= \frac{1}{k^2}\sum_{r=0}^{c-1}\mathbb{E}_z\left[\sum_{j=i}\widetilde{y}_{j,r}\widetilde{\ell}_{i,r} + \sum_{j\neq i}\widetilde{y}_{j,r}\widetilde{\ell}_{i,r}\right] \; .$$

But when $j \neq i$ we have $\mathbb{E}[\widetilde{y}_{j,r}\widetilde{\ell}_{i,r}] = \mathbb{E}[\widetilde{y}_{j,r}]\,\mathbb{E}[\widetilde{\ell}_{i,r}] = 0$, so that the above simplifies as

$$\mathbb{E}_z[g(h,z)] = \frac{1}{k}\sum_{r=0}^{c-1}\mathbb{E}\Big[\widetilde{y}_{1,r}\widetilde{\ell}_{1,r}\Big] = \frac{1}{k}\sum_{r=0}^{c-1}\mathbb{E}\Big[\Big(\{y=r\} - p_r\Big)\Big(\ell_r(x) - \mathbb{E}_r\Big)\Big] \; .$$

In turn

$$\mathbb{E}\Big[\Big(\{y=r\} - p_r\Big)\Big(\ell_r(x) - \mathbb{E}_r\Big)\Big] = \mathbb{E}\Big[\Big(\{y=r\} - p_r\Big)\ell_r(x)\Big] - \underbrace{\mathbb{E}\Big[\Big(\{y=r\} - p_r\Big)\mathbb{E}_r\Big]}_{=0}$$

$$= \mathbb{E}_x[\eta_r(x)\ell_r(x)] - p_r\mathbb{E}_r \; .$$

Hence

$$\mathbb{E}_z[g(h,z)] = \frac{1}{k}\sum_{r=0}^{c-1}\Big(\mathbb{E}[\eta_r(x)\ell_r(x)] - p_r\mathbb{E}_r\Big) = \frac{1}{k}\,\mathbb{E}[\ell(h(x),y)] - \frac{1}{k}\sum_{r=0}^{c-1}p_r\mathbb{E}_r \; .$$

As a consequence,

$$\ell_b(h,z) = k\,g(h,z) + \sum_{r=0}^{c-1}p_r\mathbb{E}_r$$

$$= \sum_{r=0}^{c-1}(\alpha_r - p_r)\sum_{i=1}^{k}\Big(\ell_r(x_i) - \mathbb{E}_r\Big) + \sum_{r=0}^{c-1}p_r\mathbb{E}_r \tag{31}$$

is an unbiased estimator of $\mathbb{E}[\ell(h(x),y)]$. This proves the first part of the lemma.

Consider now the variance of $\ell_b(h,z)$. To facilitate the effort, introduce the short-hand notation

$$\widetilde{\alpha}_r = \alpha_r - p_r \; , \qquad \widetilde{\ell}_{r,i} = \ell_r(x_i) - \mathbb{E}_r \; .$$

For bag $z$, set $\mathcal{B} = (x_1,\ldots,x_k)$, and let us switch to a vector notation.
Define

$$\boldsymbol{\alpha} = [\alpha_0,\ldots,\alpha_{c-1}]^\top$$
$$\boldsymbol{p} = [p_0,\ldots,p_{c-1}]^\top$$
$$\boldsymbol{\ell}_i = [\ell_0(x_i),\ldots,\ell_{c-1}(x_i)]^\top$$
$$\mathbb{E} = [\mathbb{E}_0,\ldots,\mathbb{E}_{c-1}]^\top$$
$$\boldsymbol{\eta}_i = [\eta_0(x_i),\ldots,\eta_{c-1}(x_i)]^\top \; ,$$

and centered versions thereof

$$\widetilde{\boldsymbol{\alpha}} = \boldsymbol{\alpha} - \boldsymbol{p}$$
$$\widetilde{\boldsymbol{\ell}}_i = \boldsymbol{\ell}_i - \mathbb{E}$$
$$\widetilde{\boldsymbol{\eta}}_i = \boldsymbol{\eta}_i - \boldsymbol{p} \,.$$

Now,

$$
\begin{aligned}
\ell_b(h, z) &= \sum_{i=1}^{k} \sum_{r=0}^{c-1} (\alpha_r - p_r)(\ell_r(x_i) - \mathbb{E}_r) \\
&= \sum_{i=1}^{k} \widetilde{\boldsymbol{\alpha}}^{\top} \widetilde{\boldsymbol{\ell}}_i \\
&= \widetilde{\boldsymbol{\alpha}}^{\top} \Big( \sum_{i=1}^{k} \widetilde{\boldsymbol{\ell}}_i \Big) \,,
\end{aligned}
$$

so that

$$
\begin{aligned}
\mathrm{Var}(\ell_b(h, z)) &= \mathbb{E}\Big[ \Big( \widetilde{\boldsymbol{\alpha}}^{\top} \Big( \sum_{i=1}^{k} \widetilde{\boldsymbol{\ell}}_i \Big) \Big)^2 \Big] - \mathbb{E}^2 \Big[ \widetilde{\boldsymbol{\alpha}}^{\top} \Big( \sum_{i=1}^{k} \widetilde{\boldsymbol{\ell}}_i \Big) \Big] \\
&\leq \mathbb{E}\Big[ \Big( \sum_{i=1}^{k} \widetilde{\boldsymbol{\ell}}_i \Big)^{\top} \widetilde{\boldsymbol{\alpha}} \widetilde{\boldsymbol{\alpha}}^{\top} \Big( \sum_{i=1}^{k} \widetilde{\boldsymbol{\ell}}_i \Big) \Big] \\
&= \mathbb{E}\Big[ \Big( \sum_{i=1}^{k} \widetilde{\boldsymbol{\ell}}_i \Big)^{\top} \mathbb{E}[\widetilde{\boldsymbol{\alpha}} \widetilde{\boldsymbol{\alpha}}^{\top} \mid \mathcal{B}] \Big( \sum_{i=1}^{k} \widetilde{\boldsymbol{\ell}}_i \Big) \Big] \,.
\end{aligned}
$$

Further,

$$\widetilde{\boldsymbol{\alpha}} \widetilde{\boldsymbol{\alpha}}^{\top} = \boldsymbol{\alpha} \boldsymbol{\alpha}^{\top} - \boldsymbol{\alpha} \boldsymbol{p}^{\top} - \boldsymbol{p} \boldsymbol{\alpha}^{\top} + \boldsymbol{p} \boldsymbol{p}^{\top} \,,$$

and

$$\mathbb{E}[\boldsymbol{\alpha}] = \frac{1}{k} \sum_{i=1}^{k} \boldsymbol{\eta}_i^{*} \,.$$

Moreover, for $r \neq s$,

$$\alpha_r \alpha_s = \frac{1}{k^2} \Big( \sum_{i,j=1}^{k} \{y_i = r, y_j = s\} \Big) = \frac{1}{k^2} \Big( \sum_{i \neq j}^{k} \{y_i = r, y_j = s\} \Big)$$

so that

$$\mathbb{E}[\alpha_r \alpha_s \mid \mathcal{B}] = \frac{1}{k^2} \sum_{i \neq j}^{k} \eta_r(x_i) \eta_s(x_j) = \frac{1}{k^2} \Big[ \Big( \sum_{i=1}^{k} \eta_r(x_i) \Big) \Big( \sum_{j=1}^{k} \eta_s(x_j) \Big) - \sum_{i=1}^{k} \eta_r(x_i) \eta_s(x_i) \Big]$$

and

$$\mathbb{E}[\boldsymbol{\alpha} \boldsymbol{\alpha}^{\top} \mid \mathcal{B}] = \frac{1}{k^2} \Big( \boldsymbol{\sigma}^{*} (\boldsymbol{\sigma}^{*})^{\top} - \Sigma^{*} \Big)$$

and

$$\mathbb{E}[\widetilde{\boldsymbol{\alpha}} \widetilde{\boldsymbol{\alpha}}^{\top} \mid \mathcal{B}] = \frac{1}{k^2} \Big( \widetilde{\boldsymbol{\sigma}}^{*} (\widetilde{\boldsymbol{\sigma}}^{*})^{\top} - \Sigma^{*} \Big)$$

where

$$\boldsymbol{\sigma}^* = \sum_{i=1}^k \boldsymbol{\eta}_i \,, \qquad \widetilde{\boldsymbol{\sigma}}^* = \sum_{i=1}^k \widetilde{\boldsymbol{\eta}}_i$$

$$\Sigma^* = \left[ \sum_{i=1}^k \eta_r(x_i)\eta_s(x_i) \right]_{r,s=0}^{(c-1)\times(c-1)} = \sum_{i=1}^k \boldsymbol{\eta}_i \boldsymbol{\eta}_i^\top$$

is positive semi-definite. As a consequence,

$$\operatorname{Var}(\ell_b(h,z)) \le \frac{1}{k^2} \, \mathbb{E}\left[ \left( \sum_{i=1}^k \widetilde{\boldsymbol{\ell}}_i \right)^\top \boldsymbol{\sigma}^*(\boldsymbol{\sigma}^*)^\top \left( \sum_{i=1}^k \widetilde{\boldsymbol{\ell}}_i \right) \right] - \frac{1}{k^2} \, \mathbb{E}\left[ \left( \sum_{i=1}^k \widetilde{\boldsymbol{\ell}}_i \right)^\top \Sigma^* \left( \sum_{i=1}^k \widetilde{\boldsymbol{\ell}}_i \right) \right]$$

$$= \frac{1}{k^2} \, \mathbb{E}\left[ \left( \sum_{i=1}^k \widetilde{\boldsymbol{\ell}}_i \right)^\top \boldsymbol{\sigma}^*(\boldsymbol{\sigma}^*)^\top \left( \sum_{i=1}^k \widetilde{\boldsymbol{\ell}}_i \right) - \sum_{j=1}^k \left( \sum_{i=1}^k \widetilde{\boldsymbol{\ell}}_i \right)^\top \boldsymbol{\eta}_j(\boldsymbol{\eta}_j)^\top \left( \sum_{i=1}^k \widetilde{\boldsymbol{\ell}}_i \right) \right]$$

$$= \frac{1}{k^2} \, \mathbb{E}\left[ \left( \left( \sum_{i=1}^k \widetilde{\boldsymbol{\ell}}_i \right)^\top \widetilde{\boldsymbol{\sigma}}^* \right)^2 - \sum_{j=1}^k \left( \left( \sum_{i=1}^k \widetilde{\boldsymbol{\ell}}_i \right)^\top \boldsymbol{\eta}_j \right)^2 \right].$$

But

$$\left( \sum_{i=1}^k \widetilde{\boldsymbol{\ell}}_i \right)^\top \widetilde{\boldsymbol{\sigma}}^* = \left( \sum_{i=1}^k \widetilde{\boldsymbol{\ell}}_i \right)^\top \left( \sum_{j=1}^k \widetilde{\boldsymbol{\eta}}_j \right) = \sum_{i=1}^k \widetilde{\boldsymbol{\ell}}_i^\top \widetilde{\boldsymbol{\eta}}_i + \sum_{i\neq j} \widetilde{\boldsymbol{\ell}}_i^\top \widetilde{\boldsymbol{\eta}}_j$$

and, because of the fact that $\mathbb{E}[\widetilde{\boldsymbol{\eta}}_i] = \mathbb{E}[\widetilde{\boldsymbol{\ell}}_i] = 0$, and the variables $\widetilde{\boldsymbol{\ell}}_i$ and $\widetilde{\boldsymbol{\eta}}_j$ are independent for $i \neq j$, we also have

$$\mathbb{E}\left[ \left( \sum_{i=1}^k \widetilde{\boldsymbol{\ell}}_i^\top \widetilde{\boldsymbol{\eta}}_i + \sum_{i\neq j} \widetilde{\boldsymbol{\ell}}_i^\top \widetilde{\boldsymbol{\eta}}_j \right)^2 \right] = k \, \mathbb{E}\left[ \left( \widetilde{\boldsymbol{\ell}}_1^\top \widetilde{\boldsymbol{\eta}}_1 \right)^2 \right] + k(k-1) \, \mathbb{E}\left[ \left( \widetilde{\boldsymbol{\ell}}_1^\top \widetilde{\boldsymbol{\eta}}_2 \right)^2 \right] + 2k(k-1) \, \mathbb{E}\left[ \left( \widetilde{\boldsymbol{\ell}}_1^\top \widetilde{\boldsymbol{\eta}}_2 \right) \left( \widetilde{\boldsymbol{\ell}}_2^\top \widetilde{\boldsymbol{\eta}}_1 \right) \right].$$

For similar reasons,

$$\mathbb{E}\left[ \sum_{j=1}^k \left( \left( \sum_{i=1}^k \widetilde{\boldsymbol{\ell}}_i \right)^\top \boldsymbol{\eta}_j \right)^2 \right] = k \, \mathbb{E}\left[ \left( \widetilde{\boldsymbol{\ell}}_1^\top \boldsymbol{\eta}_1 \right)^2 \right] + k(k-1) \, \mathbb{E}\left[ \left( \widetilde{\boldsymbol{\ell}}_1^\top \boldsymbol{\eta}_2 \right)^2 \right].$$

Hence we conclude that

$$\operatorname{Var}(\ell_b(h,z)) = \frac{1}{k} \, \mathbb{E}\left[ \left( \widetilde{\boldsymbol{\ell}}_1^\top \widetilde{\boldsymbol{\eta}}_1 \right)^2 - \left( \widetilde{\boldsymbol{\ell}}_1^\top \boldsymbol{\eta}_1 \right)^2 \right] + \frac{k-1}{k} \, \mathbb{E}\left[ \left( \widetilde{\boldsymbol{\ell}}_1^\top \widetilde{\boldsymbol{\eta}}_2 \right)^2 - \left[ \left( \widetilde{\boldsymbol{\ell}}_1^\top \boldsymbol{\eta}_2 \right)^2 \right] \right.$$

$$+ \frac{2(k-1)}{k} \, \mathbb{E}\left[ \left( \widetilde{\boldsymbol{\ell}}_1^\top \widetilde{\boldsymbol{\eta}}_2 \right) \left( \widetilde{\boldsymbol{\ell}}_2^\top \widetilde{\boldsymbol{\eta}}_1 \right) \right]$$

$$\le \frac{1}{k} \, \mathbb{E}\left[ \left( \widetilde{\boldsymbol{\ell}}_1^\top \widetilde{\boldsymbol{\eta}}_1 \right)^2 \right] + \frac{k-1}{k} \, \mathbb{E}\left[ \left( \widetilde{\boldsymbol{\ell}}_1^\top \widetilde{\boldsymbol{\eta}}_2 \right)^2 \right] + \frac{2(k-1)}{k} \, \mathbb{E}\left[ \left( \widetilde{\boldsymbol{\ell}}_1^\top \widetilde{\boldsymbol{\eta}}_2 \right) \left( \widetilde{\boldsymbol{\ell}}_2^\top \widetilde{\boldsymbol{\eta}}_1 \right) \right].$$

Also, note that

$$\mathbb{E}\left[ \left( \widetilde{\boldsymbol{\ell}}_1^\top \widetilde{\boldsymbol{\eta}}_2 \right) \left( \widetilde{\boldsymbol{\ell}}_2^\top \widetilde{\boldsymbol{\eta}}_1 \right) \right] \le \sqrt{ \mathbb{E}\left[ \left( \widetilde{\boldsymbol{\ell}}_1^\top \widetilde{\boldsymbol{\eta}}_2 \right)^2 \right] \mathbb{E}\left[ \left( \widetilde{\boldsymbol{\ell}}_2^\top \widetilde{\boldsymbol{\eta}}_1 \right)^2 \right] } = \mathbb{E}\left[ \left( \widetilde{\boldsymbol{\ell}}_1^\top \widetilde{\boldsymbol{\eta}}_2 \right)^2 \right]$$

since $\mathbb{E}\left[ \left( \widetilde{\boldsymbol{\ell}}_1^\top \widetilde{\boldsymbol{\eta}}_2 \right)^2 \right] = \mathbb{E}\left[ \left( \widetilde{\boldsymbol{\ell}}_2^\top \widetilde{\boldsymbol{\eta}}_1 \right)^2 \right]$. Plugging back yields

$$\operatorname{Var}(\ell_b(h,z)) \le \frac{1}{k} \, \mathbb{E}\left[ \left( \widetilde{\boldsymbol{\ell}}_1^\top \widetilde{\boldsymbol{\eta}}_1 \right)^2 \right] + \frac{3(k-1)}{k} \, \mathbb{E}\left[ \left( \widetilde{\boldsymbol{\ell}}_1^\top \widetilde{\boldsymbol{\eta}}_2 \right)^2 \right]. \tag{32}$$

Consider now the first expectation in (32). We have

$$
\begin{aligned}
\left(\widetilde{\boldsymbol{\ell}}_1^\top \widetilde{\boldsymbol{\eta}}_1\right)^2 &= \left((\boldsymbol{\ell}_1 - \mathbb{E})^\top (\boldsymbol{\eta}_1 - \boldsymbol{p})\right)^2 \\
&= \left((\boldsymbol{\ell}_1 - \mathbb{E})^\top \boldsymbol{\eta}_1 - (\boldsymbol{\ell}_1 - \mathbb{E})^\top \boldsymbol{p}\right)^2 \\
&\leq 2\left((\boldsymbol{\ell}_1 - \mathbb{E})^\top \boldsymbol{\eta}_1\right)^2 + 2\left((\boldsymbol{\ell}_1 - \mathbb{E})^\top \boldsymbol{p}\right)^2 \\
&\leq 2\left(||\boldsymbol{\ell}_1 - \mathbb{E}||_\infty ||\boldsymbol{\eta}_1||_1\right)^2 + 2\left(||\boldsymbol{\ell}_1 - \mathbb{E}||_\infty ||\boldsymbol{p}||_1\right)^2 \\
&= 4||\boldsymbol{\ell}_1 - \mathbb{E}||_\infty^2
\end{aligned}
$$

the last equality due to the fact that $\boldsymbol{\eta}_1$ and $\boldsymbol{p}$ are probability vectors. For the second expectation, the argument is exactly the same, just replace $\boldsymbol{\eta}_1$ with $\boldsymbol{\eta}_2$.

Plugging back into (32) and overapproximating yields the handy upper bound

$$
\mathrm{Var}_z(\ell_b(h, z)) \leq 16\mathbb{E}_x\left[||\boldsymbol{\ell}_1 - \mathbb{E}||_\infty^2\right] . \tag{33}
$$

We can also eliminate the centering by observing the following:

$$
\begin{aligned}
\mathbb{E}_x\left[||\boldsymbol{\ell}_1 - \mathbb{E}||_\infty^2\right] &\leq \mathbb{E}_x\left[\left(||\boldsymbol{\ell}_1||_\infty + ||\mathbb{E}||_\infty\right)^2\right] \\
&\leq 2\mathbb{E}_x\left[||\boldsymbol{\ell}_1||_\infty^2\right] + 2\mathbb{E}_x\left[||\mathbb{E}||_\infty^2\right] \\
&= 2\mathbb{E}_x\left[||\boldsymbol{\ell}_1||_\infty^2\right] + 2||\mathbb{E}||_\infty^2 \\
&\leq 2\mathbb{E}_x\left[||\boldsymbol{\ell}_1||_\infty^2\right] + 2\mathbb{E}_x\left[||\boldsymbol{\ell}_1||_\infty^2\right] \\
&\quad \text{(from the definition of vector } \mathbb{E} = \mathbb{E}_x[\boldsymbol{\ell}_1], \text{ and the convexity of } || \cdot ||_\infty^2) \\
&= 4\mathbb{E}_x\left[||\boldsymbol{\ell}_1||_\infty^2\right] .
\end{aligned}
$$

This results in

$$
\mathrm{Var}_z(\ell_b(h, z)) \leq 64\mathbb{E}_x\left[||\boldsymbol{\ell}_1||_\infty^2\right] = 64\,\mathbb{E}_x\left[\left(\max_{r\in\{0,\dots,c-1\}} \ell(h(x), r)\right)^2\right] ,
$$

as claimed.

### C.4. Sub-Gaussian Losses for the Full Histogram Multi-Class Case

We can alternatively start from (33) and maintain centered variables. Now, if the loss components $\ell(h(x), r)$ are bounded by $L$ for all $(x, y)$, and $r \in \{0, \dots, c-1\}$, then clearly,

$$
\mathrm{Var}(\ell_b(h, z)) \leq 64L^2
$$

independent of $c$. On the other hand, if the loss is not uniformly bounded, but is sub-Gaussian, then observe that $\boldsymbol{\ell}_1 - \mathbb{E}$ is a zero-mean vector, and we can use standard inequalities for expectations of maxima of sub-Gaussian variables. For instance, the following claim comes in handy.

**Proposition C.3.** *Let $X_0, \dots, X_{c-1}$ be $\sigma^2$-sub-Gaussian random variable with zero mean (but not necessarily independent). Then, for all $t > 0$,*

$$
\mathbb{P}\left(\left(\max_{r=0,\dots,c-1} X_r\right)^2 \geq 2\sigma^2(\log c + t)\right) \leq 2e^{-t} .
$$

*Proof.* Let $M = \max_{r=0,\dots,c-1} X_r$, and $v = \sqrt{2\sigma^2(\log c + t)}$. Then

$$\mathbb{P}\left(\left(\max_{r=0,\dots,c-1} X_r\right)^2 \geq 2\sigma^2(\log c + t)\right) = \mathbb{P}(|M| \geq v)$$
$$\leq \mathbb{P}(\exists r \,:\, |X_r| \geq v)$$
$$\leq \sum_{r=0}^{c-1} \mathbb{P}(|X_r| \geq v)$$
$$\leq 2c\, e^{-\frac{v^2}{2\sigma^2}} = 2e^{-t} \,,$$

thus concluding the proof. $\qquad\square$

Now, when $M = ||\boldsymbol{\ell}_1 - \mathbb{E}||_\infty = \max_{r=0,\dots,c-1} |\ell(h(x_1), r) - \mathbb{E}_r|$, with $\mathbb{E}_r = \mathbb{E}_x[\ell(h(x), r)]$, and $\epsilon = 2\sigma^2 \log c$, we have

$$\mathbb{E}[M^2] = \int_0^\infty \mathbb{P}(M^2 > u)\, du$$
$$= \int_0^\epsilon \mathbb{P}(M^2 > u)\, du + \int_\epsilon^\infty \mathbb{P}(M^2 > u)\, du$$
$$\leq \epsilon + 2\sigma^2 \int_0^\infty \mathbb{P}(M^2 > \epsilon + 2\sigma^2 t)\, dt$$
$$\leq \epsilon + 2\sigma^2 \int_0^\infty 2e^{-t}\, dt = \epsilon + 4\sigma^2 = 2\sigma^2(\log c + 2) \,.$$

Thus if $\ell(h(x_1), r) - \mathbb{E}_r$ are $\sigma^2$-subgaussian components of vector $\boldsymbol{\ell}_1 - \mathbb{E}$, then

$$\mathrm{Var}(\ell_b(h, z)) \leq 12\left(2\sigma^2(\log c + 2)\right),$$

hence, with a logarithmic dependence on $c$.

### C.5. Proof of Theorem C.2

We follow to the extent possible the proof of Theorem 3.2. Yet, extra care has to be taken here when trying to make the dependence on the number of classes $c$ as small as possible.

Recall the definition of $\widetilde{\Delta}\ell_b(h_1, h_2; z)$ in (30). Let $\mathcal{L}(h)$ denote the population loss of model $h$ when the underlying loss is the multi-class loss $\ell$,
$$\mathcal{L}(h) = \mathbb{E}_{(x,y)}[\ell(h(x), y)] \,,$$

and set
$$\Delta\mathcal{L}(h_1, h_2) = \mathcal{L}(h_1) - \mathcal{L}(h_2) \,.$$

Then, Theorem 2 in (Lugosi & Mendelson, 2019) shows that if $m \geq r = \lceil 8\log(1/\delta) \rceil$, we have

$$\mathbb{P}\left(\forall h_1, h_2 \in \mathcal{H} \times \mathcal{H} \; \left| Q(h_1, h_2; S) - \mathbb{E}_z[\widetilde{\Delta}\ell_b(h_1, h_2; z) \,|\, S_1, S_2] \right| \right.$$
$$\left. \leq \sigma(h_1, h_2)\sqrt{\frac{32\log(|\mathcal{H}|^2/\delta)}{m}} \,\right| S_1, S_2\right)$$
$$\geq 1 - \delta \,, \tag{34}$$

where $\sigma^2(h_1, h_2) = \mathrm{Var}_z(\widetilde{\Delta}\ell_b(h_1, h_2; z) \,|\, S_1, S_2)$, being $\widetilde{\Delta}\ell_b(h_1, h_2; z)$ now defined as in (30), and the probability above is taken over $S_3$, conditioned on $S_1$ and $S_2$.

Similar to the binary case, we assume the stronger condition $m = \Omega(\log(c|\mathcal{H}|/\delta))$.

From Theorem 2 in (Lugosi & Mendelson, 2019) one can see that,

$$\mathbb{P}\left( \forall r \, \forall h_1, h_2 \in \mathcal{H} \times \mathcal{H} \; \left| \widehat{\mathbb{E}}_{S_1}[\Delta \ell_r(h_1, h_2)] - \mathbb{E}_x[\Delta \ell_r(h_1, h_2, x)] \right| \le \sigma_r(h_1, h_2) \sqrt{\frac{32 \log(c \, |\mathcal{H}|^2 / \delta)}{mk}} \right) \ge 1 - \delta \quad (35)$$

with $\sigma_r^2(h_1, h_2) = \mathrm{Var}_x(\Delta \ell_r(h_1, h_2; x))$, the probabilities being taken w.r.t. $S_1$.

Moreover, since $\mathrm{Var}(\alpha_{j,r}) = \frac{\mathrm{Var}(\{y=r\})}{k} \le \frac{p_r}{k}$, the same theorem shows that

$$\mathbb{P}\left( \forall r \, \left| \widehat{p}_{S_2, r} - p_r \right| \le \sqrt{\frac{32 p_r \log(c / \delta)}{mk}} \right) \ge 1 - \delta \,, \quad (36)$$

the probability being over $S_2$.

Set for brevity

$$A_{\delta, r}(h_1, h_2) = \sigma_r(h_1, h_2) \sqrt{\frac{32 \log(c \, |\mathcal{H}|^2 / \delta)}{mk}} \,,$$

$$B_{\delta, r}(h_1, h_2) = \sqrt{\frac{32 p_r \log(c / \delta)}{mk}} \,.$$

From Lemma C.1 we can write

$$\left| \mathbb{E}_z[\widetilde{\Delta} \ell_b(h_1, h_2; z) \,|\, S_1, S_2] - \Delta \mathcal{L}(h_1, h_2) \right| = \left| \mathbb{E}_z[\widetilde{\Delta} \ell_b(h_1, h_2; z) \,|\, S_1, S_2] - \mathbb{E}_z[\Delta \ell_b(h_1, h_2; z)] \right| \,.$$

As in the binary case, we need to upper bound this difference, and we compute the two expectations separately.

For fixed $h_1, h_2$, and $r$, set for brevity

$$\Delta \mathbb{E}_r = \mathbb{E}_x[\Delta \ell_r(h_1, h_2; x)] \,,$$

$$\widehat{\Delta} \mathbb{E}_r = \widehat{\mathbb{E}}_{S_1}[\Delta \ell_r(h_1, h_2)] \,,$$

$$\widehat{p}_r = \widehat{p}_{S_2, r} \,,$$

$$\Sigma_r = \sum_{i=1}^{k} \Delta \ell_r(h_1, h_2; x_i) \,.$$

Then

$$\mathbb{E}_z[\widetilde{\Delta} \ell_b(h_1, h_2; z) \,|\, S_1, S_2] = \sum_{r=0}^{c-1} \mathbb{E}_z\left[ (\alpha_r - \widehat{p}_r)(\Sigma_r - k \widehat{\Delta} \mathbb{E}_r) \right] + \sum_{r=0}^{c-1} \widehat{p}_r \, \widehat{\Delta} \mathbb{E}_r$$

$$= \sum_{r=0}^{c-1} \left( \mathbb{E}_z[\alpha_r \, \Sigma_r] - k \, \widehat{\Delta} \mathbb{E}_r \, p_r - k \, \widehat{p}_r \, \Delta \mathbb{E}_r + k \, \widehat{p}_r \, \widehat{\Delta} \mathbb{E}_r \right) + \sum_{r=0}^{c-1} \widehat{p}_r \, \widehat{\Delta} \mathbb{E}_r \,.$$

But

$$\mathbb{E}_z[\alpha_r \, \Sigma_r] = \frac{1}{k} \mathbb{E}\left[ \left( \sum_{i=1}^{k} \{y_i = r\} \right) \left( \sum_{i=1}^{k} \Delta \ell_r(h_1, h_2; x_i) \right) \right]$$

$$= \mathbb{E}_{(x,y)}[\{y = r\} \Delta \ell_r(h_1, h_2; x)] + (k - 1) p_r \Delta \mathbb{E}_r \,,$$

so that

$$\mathbb{E}_z[\widetilde{\Delta}\ell_b(h_1, h_2; z) \mid S_1, S_2]$$

$$= \sum_{r=0}^{c-1} \Big( \mathbb{E}_{(x,y)}[\{y = r\}\Delta\ell_r(h_1, h_2; x)] + (k-1)p_r\Delta\mathbb{E}_r - k\,\widehat{\Delta}\mathbb{E}_r\,p_r - k\,\widehat{p}_r\,\Delta\mathbb{E}_r + k\,\widehat{p}_r\,\widehat{\Delta}\mathbb{E}_r \Big)$$

$$+ \sum_{r=0}^{c-1} \widehat{p}_r\,\widehat{\Delta}\mathbb{E}_r \, .$$

Similarly,

$$\mathbb{E}_z[\Delta\ell_b(h_1, h_2; z)]$$

$$= \sum_{r=0}^{c-1} \Big( \mathbb{E}_{(x,y)}[\{y = r\}\Delta\ell_r(h_1, h_2; x)] + (k-1)p_r\Delta\mathbb{E}_r - k\,\Delta\mathbb{E}_r\,p_r - k\,p_r\,\Delta\mathbb{E}_r + k\,p_r\,\Delta\mathbb{E}_r \Big)$$

$$+ \sum_{r=0}^{c-1} p_r\,\Delta\mathbb{E}_r \, .$$

Hence

$$\left| \mathbb{E}_z[\widetilde{\Delta}\ell_b(h_1, h_2; z) \mid S_1, S_2] - \mathbb{E}_z[\Delta\ell_b(h_1, h_2; z)] \right|$$

$$= \left| k \sum_{r=0}^{c-1} \Big( (\Delta\mathbb{E}_r\,p_r - \widehat{\Delta}\mathbb{E}_r\,p_r) + (\Delta\mathbb{E}_r\,p_r - \Delta\mathbb{E}_r\,\widehat{p}_r) + (\widehat{p}_r\,\widehat{\Delta}\mathbb{E}_r - p_r\,\Delta\mathbb{E}_r) \Big) + \sum_{r=0}^{c-1} \Big( \widehat{p}_r\,\widehat{\Delta}\mathbb{E}_r - p_r\,\Delta\mathbb{E}_r \Big) \right|$$

$$= \left| k \sum_{r=0}^{c-1} (\Delta\mathbb{E}_r - \widehat{\Delta}\mathbb{E}_r)\,(p_r - \widehat{p}_r) + \sum_{r=0}^{c-1} \Big( \widehat{p}_r\,\widehat{\Delta}\mathbb{E}_r - p_r\,\Delta\mathbb{E}_r \Big) \right|$$

$$\leq k \sum_{r=0}^{c-1} \left| \Delta\mathbb{E}_r - \widehat{\Delta}\mathbb{E}_r \right| \left| p_r - \widehat{p}_r \right| + \sum_{r=0}^{c-1} \left| \widehat{p}_r\,\widehat{\Delta}\mathbb{E}_r - p_r\,\Delta\mathbb{E}_r \right|$$

$$= k \sum_{r=0}^{c-1} \left| \Delta\mathbb{E}_r - \widehat{\Delta}\mathbb{E}_r \right| \left| p_r - \widehat{p}_r \right| + \sum_{r=0}^{c-1} \left| \widehat{p}_r\,\widehat{\Delta}\mathbb{E}_r - p_r\,\widehat{\Delta}\mathbb{E}_r + p_r\,\widehat{\Delta}\mathbb{E}_r - p_r\,\Delta\mathbb{E}_r \right|$$

$$\leq k \sum_{r=0}^{c-1} \left| \Delta\mathbb{E}_r - \widehat{\Delta}\mathbb{E}_r \right| \left| p_r - \widehat{p}_r \right| + \sum_{r=0}^{c-1} \Big( p_r \left| \widehat{\Delta}\mathbb{E}_r - \Delta\mathbb{E}_r \right| + |\widehat{\Delta}\mathbb{E}_r| \left| \widehat{p}_r - p_r \right| \Big) \, .$$

We use (35) to bound $\left| \widehat{\Delta}\mathbb{E}_r - \Delta\mathbb{E}_r \right|$ and (36) to bound $\left| \widehat{p}_r - p_r \right|$. This gives, with probability at least $1 - 3\delta$ over the generation of $S_1$ and $S_2$, uniformly over $h_1, h_2$,

$$\left| \mathbb{E}_z[\widetilde{\Delta}\ell_b(h_1, h_2; z) \mid S_1, S_2] - \mathbb{E}_z[\Delta\ell_b(h_1, h_2; z)] \right|$$

$$\leq k \sum_{r=0}^{c-1} A_{\delta,r}(h_1, h_2)\,B_{\delta,r}(h_1, h_2) + \sum_{r=0}^{c-1} \Big( \widehat{p}_r\,A_{\delta,r}(h_1, h_2) + |\Delta\mathbb{E}_r|\,B_{\delta,r}(h_1, h_2) \Big)$$

$$= O\left( \sqrt{\frac{\log(c|\mathcal{H}|/\delta)}{m}} \sqrt{\frac{\log(c/\delta)}{m}} \sum_{r=0}^{c-1} \sigma_r(h_1, h_2)\sqrt{p_r} \right)$$

$$+ O\left( \sqrt{\frac{\log(c|\mathcal{H}|/\delta)}{mk}} \sum_{r=0}^{c-1} p_r\,\sigma_r(h_1, h_2) \right) + O\left( \sqrt{\frac{\log(c/\delta)}{mk}} \sum_{r=0}^{c-1} |\widehat{\Delta}\mathbb{E}_r|\sqrt{p_r} \right) \, .$$

We apply the upper bound

$$\sqrt{\frac{\log(c|\mathcal{H}|/\delta)}{m}} \sqrt{\frac{\log(c/\delta)}{m}} \leq \frac{\log(c|\mathcal{H}|/\delta)}{m} \, ,$$

and then focus on the three sums "$\sum_{r=0}^{c-1}$" above. We have

$$\sum_{r=0}^{c-1} \sigma_r(h_1, h_2)\sqrt{p_r} \leq \max_r \sigma_r(h_1, h_2) \left( \sum_{r=0}^{c-1} \sqrt{p_r} \right)$$

$$\leq \sqrt{c} \max_r \sigma_r(h_1, h_2) \, ,$$

$$\sum_{r=0}^{c-1} p_r \, \sigma_r(h_1, h_2) \leq \max_r \sigma_r(h_1, h_2) \left( \sum_{r=0}^{c-1} p_r \right)$$

$$= \max_r \sigma_r(h_1, h_2) \, ,$$

and similarly

$$\sum_{r=0}^{c-1} |\widehat{\Delta \mathbb{E}}_r| \sqrt{p_r} \leq \sqrt{c} \max_r |\widehat{\Delta \mathbb{E}}_r| \, .$$

We have therefore obtained

$$\left| \mathbb{E}_z[\widetilde{\Delta}\ell_b(h_1, h_2; z) \,|\, S_1, S_2] - \mathbb{E}_z[\Delta \ell_b(h_1, h_2; z)] \right|$$

$$= O\left( \left( \frac{\sqrt{c} \, \log(c|\mathcal{H}|/\delta)}{m} + \sqrt{\frac{\log(c|\mathcal{H}|/\delta)}{mk}} \right) \sigma(h_1, h_2) \right) + O\left( \sqrt{\frac{c \, \log(c/\delta)}{mk}} \max_r |\widehat{\Delta \mathbb{E}}_r| \right) ,$$

where $\sigma(h_1, h_2) = \max_r \sigma_r(h_1, h_2)$. But clearly, from (35)

$$|\widehat{\Delta \mathbb{E}}_r| \leq |\Delta \mathbb{E}_r| + A_{\delta, r}(h_1, h_2) = |\Delta \mathbb{E}_r| + O\left( \sigma_r(h_1, h_2) \sqrt{\frac{\log(c \, |\mathcal{H}|/\delta)}{mk}} \right) ,$$

which we replace back into the previous display. This results in

$$\left| \mathbb{E}_z[\widetilde{\Delta}\ell_b(h_1, h_2; z) \,|\, S_1, S_2] - \mathbb{E}_z[\Delta \ell_b(h_1, h_2; z)] \right|$$

$$= O\left( \left( \frac{\sqrt{c} \, \log(c|\mathcal{H}|/\delta)}{m} + \sqrt{\frac{\log(c|\mathcal{H}|/\delta)}{mk}} \right) \sigma(h_1, h_2) \right) + O\left( \sqrt{\frac{c \, \log(c/\delta)}{mk}} \max_r |\Delta \mathbb{E}_r| \right) . \tag{37}$$

As for $\mathrm{Var}_z(\widetilde{\Delta}\ell_b(h_1, h_2; z) \,|\, S_1, S_2)$, we again observe that in the conditional space where $S_1$ and $S_2$ are given, the bound on this variance can be obtained by adapting the argument contained in Lemma C.1, as specified next.

Let $z = (\mathcal{B}, \boldsymbol{\alpha})$, with $\mathcal{B} = (x_1, \ldots, x_k)$, and introduce the short-hand notation

$$\boldsymbol{\alpha} = [\alpha_0, \ldots, \alpha_{c-1}]^\top$$

$$\widehat{\boldsymbol{p}} = [\widehat{p}_0, \ldots, \widehat{p}_{c-1}]^\top$$

$$\Delta \boldsymbol{\ell}_i = [\Delta \ell_0(h_1, h_2; x_i), \ldots, \Delta \ell_{c-1}(h_1, h_2; x_i)]^\top$$

$$\widehat{\Delta \mathbb{E}} = [\widehat{\Delta \mathbb{E}}_0, \ldots, \widehat{\Delta \mathbb{E}}_{c-1}]^\top ,$$

$$\boldsymbol{\eta}_i = [\eta_0(x_i), \ldots, \eta_{c-1}(x_i)]^\top ,$$

and its (approximately) centered versions

$$\widetilde{\boldsymbol{\alpha}} = \boldsymbol{\alpha} - \widehat{\boldsymbol{p}}$$

$$\widetilde{\Delta}\boldsymbol{\ell}_i = \Delta \boldsymbol{\ell}_i - \widehat{\Delta \mathbb{E}}$$

$$\widetilde{\boldsymbol{\eta}}_i = \boldsymbol{\eta}_i - \widehat{\boldsymbol{p}} \, .$$

With this notation, we have

$$\widetilde{\Delta}\ell_b(h_1, h_2; z) = \sum_{r=0}^{c-1}(\alpha_r - \widehat{p}_{S_2,r}) \sum_{i=1}^{k}\left(\Delta\ell_r(h_1, h_2; x_i) - \widehat{\mathbb{E}}_{S_1}[\Delta\ell_r(h_1, h_2)]\right) + \sum_{r=0}^{c-1}\widehat{p}_{S_2,r}\,\widehat{\mathbb{E}}_{S_1}[\Delta\ell_r(h_1, h_2)]$$

$$= \widetilde{\boldsymbol{\alpha}}^{\top}\left(\sum_{i=1}^{k}\widetilde{\Delta}\boldsymbol{\ell}_i\right) + \widehat{\boldsymbol{p}}^{\top}\widehat{\Delta\mathbb{E}}\,.$$

Thus, following the corresponding steps in the proof of Lemma C.1, in the conditional space where $S_1$ and $S_2$ are given (we again omit from the notation the conditioning on $S_1$ and $S_2$ for notational comfort), we can write

$$\text{Var}_z\left(\widetilde{\Delta}\ell_b(h_1, h_2; z)\right) \le \mathbb{E}\left[\left(\sum_{i=1}^{k}\widetilde{\Delta}\boldsymbol{\ell}_i\right)^{\top}\mathbb{E}[\widetilde{\boldsymbol{\alpha}}\widetilde{\boldsymbol{\alpha}}^{\top} \,|\, \mathcal{B}]\left(\sum_{i=1}^{k}\widetilde{\Delta}\boldsymbol{\ell}_i\right)\right],$$

with

$$\mathbb{E}[\widetilde{\boldsymbol{\alpha}}\widetilde{\boldsymbol{\alpha}}^{\top} \,|\, \mathcal{B}] = \frac{1}{k^2}\left(\widetilde{\boldsymbol{\sigma}}^*(\widetilde{\boldsymbol{\sigma}}^*)^{\top} - \Sigma^*\right),$$

where

$$\boldsymbol{\sigma}^* = \sum_{i=1}^{k}\boldsymbol{\eta}_i\,, \qquad \widetilde{\boldsymbol{\sigma}}^* = \sum_{i=1}^{k}\widetilde{\boldsymbol{\eta}}_i$$

$$\Sigma^* = \left[\sum_{i=1}^{k}\eta_r(x_i)\eta_s(x_i)\right]_{r,s=0}^{(c-1)\times(c-1)} = \sum_{i=1}^{k}\boldsymbol{\eta}_i\boldsymbol{\eta}_i^{\top}\,.$$

As a consequence, as in the proof of Lemma C.1,

$$\text{Var}_z\left(\widetilde{\Delta}\ell_b(h_1, h_2; z)\right) \le \frac{1}{k^2}\,\mathbb{E}\left[\left(\left(\sum_{i=1}^{k}\widetilde{\Delta}\boldsymbol{\ell}_i\right)^{\top}\widetilde{\boldsymbol{\sigma}}^*\right)^2\right].$$

But

$$\widetilde{\Delta}\boldsymbol{\ell}_i = \Delta\boldsymbol{\ell}_i - \Delta\mathbb{E} + (\Delta\mathbb{E} - \widehat{\Delta}\mathbb{E})\,, \qquad \widetilde{\boldsymbol{\eta}}_i = \boldsymbol{\eta}_i - \boldsymbol{p} + (\boldsymbol{p} - \widehat{\boldsymbol{p}})\,,$$

so that

$$\left(\sum_{i=1}^{k}\widetilde{\Delta}\boldsymbol{\ell}_i\right)^{\top}\widetilde{\boldsymbol{\sigma}}^* = \left(\sum_{i=1}^{k}(\Delta\boldsymbol{\ell}_i - \Delta\mathbb{E}) + k(\Delta\mathbb{E} - \widehat{\Delta}\mathbb{E})\right)^{\top}\left(\sum_{i=1}^{k}(\boldsymbol{\eta}_i - \boldsymbol{p}) + k(\boldsymbol{p} - \widehat{\boldsymbol{p}})\right)$$

$$= \left(\sum_{i=1}^{k}(\Delta\boldsymbol{\ell}_i - \Delta\mathbb{E})\right)^{\top}\left(\sum_{i=1}^{k}(\boldsymbol{\eta}_i - \boldsymbol{p})\right) + k\left(\sum_{i=1}^{k}(\Delta\boldsymbol{\ell}_i - \Delta\mathbb{E})\right)^{\top}(\boldsymbol{p} - \widehat{\boldsymbol{p}})$$

$$+ k\left(\Delta\mathbb{E} - \widehat{\Delta}\mathbb{E}\right)^{\top}\left(\sum_{i=1}^{k}(\boldsymbol{\eta}_i - \boldsymbol{p})\right) + k^2\left(\Delta\mathbb{E} - \widehat{\Delta}\mathbb{E}\right)^{\top}(\boldsymbol{p} - \widehat{\boldsymbol{p}})\,,$$

and, using $(a + b + c + d)^2 \leq 4a^2 + 4b^2 + 4c^2 + 4d^2$,

$$\left( \left( \sum_{i=1}^{k} \widetilde{\Delta} \boldsymbol{\ell}_i \right)^{\top} \widetilde{\boldsymbol{\sigma}}^* \right)^2 \leq 4 \left( \left( \sum_{i=1}^{k} (\Delta \boldsymbol{\ell}_i - \Delta \, \mathbb{E}) \right)^{\top} \left( \sum_{i=1}^{k} (\boldsymbol{\eta}_i - \boldsymbol{p}) \right) \right)^2 + 4k^2 \left( \left( \sum_{i=1}^{k} (\Delta \boldsymbol{\ell}_i - \Delta \, \mathbb{E}) \right)^{\top} (\boldsymbol{p} - \widehat{\boldsymbol{p}}) \right)^2$$

$$+ 4k^2 \left( \left( \Delta \, \mathbb{E} - \widehat{\Delta} \, \mathbb{E} \right)^{\top} \left( \sum_{i=1}^{k} (\boldsymbol{\eta}_i - \boldsymbol{p}) \right) \right)^2 + 4k^4 \left( \left( \Delta \, \mathbb{E} - \widehat{\Delta} \, \mathbb{E} \right)^{\top} (\boldsymbol{p} - \widehat{\boldsymbol{p}}) \right)^2$$

$$\leq 4 \underbrace{\left( \left( \sum_{i=1}^{k} (\Delta \boldsymbol{\ell}_i - \Delta \, \mathbb{E}) \right)^{\top} \left( \sum_{i=1}^{k} (\boldsymbol{\eta}_i - \boldsymbol{p}) \right) \right)^2}_{(I)} + 4k^2 \underbrace{\left\| \sum_{i=1}^{k} (\Delta \boldsymbol{\ell}_i - \Delta \, \mathbb{E}) \right\|_2^2 \| \boldsymbol{p} - \widehat{\boldsymbol{p}} \|_2^2}_{(II)}$$

$$+ 4k^2 \underbrace{\left\| \Delta \, \mathbb{E} - \widehat{\Delta} \, \mathbb{E} \right\|_2^2 \left\| \sum_{i=1}^{k} (\boldsymbol{\eta}_i - \boldsymbol{p}) \right\|_2^2}_{(III)} + 4k^4 \underbrace{\left\| \Delta \, \mathbb{E} - \widehat{\Delta} \, \mathbb{E} \right\|_2^2 \left\| \boldsymbol{p} - \widehat{\boldsymbol{p}} \right\|_2^2}_{(IV)} .$$

Plugging back,

$$\mathrm{Var}_z \left( \widetilde{\Delta} \ell_b(h_1, h_2; z) \right) \leq \frac{1}{k^2} \, \mathbb{E}\left[(I)\right] + \frac{1}{k^2} \, \mathbb{E}\left[(II)\right] + \frac{1}{k^2} \, \mathbb{E}\left[(III)\right] + \frac{1}{k^2} \, \mathbb{E}\left[(IV)\right] .$$

We treat the four terms separately. For the first term, we can immediately apply the variance bound in Lemma C.1, leading to

$$\frac{1}{k^2} \, \mathbb{E}\left[(I)\right] \leq 64 \, \mathbb{E}\left[ \max_r \left( \Delta \ell_r(h_1, h_2; x) \right)^2 \right] .$$

We know from (36) that

$$\| \boldsymbol{p} - \widehat{\boldsymbol{p}} \|_2^2 \leq \sum_{r=0}^{c-1} (B_{\delta,r}(h_1, h_2))^2 = O\left( \frac{\log(c/\delta)}{mk} \right) ,$$

and, from (35), that

$$\left\| \Delta \, \mathbb{E} - \widehat{\Delta} \, \mathbb{E} \right\|_2^2 \leq \sum_{r=0}^{c-1} (A_{\delta,r}(h_1, h_2))^2 = O\left( \frac{\log(c \, |\mathcal{H}|/\delta)}{mk} \sum_{r=0}^{c-1} \sigma_r^2(h_1, h_2) \right) .$$

Moreover, since the vectors $\Delta \boldsymbol{\ell}_i - \Delta \, \mathbb{E}$ and the vectors $\boldsymbol{\eta}_i - \boldsymbol{p}$ are centered,

$$\mathbb{E}\left[ \left\| \sum_{i=1}^{k} (\Delta \boldsymbol{\ell}_i - \Delta \, \mathbb{E}) \right\|_2^2 \right] = \mathbb{E}\left[ \sum_{r=0}^{c-1} \left( \sum_{i=1}^{k} (\Delta \ell_r(h_1, h_2; x_i) - \Delta \mathbb{E}_r) \right)^2 \right] = k \sum_{r=0}^{c-1} \sigma_r^2(h_1, h_2)$$

and

$$\mathbb{E}\left[ \left\| \sum_{i=1}^{k} (\boldsymbol{\eta}_i - \boldsymbol{p}) \right\|_2^2 \right] = \mathbb{E}\left[ \sum_{r=0}^{c-1} \left( \sum_{i=1}^{k} (\eta_r(x_i) - p_r) \right)^2 \right]$$

$$= k \sum_{r=0}^{c-1} \mathrm{Var}(\eta_r(x))$$

$$\leq k \sum_{r=0}^{c-1} p_r (1 - p_r)$$

$$\leq k \sum_{r=0}^{c-1} p_r = k .$$

Hence

$$\frac{1}{k^2} \, \mathbb{E}\Big[(II)\Big] = O\left(\frac{\log(c/\delta)}{m} \sum_{r=0}^{c-1} \sigma_r^2(h_1, h_2)\right)$$

$$\frac{1}{k^2} \, \mathbb{E}\Big[(III)\Big] = O\left(\frac{\log(c|\mathcal{H}|/\delta)}{m} \sum_{r=0}^{c-1} \sigma_r^2(h_1, h_2)\right)$$

$$\frac{1}{k^2} \, \mathbb{E}\Big[(IV)\Big] = O\left(\left(\frac{\log(c|\mathcal{H}|/\delta)}{m}\right)^2 \sum_{r=0}^{c-1} \sigma_r^2(h_1, h_2)\right) .$$

Piecing together, and leveraging the condition $m = \Omega(\log(c|\mathcal{H}|/\delta))$ yields

$$\mathrm{Var}_z\Big(\widetilde{\Delta}\ell_b(h_1, h_2; z)\Big) = O\left(\mathbb{E}\Big[\max_r\Big(\Delta\ell_r(h_1, h_2; x)\Big)^2\Big] + \frac{\log(c|\mathcal{H}|/\delta)}{m} \sum_{r=0}^{c-1} \sigma_r^2(h_1, h_2)\right) . \qquad (38)$$

We are now ready to combine (34), (37), and (38). If we assume that $S_1$ and $S_2$ have been drawn in such a way that both (34) and (37) simultaneously hold for all $h_1, h_2 \in \mathcal{H}$, we conclude that

$$\mathbb{P}\Bigg(\forall h_1, h_2 \in \mathcal{H} \times \mathcal{H} \ \Big|Q(h_1, h_2; S) - \Delta\mathcal{L}(h_1, h_2)\Big|$$

$$\leq O\Bigg(\sqrt{\mathbb{E}\Big[\max_r\Big(\Delta\ell_r(h_1, h_2; x)\Big)^2\Big] + \frac{\log(c|\mathcal{H}|/\delta)}{m} \sum_{r=0}^{c-1} \sigma_r^2(h_1, h_2)} \sqrt{\frac{\log(|\mathcal{H}|/\delta)}{m}}$$

$$+ \sqrt{\frac{c \, \log(c/\delta)}{mk}} \max_r |\Delta\mathbb{E}_r|\Bigg)\Bigg)$$

$$\geq 1 - \delta .$$

Similar to the proof of Theorem 3.2, we define

$$C_\delta(h_1, h_2, |\mathcal{H}|, m, k, c)$$

$$= O\Bigg(\sqrt{\mathbb{E}\Big[\max_r\Big(\Delta\ell_r(h_1, h_2; x)\Big)^2\Big] + \frac{\log(c|\mathcal{H}|/\delta)}{m} \sum_{r=0}^{c-1} \sigma_r^2(h_1, h_2)} \sqrt{\frac{\log(|\mathcal{H}|/\delta)}{m}}$$

$$+ \sqrt{\frac{c \, \log(c/\delta)}{mk}} \max_r |\Delta\mathbb{E}_r|\Bigg)$$

and make further overapproximations to simplify the above expression. In particular, we set

$$\Delta\,\mathbb{E} = [\mathbb{E}_0, \ldots, \mathbb{E}_{c-1}]^\top = [\mathbb{E}[\Delta\ell_0(h_1, h_2; x)], \ldots, \mathbb{E}[\Delta\ell_{c-1}(h_1, h_2; x)]]^\top = \mathbb{E}[\Delta\boldsymbol{\ell}] ,$$

with

$$\Delta\boldsymbol{\ell} = [\Delta\ell_0(h_1, h_2; x)], \ldots, \Delta\ell_{c-1}(h_1, h_2; x)]]^\top .$$

Then note that, by the convexity of $||\cdot||_\infty$,

$$\max_r |\Delta\mathbb{E}_r| = ||\Delta\mathbb{E}||_\infty \leq \mathbb{E}[||\Delta\boldsymbol{\ell}||_\infty] = \mathbb{E}_x\Big[\max_r |\Delta\ell_r(h_1, h_2; x)|\Big] .$$

Then

$$C_\delta(h_1, h_2, |\mathcal{H}|, m, k, c)$$

$$= O\left(\sqrt{\mathbb{E}\left[\max_r\left(\Delta\ell_r(h_1, h_2; x)\right)^2\right] + \frac{\log(c|\mathcal{H}|/\delta)}{m}\sum_{r=0}^{c-1}\sigma_r^2(h_1, h_2)} \sqrt{\frac{\log(|\mathcal{H}|/\delta)}{m}}\right.$$

$$\left. + \sqrt{\frac{c\log(c/\delta)}{mk}}\,\mathbb{E}_x\left[\max_r|\Delta\ell_r(h_1, h_2; x)|\right]\right).$$

We then apply Proposition B.1 with $h_1 = h \in \mathcal{H}_{\beta/4}$ and $h_2 = \widehat{h}^\star$, and following the same argument as in the proof of Theorem 3.2, we conclude that the resulting sample complexity is

$$m = O\left(\frac{\Delta\ell^2(\mathcal{H})\,\log(|\mathcal{H}|/\delta)}{\beta^2} + \frac{\Sigma(\mathcal{H})\,\log(c|\mathcal{H}|/\delta)}{\beta} + \frac{c\left(\Delta|\ell|(\mathcal{H})\right)^2\log(c/\delta)}{\beta^2\,k}\right),$$

where

$$\Delta\ell^2(\mathcal{H}) = \max_{h\in\mathcal{H}}\mathbb{E}_x\left[\max_r\left(\Delta\ell_r(h, \widehat{h}^\star; x)\right)^2\right],$$

$$\Delta|\ell|(\mathcal{H}) = \max_{h\in\mathcal{H}}\mathbb{E}_x\left[\max_r\left|\Delta\ell_r(h, \widehat{h}^\star; x)\right|\right],$$

and

$$\Sigma(\mathcal{H}) = \sqrt{\max_{h\in\mathcal{H}}\sum_{r=0}^{c-1}\sigma_r^2(h, \widehat{h}^\star)}.$$

We further simplify the above expression by noting that

$$\left(\Delta|\ell|(\mathcal{H})\right)^2 = \left(\max_{h\in\mathcal{H}}\mathbb{E}_x\left[\max_r\left|\Delta\ell_r(h, \widehat{h}^\star; x)\right|\right]\right)^2$$

$$\leq \max_{h\in\mathcal{H}}\mathbb{E}_x\left[\left(\max_r\left|\Delta\ell_r(h, \widehat{h}^\star; x)\right|\right)^2\right]$$

$$= \max_{h\in\mathcal{H}}\mathbb{E}_x\left[\max_r\left(\Delta\ell_r(h, \widehat{h}^\star; x)\right)^2\right]$$

$$= \Delta\ell^2(\mathcal{H}).$$

Plugging back and rearranging concludes the proof.

# D. Experimental Setup for Section 4 and Further Experimental Results

This appendix contains further details on our experimental setup, as well as test set results based on the Area Under the Curve (AUC) metric.

## D.1. Datasets and Models

**MNIST:** MNIST (LeCun et al., 2010) is a multi-class image prediction task where each example contains a $28 \times 28$ grayscale image of a hand-written digit and the label indicates which digit it is. The data consists of 60,000 training images and 10,000 test images. We consider a binarized version of MNIST where digits in $\{0, 2, 4, 6, 8\}$ are positive examples and digits on $\{1, 3, 5, 7, 9\}$ are negative, so that the task is to predict whether the digit is even or odd. The model we use for MNIST is a Convolutional Neural Network (CNN) with the following layers: Convolution with 32 filters, max pooling with $2 \times 2$ window and stride, Convolution with 64 features, dropout layer with rate $0.5$, and finally a fully connected layer with a single output. The two convolutional layers are followed by ReLU activations, while the final output has no activation so that it outputs a logit.

**CIFAR-10:** CIFAR-10 (Krizhevsky, 2009) is a multi-class image prediction task where each example contains a $32 \times 32$ color image which belongs to one of the following classes: AIRPLANE, AUTOMOBILE, BIRD, CAT, DEER, DOG, FROG, HORSE, SHIP, or TRUCK. The training data consists of 50,000 images and 10,000 test images. We consider two binarized versions of CIFAR-10: Animal-vs-Machine, where the positive labels are BIRD, CAT, DEER, DOG, FROG, and HORSE, and Cat-vs-Rest, where CAT is the only positive label. We use the same CNN architecture for CIFAR-10 as was used in MNIST.

**Higgs:** The Higgs dataset (Baldi et al., 2014) is a simulated dataset where the goal is to distinguish between processes that produce Higgs bosons and a background process that does not. The dataset consists of 11,000,000 examples and we split them into training and testing by taking the first 10,000 examples as test examples with the remaining examples as being training examples. Each example has 21 features which are a mix of directly measured properties of the system and hand-crafted high level features. Our model class is a fully connected model with 4 hidden layers each having 300 neurons and ReLU activations followed by a fully connected layer with 1 output and no activation so that it outputs a logit.

**Adult:** The Adult dataset (Kohavi & Becker, 1996) is a dataset derived from the 1994 US Census Database where the goal is to predict whether an individual's income exceeds $50,000 per year or not. The features include several categorical and numerical features describing the individual. We pre-process the data by one-hot encoding all categorical features and rescaling each numerical feature so that the values fall within the interval $[0, 1]$. Our model class is a fully connected network with a single hidden layer with 32 neurons and ReLU activation, followed by the output layer with no activation.

**Criteo:** The Criteo Display Advertising Challenge (Tien et al., 2014) is a competition where the goal is to predict whether a given ad impression will result in the user clicking on the ad. Each example is described by a total of 39 features, 13 of which are integer "count" based features, and 26 of which are categorical features. To preserve privacy, Criteo pre-processed the dataset by hashing each of the categorical features into 32 bit hashes, and they do not provide high-level descriptions of what the original feature values corresponded to. We use only the training data which consits of 37M ad impressions together with whether they were clicked or not. The examples are ordered by ad impression time and come from a span of 7 days. An important characteristic of this dataset is that the click rate changes periodically. The model class we use for the Criteo dataset is a Deep Embedding Model, which works by mapping each feature value to an embedding vector and then applying several fully connected layers to the concatenated embeddings. To embed the integer-valued features, we first discretize each feature into buckets using logarithmically-spaced bucket boundaries with 40 boundaries. Then, each bucket is associated with a learned embedding vector. For each categorical feature, we map the hashed value to an integer bucket by mapping hashing again and taking the modulus with a per-feature number of buckets. For all integer features, we use 40 buckets and an embedding dimension of 10. Table 1 shows the number of buckets and embedding dimension used for the categorical features. In general, the number of buckets was chosen to be a round number approximately equal to the number of distinct values that appeared at least 100 times in the training data. Once the features are embedded, the model uses three fully connected hidden layers with 256, 128, and 64 neurons respectively, all using ReLU activations. The final output layer has no activation as it outputs a logit.

### D.2. Instance-level Losses and Label Smoothing

For completeness, in this section we briefly define the instance-level losses that we use in our experiments. These losses are transformed into bag-level losses via the recipes for each of GENERALUPM, EASYLLP, and PM.

In the online Criteo learning experiment, we instantiate each LLP loss for the binary cross-entropy instance-level loss. That is, when the model predicts a logit $\hat{y} \in \mathbb{R}$ for an example with label $y \in \{0, 1\}$, the loss is given by

$$\ell_{\text{ce}}(\hat{y}, y) = -y \log(\sigma(\hat{y})) - (1 - y) \log(1 - \sigma(\hat{y})),$$

where $\sigma$ is the logistic function. We use the optax (DeepMind et al., 2020) implementation of the cross entropy loss which is implemented in log space to avoid numerical issues.

For the batch learning experiments, we modify the cross-entropy loss to use *label smoothing*, which is a common technique for preventing models from becoming overly confident over long training runs. Concretely, this corresponds to defining a new loss

$$\ell_{\text{ces}}(\hat{y}, y) = \ell_{\text{ce}}(\hat{y}, (1 - \epsilon)y + \epsilon/2),$$

where $\epsilon$ is a parameter controlling the strength of the label smoothing. In our batch experiments we simply choose $\epsilon = 0.1$.

| Feature Name | Number of Buckets | Embedding Dimension |
|---|---|---|
| categorical-feature-14 | 700 | 30 |
| categorical-feature-15 | 500 | 30 |
| categorical-feature-16 | 10000 | 70 |
| categorical-feature-17 | 10000 | 70 |
| categorical-feature-18 | 200 | 22 |
| categorical-feature-19 | 20 | 10 |
| categorical-feature-20 | 10000 | 70 |
| categorical-feature-21 | 400 | 30 |
| categorical-feature-22 | 10 | 10 |
| categorical-feature-23 | 10000 | 70 |
| categorical-feature-24 | 5000 | 50 |
| categorical-feature-25 | 10000 | 70 |
| categorical-feature-26 | 5000 | 50 |
| categorical-feature-27 | 50 | 10 |
| categorical-feature-28 | 5000 | 53 |
| categorical-feature-29 | 10000 | 70 |
| categorical-feature-30 | 10 | 10 |
| categorical-feature-31 | 5000 | 50 |
| categorical-feature-32 | 5000 | 50 |
| categorical-feature-33 | 10 | 10 |
| categorical-feature-34 | 10000 | 70 |
| categorical-feature-35 | 10 | 10 |
| categorical-feature-36 | 10 | 10 |
| categorical-feature-37 | 10000 | 70 |
| categorical-feature-38 | 10 | 20 |
| categorical-feature-39 | 10000 | 70 |

*Table 1.* Criteo Categorical Feature Specifications

Note that we do not actually modify the labels of the training data. We just apply EASYLLP, GENERALUPM and PM to the loss $\ell_{\text{ces}}$.

### D.3. Bag-level Loss Implementations

In this section we review the definitions of EASYLLP and PM, together with details of how our implementations estimate the quantities $p$, $\mathbb{E}[f_1(x)]$ and $\mathbb{E}[f_2(x)]$ appearing in the definitions of EASYLLP and GENERALUPM. Throughout this section, fix a bag $z = (\mathcal{B}, \alpha) = ((x_1, \ldots, x_k), \alpha)$ and instance-level loss function $\ell$ (like $\ell_{\text{ce}}$ or $\ell_{\text{ces}}$ from the previous section).

**Proportion Matching (PM).** The bag-level log loss for PM is simply defined as

$$\ell_{\text{PM}}(h, z) = \ell(h(\mathcal{B}), \alpha)$$

where

$$h(\mathcal{B}) = \frac{1}{k} \sum_{i=1}^{k} h(x_i) \,.$$

**EASYLLP (Busa-Fekete et al., 2023).** Starting from an instance-level loss $\ell$, the bag-level Easy LLP loss is defined as

$$\ell_{\text{EASYLLP}}(h, z) = \frac{1}{k} \sum_{i=1}^{k} \big(k(\alpha - p) + p\big) \cdot \ell(h(x_i), 1) + \big(k(p - \alpha) + (1 - p)\big) \cdot \ell(h(x_i), 0).$$

**Implementation of Parameter Estimation.** The definitions of both EASYLLP and GENERALUPM require estimates of the label marginal $p = \mathbb{P}(y = 1)$, which is the probability of observing a positive example from the underlying distribution.

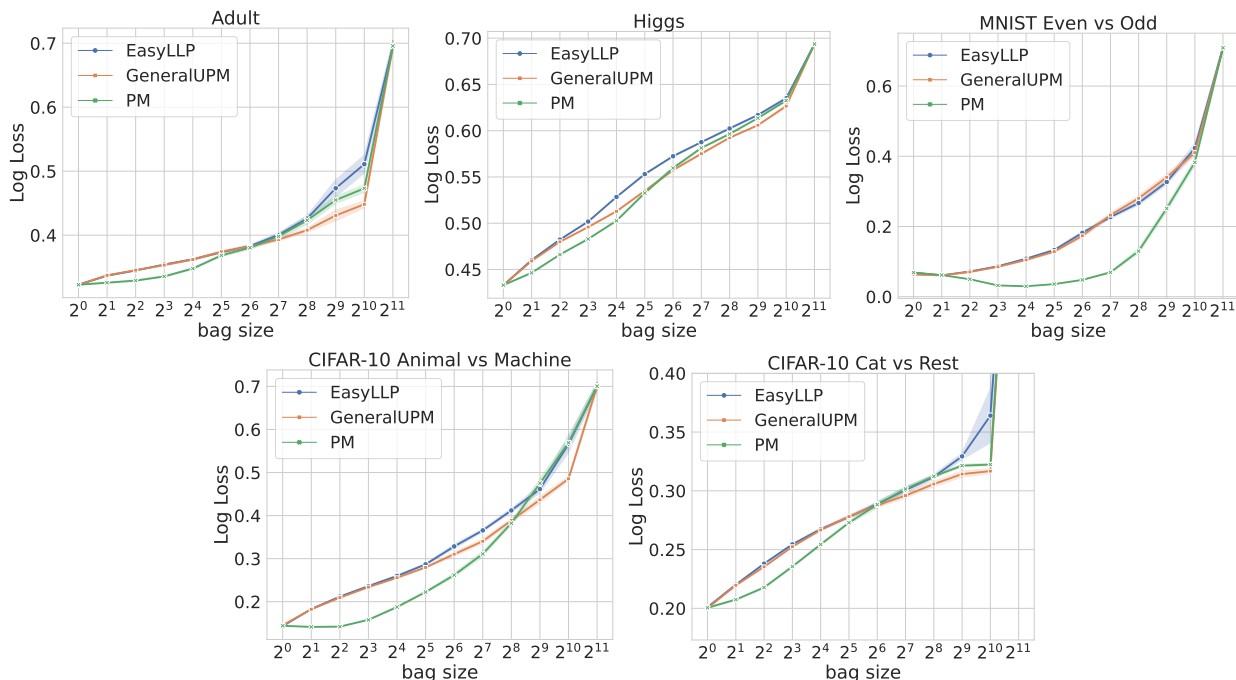

*Figure 3.* Average test log loss when training using each aggregate loss in the batch setting for all bag sizes. For each bag size we report the lowest log loss achieved over all learning rate and stopping epoch combinations. Error bars indicate one standard error in the mean across repetitions.

Beyond this, GENERALUPM also requires estimates of $\mathbb{E}[f_1(x)]$ and $\mathbb{E}[f_2(x)]$. In this section we describe how those quantities are estimated in our experiments.

*Label Marginal $p$.* In our batch learning experiments we estimate $p$ to be the average label proportion among all the bags in the training data. This calculation is done once before training begins and the same value of $p$ is used in every evaluation of the LLP loss functions. For the online Criteo experiment, as described in the main body, we process the data in chunks containing $2^{16}$ examples. After reading each chunk, we estimate $p$ to be the average label proportion among all bags in the chunk, and that value is used in every evaluation of the LLP loss function for that chunk alone. This is essential for achieving good performance on the Criteo dataset because the actual click rate (i.e. value of $p$) changes over time due to changes in the set of users interacting with ads at different times of the day, new ad campaigns launching, etc.

*Model's Average Loss.* Recall that $f_1(x) = \ell(h(x), 0)$ and $f_2(x) = \ell(h(x), 1) - \ell(h(x), 0)$ correspond to the model's loss on example $x$ if the true label were 0, and the difference in loss between label 1 and 0, respectively. Since these quantities depend on the model parameters, we cannot estimate them once at the beginning of training and treat them as a constant. In particular, their values change during training, but also the gradient of these terms with respect to the model parameters is non-zero. To estimate the expected value of $f_1$ and $f_2$ on a random $x$ drawn from the data distribution we employ the following scheme: Each SGD training batch consists of at least $B \geq 2$ bags. When evaluating the GENERALUPM loss on one bag within a batch, we estimate the expected value of $f_1(x)$ and $f_2(x)$ by averaging their values over the feature vectors contained in the remaining $B - 1$ bags. This results in unbiased estimates that are uncorrelated with the other components of the LLP loss for that bag. The variance of the resulting estimate depends on the relative size of the bag and batch size, since when the bag size approaches the batch size, we have fewer examples for estimating the expectations.

### D.4. Complete Log Loss Plots

Figure 3 shows the complete set of results for the batch experiments reporting the testing average log loss (including bag sizes smaller than $2^4$ and bag size $2^{11}$).

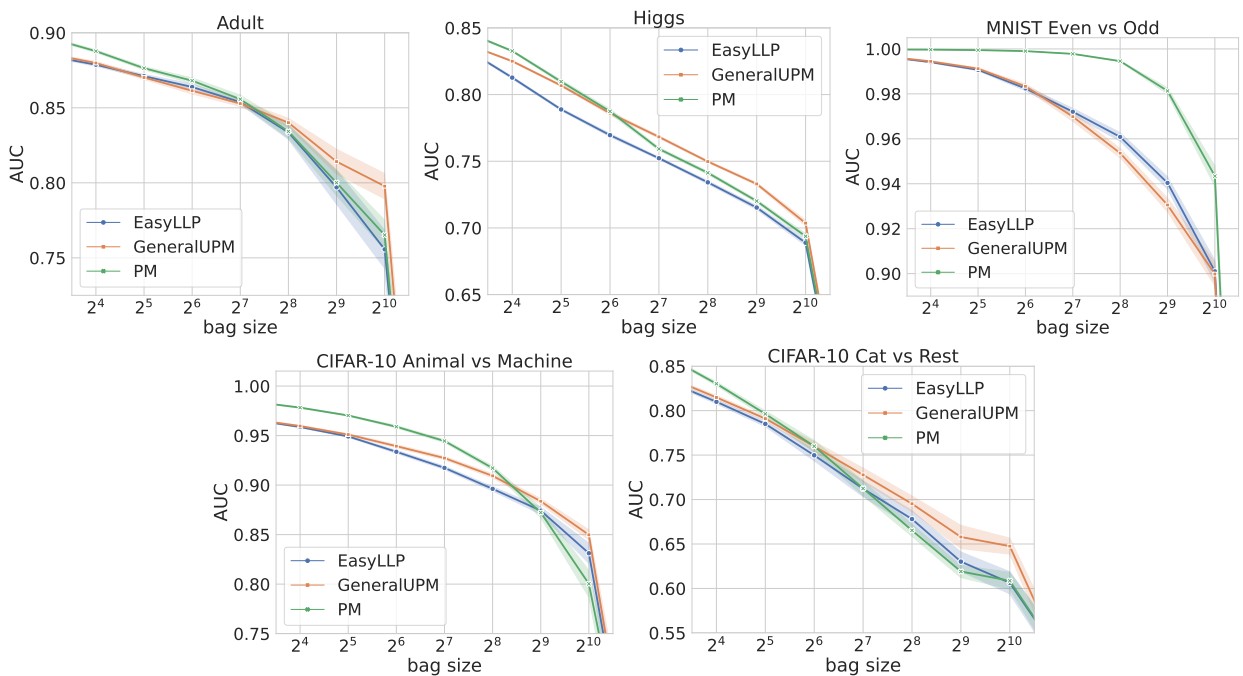

*Figure 4.* Average test AUC when training using each aggregate loss in the batch setting. For each bag size we report the highest AUC achieved over all learning rate and stopping epoch combinations. Error bars indicate one standard error in the mean across repetitions.

### D.5. AUC Plots

Figure 4 and Figure 6 present results that are identical to the setting of Figure 1 and Figure 2 except that we report the AUC of models rather than the average log loss. Also Figure 5 contains AUC results for all bag sizes in the batch setting.

The conclusions based on AUC are essentially the same as those from log loss with the exception of the Online Criteo experiment, where we find that when performance is measured by AUC, EASYLLP also out-performs PM at large bag sizes.

### D.6. Computing Resources

All experiments were conducted on a cluster of virtual machines using NVIDIA Tesla p100 GPUs. Each machine was equipped with 64GB memory and a virtualized CPU.

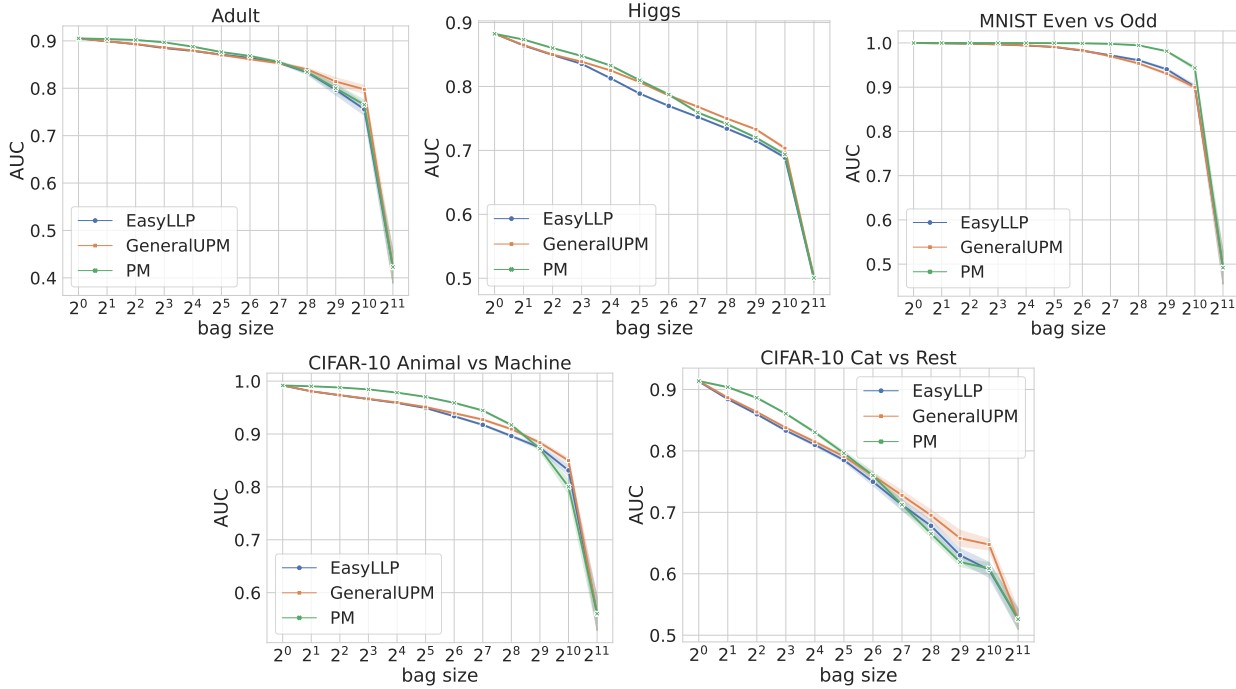

*Figure 5.* Average test AUC when training using each aggregate loss in the batch setting for all bag sizes. For each bag size we report the highest AUC achieved over all learning rate and stopping epoch combinations. Error bars indicate one standard error in the mean across repetitions.

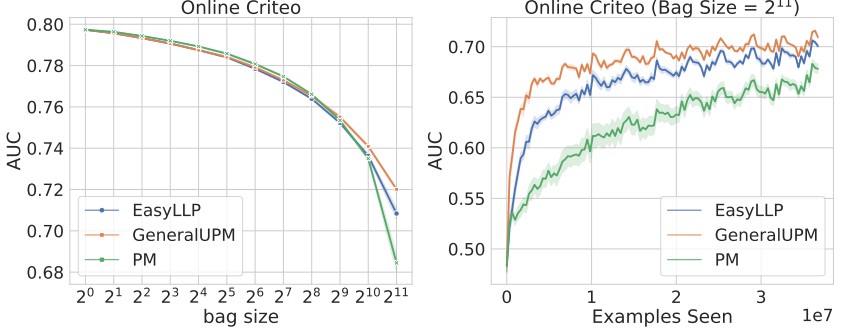

*Figure 6.* The left figure depicts the average chunk AUC for each LLP loss and bag size. The right figure shows the per-chunk AUC during online training for bag size $2^{11}$. Error bars indicate one standard error in the mean across repetitions.

