# OpenReview forum: "Optimal Learning from Label Proportions with General Loss Functions"
_ICML.cc/2026/Conference — ICML 2026 regular_

### Official Review · Reviewer_3a7T · 2026-03-10

**Soundness:** 3
**Presentation:** 4
**Significance:** 3
**Originality:** 3
**Overall Recommendation:** 4
**Confidence:** 3

**Summary:**

The paper tackles the weakly supervised problem of learning from label
proportions, in which one does not have access to individual labels, but only
to the proportion within groups (bags). In particular, the authors focus on the
case of non-overlapping independent bags to show that the regret of a
median-of-means tournament algorithm admits an upper bound that depends on the
number of bags but, crucially, not on the size of each bag. Their analysis is
successful in achieving optimal rates under a broad class of loss functions; real-world
datasets are used to compare their proposal with some prior work.

**Compliance With Llm Reviewing Policy:**

Affirmed.

**Final Justification:**

My positive view of the paper remains after the rebuttal period. Some of my non-major concerns have been addressed and it is understandable that experiments may not be readily available by the end of the short rebuttal window. I will keep my initial assessment as the core of my review remains the same, and I thank the authors for the lively discussion.

**Key Questions For Authors:**

- The paper claims in passing that the results can be lifted to cover
infinite-dimensional $\mathcal{H}$ (line 120). Has this been carried out formally and if so could more details be provided about the technical challenges and qualitative conclusions?

- Are there any reasons the baseline models mentioned in the "strengths and
weaknesses" section above were not included in the benchmarks?

- In Figure 1, is it the case that multiple hyperparameter configurations are
considered and the one with minimum loss at the test set is plotted? Are
there any justifications for not using a validation set for hyperparameter
selection?

- Line 2418 says the smoothing parameter $\epsilon$ was set to 0.1, which left
me wondering how important label smoothing is. In particular, are the paper's
results and conclusions robust to the choice of $\epsilon$?

- Algorithm 1 requires $(|\mathcal{H}| (|\mathcal{H}| - 1)) / 2$ comparisons, which can be
quite costly. Have the authors ever had $|\mathcal{H}|$ small enough in a real-world setting such that Algorithm 1
was practical?

- What supports the claim that log loss is more in line with the motivation of
conversion prediction (line 107)?

- Can you offer any guidance for a practitioner with a problem in which bags
overlap or are not independent of the data content?

- If the proportion $\alpha$ is not the mean but a different statistic (e.g.,
the median or another quantile), do you anticipate all the same conclusions
from the paper to hold?

- Is there any specific reason to require $|S_1| = |S_2| = |S_3|$ in Equation (4), or could one budget more data to the estimation of one of the quantities?

**Limitations:**

Yes.

**Strengths And Weaknesses:**

## Strengths

- Clear presentation: the paper is well structured, well written and clear, with easy-to-follow proofs.

- Motivation: the work is well motivated, with web advertising a concrete application.

- Generality: the theory accommodates some unbounded losses such as logarithmic loss and extends to some multi-class classification settings, making it considerably general and applicable.

- Bounds independent of the bag size: this is a significant theoretical improvement over prior
work.

## Weaknesses

- Theory-practice disconnect: the median-of-means tournament algorithm (Algorithm 1) for which
the main results hold (Theorems 3.2 and 3.5) is computationally demanding and impractical in many real-world problems; the authors instead carry out experiments using an SGD-like algorithm (GeneralUPM).

- Comparison to baselines: the empirical results could include comparisons to more baselines, such as DLLP (Ardehaly and Culotta, 2017), GenBags (Saket et al., 2022), and the iterative algorithm from Havaldar et al. (2024).

[1] Ehsan Mohammady Ardehaly and Aron Culotta. Co-training for demographic classification using deep learning from label proportions. In IEEE International Conference on Data Mining Workshops. 2017

[2] Rishi Saket, Aravindan Raghuveer, and Balaraman Ravindran. On combining bags to better learn from label proportions. In International Conference on Artificial Intelligence and Statistics. 2022.

[3] Shreyas Havaldar, Navodita Sharma, Shubhi Sareen, Karthikeyan Shanmugam, and Aravindan Raghuveer. Learning from Label Proportions: Bootstrapping Supervised Learners via Belief Propagation. In the International Conference on Learning Representations. 2024.

- Literature review: it would also be helpful to discuss papers [1] and [3]
above, as they provide different approaches to the problem of learning from
label proportions.

- Minor: $\sigma^2_{f_2(\mathcal{H})}$ should be $\sigma^2_{f_2}(\mathcal{H})$ in Theorem B.2;
"set" should be "let" in line 223; "otpax" should be "optax" in line 2411;
section 2 defines $\eta$ but it's not used in the main paper, just in the
appendix, so it might make sense to keep the definition in the appendix too;
the overloading of $h$ in lines 283 and 288 feels a bit clunky; some practical
details are left unanswered (see next section) and reproducibility is hindered
by lack of code availability; $\mathbf{\eta}_i$ in proof of Lemma C.1 should be
a column vector, otherwise following operations are not well defined; although
standard, might be worth noting why $r$ was set to $8 \log(1/\delta)$ in line 299.

---

> ### Author Rebuttal · Authors · 2026-03-30
>
> We thank Reviewer 3a7T for their thorough reading of our paper.
>
> - Theory-practice disconnect
>
> This is a point raised by all reviewers, we are responding to each reviewer individually.
>
> The main point of this paper is the design of our centered beg level estimators (for either binary or multiclass), and the analysis of their properties (unbiasedness, variance independent of $k$, etc.). The algorithmics we can build on top of these estimators, whether it's a MoM tournament, an ERM estimator, or an SGD procedure, is quite standard statistical learning constructions. Crucially, their nuances *are not* specific to LLP, but mirror those in standard supervised learning. In our theoretical analysis, we opted for a MoM tournament to facilitate the fast convergence rates analysis, and avoid the limitations of local Rademacher complexity which does not readily accommodate unbounded losses.
>
> Moreover, while standard Rademacher complexity guarantees apply to ERM with unbounded losses for slow $O(1/\sqrt{n})$ rates, they don't align with deep networks trained via SGD. A generic ERM estimator differs significantly from the one computed by SGD applied to deep networks. On the other hand, directly analyzing SGD on our bag-level loss would require convexity, a condition often violated in practice, leading to a common theory-practice disconnect.
>
> - Experimental comparison to more baselines
>
> This paper is about designing loss functions for LLP, and associated generalization guarantees, so we thought natural to empirically compare to papers with similar aims. We acknowledge various LLP approaches (cited in Appendix A) and will include reviewer-suggested citations. Ardehaly and Culotta, 2017 is closer in spirit to our Proportion Matching (PM) in using KL-vicinity between average model prediction and average label within bags. Havaldar et al. is another experimental paper that uses a Gibbs distribution to softly enforce a similar (but square loss-based) vicinity constraint. We believe they are orthogonal to our approach, in that we think our loss design can be combined with these approaches. Saket et al., 2022 seems mostly concerned with training performance, not generalization. Besides, they use an  expensive (SDP-based) training method for sampling bags.
>
> A full-fledged experimental comparison to a wide suite of representative LLP methods would be a paper on its own.
>
> - ``The paper claims in passing that . . . lifted to cover infinite-dimensional (line 120) . . . Has this been carried out formally?"
>
> Not in this paper. We decided not to include more content in this submission, since the paper already contains a lot of material. Besides, we believe these extensions are not specific to LLP.
>
> - Figure 1, choice of hyperparameters and validation set
>
> We used the best test set results obtained after tuning learning rates for each method (LL. 422-423). This ensures a fair relative performance comparison. While using a validation set for hyperparameter selection would lower all methods' absolute test performance, their $relative$ performance order would be similar.
>
> - L 2418, label smoothing and $\epsilon = 0.1$
>
> When dealing with log loss, label smoothing is an important regularization technique to avoid the ``overconfidence" phenomenon. This is common practice in DNN training, not specifically related to LLP. There is no special reason for the choice of $\epsilon=0.1$, we did not explore other values of $\epsilon$.
>
> - Practicality of Alg. 1
>
> We never tested the MoM tournament in Alg. 1. As said above, the MoM approach we followed here was only meant to facilitate our theoretical analysis.
>
> - Why log loss is more in line with conversion prediction (line 107)
>
> A main motivation for this research is estimator (2)'s ability to handle log-like losses. We use log loss in our experiments because we aim to output probabilities (conversion predictions are probability values) rather than binary labels. Minimizing log loss for probability outputs is a standard training practice that also results in calibrated predictions (under some assumptions).
>
> - Practical guidance when bags overlap or are not independent of data content
>
> Even when bags overlap or are data-dependent (e.g., due to clustering), our bag-level loss design might still offer practical variance reduction, despite the lack of full theoretical backing. Yet, we never collected empirical evidence for these cases.
>
> - Median or another quantile instead of mean
>
> The reviewer raises an interesting, non-obvious question, particularly relevant to the total multiclass setting. Addressing label quantiles instead of averages requires a substantial redesign of estimators and warrants dedicated research.
>
> - Eq. (4), why $|S_1| = |S_2| = |S_3|$?
>
> No reason other than simplifying the analysis presentation. Modifying size proportions would only affect results by constants, which are hidden by the big-O notation.
>
> - We will fix the minor points as well, thanks for bringing them to our attention!

---

> > ### Author Rebuttal · Reviewer_3a7T · 2026-04-02
> >
> > Thank you for the detailed responses. I remain with a positive view of the paper but at the same time believe some robustness checks would strengthen the empirical section. Are the conclusions the same if you use a validation set for hyperparameter
> > selection, which is closer to practice? Are the results robust to different choices of smoothing parameter $\epsilon$? Do the results hold when bags overlap or are data-dependent? Answering these and related questions in the positive would give more weight to the conclusions and improve the results. I agree with the authors' position that a "full-fledged experimental comparison to a wide suite of representative LLP methods would be a paper on its own", and the theory-practice disconnect is addressed to a certain degree, but hopefully these robustness checks are not too costly to carry out. Of course, I understand if time constraints prevent the authors from running these extra experiments.

---

> > > ### Author Response · Authors · 2026-04-03
> > >
> > > Thanks to Reviewer 3a7T for their further comments. As the ack was marked "(b) Partially resolved — I have follow-up questions," we view Reviewer 3a7T's inquiries as specific points that the reviewer feels require further clarification or were not fully addressed in our previous response.
> > >
> > > - ``Are the conclusions the same if you use a validation set for hyperparameter selection, which is closer to practice?"
> > >
> > > As we said in our rebuttal, we expect the conclusions will be the same (in the relative comparison sense), but we did not run experiments where hyperparameters have been tuned via a validation set, nor do we have the time to run them during the short turnaround time of a rebuttal.
> > >
> > > - ``Are the results robust to different choices of smoothing parameter $\epsilon$? "
> > >
> > > If $\epsilon$ is chosen within a reasonable range so as not to disrupt the performance of any of the algorithms engaged in our comparative study then, yes, we expect the relative performance of the algorithms will roughly be the same. But, again, we did not run these experiments, nor do we have the time to run them during this rebuttal period.
> > >
> > > - ``Do the results hold when bags overlap or are data-dependent?"
> > >
> > > This is an undisputedly interesting question. Yet, we would like to mention that overlapping or data-dependent bags are **outside** the scope of this paper. As for what we expect:
> > >
> > > (a) The theoretical results we have do *not* extend to data-dependent bags, but in principle they do extend to overlapping bags (each containing i.i.d data). This is because one can always treat each bag separately, and then pay for the extra variance originating from the overlap across bags. However, this extra variance might be *too large* if the amount of bag overlap is major one. In the latter case, more ad hoc methods should perhaps be applied that leverage the specific nature of the overlap.
> > >
> > > (b) Regarding empirical performance, our bag-level loss design may provide practical variance reduction even in scenarios with overlapping or data-dependent bags. However, as these cases fall outside the scope of the current work, we have not included empirical evaluations for them in this paper.

---

### Official Review · Reviewer_zdnN · 2026-03-12

**Soundness:** 4
**Presentation:** 2
**Significance:** 2
**Originality:** 3
**Overall Recommendation:** 3
**Confidence:** 3

**Summary:**

This paper studies ``learning from label proportion'', a special case of classification where the learner do not directly see the labels for individual samples, but rather the label proption for ``bags'' of multiple samples. There's several motivations for this, but one of the primary reason is protecting anonymity of individual samples by pooling. The author proves regret bounds for the median of means (MoM) tournament algorithm.

**Compliance With Llm Reviewing Policy:**

Affirmed.

**Final Justification:**

thank you for the rebuttal. i have updated the score.

**Key Questions For Authors:**

I'm confused by the experiments and how it relates to the theoretical results in the previous sections: are you using Algorithm 1 (or something like it)? What does GeneralUPM have to do with the results in Section 3?

Can the authors be more explicit about the family of practially relevant losses that the results apply to?

**Limitations:**

yes

**Strengths And Weaknesses:**

I appreciate the author's detailed literature review of why learning from label proportions is important, especially regarding machine learning and ads. The regret bound for MoM is an interesting contribution, compared to previous approaches using ERM.

Weaknesses:

The contributions section (starting on line 79) should make references to Theorem numbers in the work, so that it is easy to evaluate how this work compares to the previous works that the authors cite (for example, for the sentence on line 90 left column).

The authors claim to have proven a fast rate on a wider family of loss functions (as alluded to by the title of the paper), but upon reading the paragraphs around Theorem 3.5, it is unclear what this wider family of loss is. On line 300 right column, it is said that for the log loss, the logits are required to be bounded. Do we know that Algorithm 1 satisfy this condition? Without clarifying this, I would argue that the claim of fast rate for wider family of loss functions is not established.

In the abstract, it is said that the authors approach exhibits remarkable flexibility and cover practically relevant loss functions including in the multiclass setting. However, the two examples given are the square loss and the Poisson Log Loss. However, without reference to prior literature, the relevance of Poisson Log Loss is not established. Cross entropy, arguably the most widely used multiclass loss, is not covered by the analysis (which I commend the authors for pointing out, I appreciate that the authors are forthcoming about this. However, I believe this limitation undercut the main claim of the work in the abstract).

The experimental sections makes a sudden jump to discussing unbiased proportion matching loss, which doesn't appear in the previous sections. There should be connections between the theory and the experimental portion of the work. It is unclear.

---

> ### Author Rebuttal · Authors · 2026-03-29
>
> We thank Reviewer zdnN for reading our paper.
>
> - What does GeneralUPM have to do with the results in Section 3?
>
> GeneralUPM is just the DNN optimizer operating with the loss defined in Eq. (2). The parameter $p$ and the two expectations $E[f_1(h(x))]$ and $E[f_2(h(x))]$ in Eq. (2) have been replaced by empirical counterparts--see appendix D.3 for details.
> We apologize for this crucial confusion and we will fix and clarify this point.
>
> - Discrepancy between the theoretical algorithm and the practical implementation.
>
> This is a point raised by all reviewers, we are responding to each reviewer individually.
>
> The main point of this paper is the design of our centered beg level estimators (for either binary or multiclass), and the analysis of their properties (unbiasedness, variance independent of $k$, etc.). The algorithmics we can build on top of these estimators, whether it's a MoM tournament, an ERM estimator, or an SGD procedure, is quite standard statistical learning constructions. Crucially, their nuances *are not* specific to LLP, but mirror those in standard supervised learning. This, in particular, applies to conditions on the interplay between loss and data distribution to obtain fast rates of convergence. In our theoretical analysis, we opted for a MoM tournament to facilitate the study of fast convergence rates, and avoid the limitations of local Rademacher complexity, which does not accommodate unbounded losses. Our condition in Eq. (6) allowing us to deal with unbounded losses is more like a shortcut that avoids more nuanced conditions that are indeed available in the literature of statistical learning--see L. 333-337 (left column). Again, these are not specific to LLP.
>
> While standard Rademacher complexity guarantees apply to ERM with unbounded losses for slow $O(1/\sqrt{n})$ rates, they don't align with deep networks trained via SGD. A generic ERM estimator differs significantly from the one computed by SGD applied to deep networks. On the other hand, directly analyzing SGD on our bag-level loss would require convexity, a condition often violated in practice, leading to a common theory-practice disconnect.
>
> - On the fast rates and text surrounding Thm 3.5.
>
> Even in the standard statistical learning setting obtaining fast rates of convergence comes with extra assumptions about the loss and the data distribution (see LL. 335-337, left column). Hence in LLP (a strictly harder setting) we cannot avoid placing some such restrictions. The boundedness conditions on the logits is one of the simplest ones.
>
> - On the usage of Poisson log loss.
>
> Poisson log loss is a standard negative log-likelihood loss for counting processes, widely used in ads and e-commerce pCVR models. These models predict the number of conversions (0, 1, 2+), necessitating a loss based on a distribution over non-negative integers. We will add literature pointers, but the reviewer can also search for titles like "A Comparative Study of Count Data Models for Conversion Prediction" or "Modeling Online Advertising Effectiveness with Poisson Regression."
>
> - Cross-entropy loss not covered.
>
> This is not accurate. Our claim in L. 363 (left column) that ``The multi-class log loss/cross-entropy does not admit this
> representation" refers to what we called total multiclass setting. Cross entropy can still be handled in the (more standard, but more informative) full histogram setting that we analyze in the paper as well (Thm C.2, L. 1641). *Any* multiclass loss can in principle be handled in the full histogram setting. Again, if the loss is unbounded (as cross entropy is) bounded moments conditions are needed, even to achieve slow rates of convergence, even in the standard statistical learning setting.
>
> - Reference to theorem numbers, e.g., L. 90 left column.
>
> Thanks for the suggestion. We will add specific pointers to the theorems. In the case of L. 90, the theorem alluded to is Thm 3.2.
>
> - ``Can the authors be more explicit about the family of practically relevant losses that the results apply to?"
>
> Below is a quick summary. Just recall that our predictions are probabilities, not binary values.
>
> -- Binary, slow rates (Thm 3.2). Any loss (bounded or unbounded with bounded second moments), e.g.,  square loss and log loss (with bounded second moments).
>
> -- Multiclass [full histogram], slow rates (Thm C.2, L. 1641). Any loss (bounded or unbounded with bounded second moments), e.g., cross entropy (with bounded second moments).
>
> -- Multiclass [total], slow rates (Sect. C.2--this is not detailed, as the analysis coincides with the one for the binary case), e.g, Poisson log loss, square loss and, in general, any regression loss with bounded second moments that admits an affine representation.
>
> -- Binary, fast rates (Thm 3.5). Any loss satisfying condition (6), which is a curvature condition, e.g., square loss and log loss with bounded logits.
>
> -- Currently, we do not have fast rates for multiclass [set as an open question in Sect. 5].

---

> > ### Author Rebuttal · Reviewer_zdnN · 2026-04-07
> >
> > thank you for the rebuttal. i have updated the score.

---

### Official Review · Reviewer_5A4c · 2026-03-13

**Soundness:** 3
**Presentation:** 3
**Significance:** 2
**Originality:** 2
**Overall Recommendation:** 4
**Confidence:** 4

**Summary:**

This paper studies learning from label proportions (LLP), where training data consist of bags of instances with only aggregated label information.

The paper proposes a new estimator constructed by rewriting a standard loss into a form that allows estimation from bag-level proportions. The resulting bag-level loss is an unbiased estimator of the instance-level loss. A key component of the estimator is a centering step, which reduces the variance and enables convergence rates that are independent of the bag size.

To handle unbounded losses, the authors analyze a learning algorithm based on median-of-means (MoM), which leads to improved sample complexity guarantees. The paper also proposes a practical surrogate implementation and evaluates it empirically.

**Compliance With Llm Reviewing Policy:**

Affirmed.

**Key Questions For Authors:**

Question:
1. what's the discrepancy between the theoretical algorithm and the practical implementation? can you quantify that?
2. why report log loss as the evaluation metric? is it the most appropriate metric for this problem?

Minor issues:
1. line 229: "losing functions"
2. Some equations extend beyond the page margin.
3. The paper shall explicitly introduce and define the GENERALUPM procedure in the main text.

**Limitations:**

yes

**Strengths And Weaknesses:**

Strengths:
1. This paper is technically sound. The paper proposes a new centered bag-level loss estimator whose variance is independent of the bag size. This is a technically sound contribution that leads to improved sample-complexity guarantees.
2. The paper extends LLP theory to unbounded losses, this is handled by a median-of-means algorithm, which is an interesting and meaningful theoretical extension.
3. The early sections of the paper are very well-written and clearly motivate the problem.


Weaknesses:
1. My concerns are mainly about the discrepancy between the theoretical algorithm and the practical implementation.

The theoretical guarantees are derived for a Median-of-Means tournament algorithm, whereas the experiments optimize a different surrogate loss (GENERALUPM) using SGD. As a result, the theoretical guarantees do not directly apply to the method evaluated empirically. This gap between theory and practice weakens the empirical validation of the theoretical claims. It would be helpful for the authors to clarify this relationship and, if possible, quantify the difference between the two procedures.

2. The paper is closely related to prior LLP methods (e.g., Busa-Fekete et al., 2023; Li et al., 2024; Busa-Fekete et al., 2025). A more detailed technical comparison (more than just words) with these works would help clarify the novelty of the proposed approach.

3. The experiments mainly report test log loss. It would be helpful to justify why log loss is the most appropriate metric for this problem and to report results for other metrics such as accuracy.

---

> ### Author Rebuttal · Authors · 2026-03-29
>
> We thank Reviewer 5A4c for reading our paper.
>
> - Discrepancy between the theoretical algorithm and the practical implementation.
>
> This is a point raised by all reviewers, but we are responding to each reviewer individually.
>
> The main point of this paper is the design of our centered beg level estimators (for either binary or multiclass), and the analysis of their properties (unbiasedness, variance independent of $k$, etc.). The algorithmics we can build on top of these estimators, whether it's a Median of Means (MoM) tournament, an Empirical Risk Minimizer (ERM), or a Stochastic Gradient Descent (SGD) procedure, is quite standard statistical learning constructions. Crucially, their nuances *are not* specific to LLP, they are rather inherited from the corresponding nuances of the standard supervised setting in statistical learning. In our theoretical analysis, we opted for a MoM tournament to facilitate fast convergence rates, and avoid the limitations of local Rademacher complexity which, to our knowledge, does not readily accommodate unbounded losses.
>
> Had we restricted our analysis to slow rates ($O(1/\sqrt{n})$), we could have employed standard Rademacher complexity guarantees. While these apply to ERM and accommodate unbounded losses with bounded moments, they would not bridge the gap between theory and practice. A generic ERM estimator differs significantly from the one computed by SGD applied to deep networks. Alternatively, we could have analyzed SGD directly using our bag-level loss (2). Yet, such theoretical guarantees would necessitate convexity assumptions--which are typically violated in practice--thus maintaining the disconnect between theoretical analysis and experimental results. Such a disconnect is standard in learning theory papers.
>
> - " . . . optimize a different surrogate loss GeneralUPM . . . ".
>
> We believe there is a misunderstanding here. GeneralUPM is the DNN optimizer operating with the loss defined in Eq. (2), not a different surrogate loss. The parameter $p$ and the two expectations $E[f_1(h(x))]$ and $E[f_2(h(x))]$ in Eq. (2) have been replaced by empirical counterparts--see appendix D.3 for details. We apologize for this crucial confusion and we will fix and clarify this point.
>
> - ``The paper is closely related to prior LLP methods . . .  A more detailed technical comparison (more than just words) . . . would help clarify the novelty . . ."
>
> We are not sure what the reviewer is referring to when advocating for a ``more detailed technical comparison (more than just words)''. This comparison is carried out in the paper already. See, e.g.:
>
> -- LL. 79-100 (left column)
>
> -- L. 208 (left column) to L. 184 (right column)
>
> -- LL. 205--212 (right column)
>
> -- LL. 303 -- 325 (left column).
>
> In short: Compared to Busa-Fekete et al., 2023 and Li et al., 2024, we widen the scope and improve the bounds; compared to Busa-Fekete et al., 2025, we widen the scope our our results (from square loss to general losses, from binary to multiclass). In all cases, the extensions/improvments are nontrivial, as our bag-level estimators are *different* from those in these papers, even when the underlying losses are the same.
>
> - Log loss in the experiments.
>
> The ability of our estimator (2) to handle log-like losses is a main motivation behind this research. In our experiments, we report log loss as our goal is to output probabilities, rather than binary values (see LL. 333-334, right columm), so accuracy would not be fully informative/appropriate. Nonetheless, we report in Appendix D.5 also the results for AUC (which is closer in spirit to accuracy).
>
> - We will fix the minor issues -- thanks!

---

> > ### Author Rebuttal · Reviewer_5A4c · 2026-04-04
> >
> > Thanks to the authors for the clarification. I will keep the score as it is.

---

### Official Review · Reviewer_cGau · 2026-03-14

**Soundness:** 3
**Presentation:** 3
**Significance:** 3
**Originality:** 3
**Overall Recommendation:** 4
**Confidence:** 3

**Summary:**

The paper studies learning from label proportions (LLP) with random, non-overlapping bags. It proposes a centered bag-level estimator for binary losses, gives finite-H sample-complexity guarantees for non-realizable and realizable settings, extends the framework to full-histogram and total multi-class variants, and evaluates a practical SGD-based version (GENERALUPM) against EasyLLP and proportion matching (PM) on binarized MNIST/CIFAR-10, Higgs, Adult, and an online Criteo stream.

**Compliance With Llm Reviewing Policy:**

Affirmed.

**Key Questions For Authors:**

See weaknesses section.

**Limitations:**

Yes

**Strengths And Weaknesses:**

**Strenghts**
- For binary LLP, the proposed centered bag-level estimator is unbiased and has variance independent of bag size
k - removing important LLP bottleneck.
- The binary theory goes beyond square loss and explicitly covers log loss and unbounded losses, this makes the theory much more relevant to practical classification.
- The empirical section spans images, tabular data, and a large online ad-click stream, including large-bag regimes and appendix contains more details (but stuff like hyperparameters are still not reported so hard to say if paper itself is enough to replicate the results).
- In general, the paper is well written and the theory is sound (I haven't checked details)

**Weaknesses**
- If I got this right, the theory and experiments are not aligned: the paper “restricts to the case where” H is finite, but the experiments use neural nets, and the paper explicitly says it “departs from the MoM tournament scheme", so the empirical section does not directly validate the proved guarantees?
- It seems that scope of experiments is narrower than the initial claims “first unified treatment of multi-class LLP”, but experiments “currently restrict to binary classification”, and in total multi-class “cross-entropy does not admit this representation”, so the multi-class claims are only partially demonstrated.
- This is minor, but some equations significantly go over the margin. If moved to the new line, the paper would be ~2 paragraphs longer, exceeding the page limit.

---

> ### Author Rebuttal · Authors · 2026-03-29
>
> We thank Reviewer cGau for reading our paper.
>
> - Alignment between theory and experiments.
>
> This is a point raised by all reviewers, but we are responding to each reviewer individually.
>
> The main point of this paper is the design of our centered beg level estimators (for either binary or multiclass), and the analysis of their properties (unbiasedness, variance independent of $k$, etc.). The algorithmics we can build on top of these estimators, whether it's a Median of Means (MoM) tournament, an Empirical Risk Minimizer (ERM), or a Stochastic Gradient Descent (SGD) procedure, is quite standard statistical learning constructions. Crucially, their nuances *are not* specific to LLP, they are rather inherited from the corresponding nuances of the standard supervised setting in statistical learning. In our theoretical analysis, we opted for a MoM tournament to facilitate fast convergence rates, and avoid the limitations of local Rademacher complexity which, to our knowledge, does not readily accommodate unbounded losses.
>
> Had we restricted our analysis to slow rates ($O(1/\sqrt{n})$), we could have employed standard Rademacher complexity guarantees. While these apply to ERM over infinite hypothesis spaces and accommodate unbounded losses with bounded moments, they would not bridge the gap between theory and practice. A generic ERM estimator differs significantly from the one computed by SGD applied to deep networks. Alternatively, we could have analyzed SGD directly using our bag-level loss (2). Yet, such theoretical guarantees would necessitate convexity assumptions--which are typically violated in practice--thus maintaining the disconnect between theoretical analysis and experimental results. Such a disconnect is standard in learning theory papers.
>
>
> - "First unified treatment of multiclass LLP" vis a vis theory and experiments.
>
> Yes, the experiments only explore the binary case. The unified treatment alluded to above refers to our theoretical findings, not the experiments.
>
> - "cross entropy does not admit this representation".
>
> Note that we describe two multiclass settings that differ by the observations that are available. We call them "total" and "full histogram". The sentence that you cite is relevant only to the total setting (where we get the sum of the labels only).
> Cross entropy can still be handled in the (more standard, but more informative) full histogram setting. In fact, *any* multiclass loss can in principle be handled in the full histogram multiclass setting.

---

> > ### Author Rebuttal · Reviewer_cGau · 2026-04-02
> >
> > Thank you for your response. "The sentence that you cite is relevant only to the total setting" - note I wrote "**total** multi-class". Overall, not surprised with the response, and it mostly confirms points from the review. I will keep my score as it is.

---

### Decision · Program_Chairs · 2026-04-30

**Decision:**

Accept (regular)

**Comment:**

This paper studies learning from label proportions (LLP) and proposes a centered bag-level estimator that achieves unbiasedness and variance independent of bag size. The work further provides generalization guarantees under a broad class of loss functions, including unbounded losses, and extends the analysis to multiclass settings.

Reviewers generally agree that the paper is technically sound and that the estimator and its variance properties constitute a meaningful theoretical contribution. In particular, independence from bag size and the ability to handle general (including unbounded) losses are viewed as notable advances over prior work.

The main concerns raised include:
- Theory–practice gap: mismatch between the MoM tournament (theory) and the SGD-based implementation (experiments).
- Assumptions: how strongly the results support “general loss / multiclass LLP” claims, given restrictions in some settings (e.g., cross-entropy coverage, boundedness/moment assumptions, and limited empirical validation for multiclass).
- Experimental sufficiency: whether the evaluation is sufficiently convincing given the largely binary setup, limited dataset scale, and baseline coverage.

The authors provided a detailed rebuttal, and most concerns were clarified to a reasonable extent. In particular, the discussion around cross-entropy and total multiclass setting resolves some initial misunderstandings. The theory–practice gap, while present, is largely consistent with common patterns in theoretical work and does not undermine the core contribution.

Overall, I agree with the reviewers that the primary strength of the paper lies in its theoretical contributions, which provide a nontrivial and broadly applicable improvement to LLP. While the empirical section is comparatively weaker, it still provides indicative evidence of practical relevance.

**Recommendation: Accept.**

The paper makes a solid and general theoretical contribution that is likely to be of interest to the weak supervision communities. The authors are urged to improve clarity and presentation in the camera-ready version.